# WILDCAT: Near-Linear Attention in Theory and Practice

**Tobias Schröder** [1]   **Lester Mackey** [2]

## Abstract

We introduce WILDCAT, a high-accuracy, low-cost approach to compressing the attention mechanism in neural networks. While attention is a staple of modern network architectures, it is also notoriously expensive to deploy due to resource requirements that scale quadratically with the input sequence length $n$. WILDCAT avoids these quadratic costs by only attending over a small weighted coreset. Crucially, we select the coreset using a fast but spectrally-accurate subsampling algorithm – randomly pivoted Cholesky – and weight the elements optimally to minimise reconstruction error. Remarkably, given bounded inputs, WILDCAT approximates exact attention with super-polynomial $O(n^{-\sqrt{\log(\log(n))}})$ error decay while running in near-linear $O(n^{1+o(1)})$ time. In contrast, prior practical approximations either lack error guarantees or require quadratic runtime to guarantee such high fidelity. We couple this advance with a GPU-optimised PyTorch implementation and a suite of benchmark experiments demonstrating the benefits of WILDCAT for image generation, image classification, and language model KV cache compression.

## 1. Introduction

A central component of transformer-based models (Vaswani et al., 2017) is the attention mechanism, which enables the modelling of long-range dependencies in sequences. The importance of the attention mechanism in today's machine learning landscape cannot be overstated. Practically all large-scale models use this operation, whether in natural language processing (e.g., BERT (Devlin et al., 2019) and GPT (Radford et al., 2018)), image synthesis (Esser et al.,

2021), or protein structure prediction (Jumper et al., 2021). However, attention is also notoriously expensive to deploy as its resource requirements grow quadratically in the input sequence length $n$.

This quadratic cost has motivated the development of fast approximate attention methods, both in theory and in practice. In practice, Reformer (Kitaev et al., 2020), for example, reduces runtime by evaluating a sparse subset of the attention weights while Performer (Choromanski et al., 2021) approximates an assumed low-rank structure of the attention matrix and Scatterbrain (Chen et al., 2021a) combines the two approaches. Meanwhile, in theory, Alman & Song (2023) showed that, for suitably-bounded inputs, one can approximate the attention output with fast, polynomial ($O(n^{-t})$ for any $t > 0$) error decay in near-linear $O(n^{1+o(1)})$ time.

However, a substantial gap remains between the theory and practice. To date, only a few works have developed practical attention approximations with correctness guarantees (Zandieh et al., 2023; Han et al., 2024; Carrell et al., 2025; Han et al., 2025), and the best of these (a) require quadratic time for fast, polynomial error decay and (b) only ensure slow, near-constant $n^{-o(1)}$ error decay in near-linear time.

To bridge the theory-practice gap, we introduce WILDCAT (Weighted Iterative Low-rank Decomposition for Coreset ATtention), a weighted coreset approach to approximate attention that is simultaneously (1) computationally efficient, with a runtime that grows near-linearly in $n$; (2) spectrally-accurate, allowing for super-polynomial error decay; and (3) practical, with an efficient GPU-optimised implementation. Specifically, as our core contributions, we establish the following desirable properties for WILDCAT:

1. WILDCAT avoids the quadratic cost of exact attention by attending only over a small weighted coreset of $r$ input keys. The keys are selected in $O(nr^2)$ time using a parallelised randomly pivoted Cholesky algorithm (Chen et al., 2022a), and reweighted optimally to minimise attention reconstruction error in $O(nrd)$ time. Hence, WILDCAT runs in near-linear time whenever $r \in n^{o(1)}$.

2. Thanks to its selection rule and optimal reweighting, WILDCAT approximates the attention output with near-optimal low-rank-approximation error. As a result, when attention inputs are bounded, a near-constant

[1]Imperial College London [2]Microsoft Research New England. Correspondence to: Tobias Schröder <t.schroeder21@imperial.ac.uk>, Lester Mackey <lmackey@microsoft.com>.

*Proceedings of the 43rd International Conference on Machine Learning*, Seoul, South Korea. PMLR 306, 2026. Copyright 2026 by the author(s).

$r \in n^{o(1)}$ coreset size suffices for *super*-polynomial $O(n^{-\sqrt{\log(\log(n))}})$ error decay.

3. More generally, pushing beyond the limits of prior work on the computational hardness of attention (Alman & Song, 2023; Keles et al., 2023), we show that WILDCAT can deliver super-polynomial error decay in near-linear time even when the input entries or dimensions grow super-logarithmically in $n$.

4. Our benchmark experiments with image generation and image classification show that WILDCAT can generate higher-quality outputs more quickly than five leading attention approximations.

5. Our benchmark experiments with 13 long-context language understanding tasks shows that WILDCAT can also reduce the memory requirements of long-context language models more effectively than five leading KV cache compression methods.

**Notation.** For each $n \in \mathbb{N}$ we define $[n] \triangleq \{1, 2, \ldots, n\}$. For a set $\mathcal{S}$ we write $|\mathcal{S}|$ for the number of elements in the set. We often treat a matrix $\mathbf{A} \in \mathbb{R}^{n \times d}$ as a tuple of row vectors $\mathbf{A} = (\mathbf{a}_i)_{i \in [n]}$ with $\mathbf{a}_i \in \mathbb{R}^d$, and denote sub-selections as $\mathbf{A}_{\mathcal{S}} = (\mathbf{a}_i)_{i \in \mathcal{S}}$. For two such ordered sets $\mathbf{A} = (\mathbf{a}_i)_{i \in [n]}$, $\mathbf{B} = (\mathbf{b}_l)_{l \in [m]} \subseteq \mathbb{R}^d$ and a real-valued function $h : \mathbb{R}^d \times \mathbb{R}^d \to \mathbb{R}$ we write $h(\mathbf{A}, \mathbf{B}) = (h(\mathbf{a}_i, \mathbf{b}_l))_{i \in [n], l \in [m]}$. $\langle \cdot, \cdot \rangle : \mathbb{R}^d \times \mathbb{R}^d \to \mathbb{R}$ denotes the Euclidean inner product. For a symmetric matrix $\mathbf{H} \in \mathbb{R}^{n \times n}$ we denote the pseudo-inverse by $\mathbf{H}^+$ and the $r$-th largest eigenvalue by $\lambda_r(\mathbf{H})$. Further, we define for $\mathbf{A} \in \mathbb{R}^{n \times d}$ the matrix norms $\|\mathbf{A}\|_{\mathrm{op}} \triangleq \sqrt{\lambda_1(\mathbf{A}^\top \mathbf{A})}$, $\|\mathbf{A}\|_{\max} \triangleq \max_{i \in [n], j \in [d]} |\mathbf{A}_{ij}|$, and $\|\mathbf{A}\|_{2,\infty} \triangleq \max_{i \in [n]} \|\mathbf{A}_{i,:}\|_2$.

## 2. Weighted Coreset Attention

The softmax attention mechanism takes as input a sequence of queries $\mathbf{Q} \triangleq (\mathbf{q}_i)_{i \in [m]} \in \mathbb{R}^{m \times d}$, keys $\mathbf{K} \triangleq (\mathbf{k}_l)_{l \in [n]} \in \mathbb{R}^{n \times d}$, and values $\mathbf{V} \triangleq (\mathbf{v}_l)_{l \in [n]} \in \mathbb{R}^{n \times d}$ and outputs the *softmax matrix*

$$\mathbf{O} \triangleq \left( \frac{\sum_{l=1}^n \exp(\beta \langle \mathbf{q}_i, \mathbf{k}_l \rangle) \mathbf{v}_l}{\sum_{l=1}^n \exp(\beta \langle \mathbf{q}_i, \mathbf{k}_l \rangle)} \right)_{i \in [m]} = \mathbf{D}^{-1} \mathbf{A} \mathbf{V} \quad (1)$$

with attention matrix $\mathbf{A}_{il} \triangleq \exp(\beta \langle \mathbf{q}_i, \mathbf{k}_l \rangle)$, scaling matrix $\mathbf{D} \triangleq \mathrm{diag}(\mathbf{A} \mathbf{1}_n)$, and scale factor $\beta$, often chosen as $\beta = 1/\sqrt{d}$. The chief bottleneck in attention is the $m \times n$ attention matrix $\mathbf{A}$. When $m$ and $n$ are comparably large—a common occurrence in vision and language modelling—exact computation of $\mathbf{O}$ requires quadratic $\Theta(n^2 d)$ runtime simply to evaluate and multiply by $\mathbf{A}$ (using standard matrix multiplication).

### 2.1. Low-rank attention approximation

Our high-level strategy to reduce this cost is to approximate the softmax matrix $\mathbf{O}$ using a low-rank approximation of $\mathbf{A}$. Notably, for any $\widehat{\mathbf{A}} = \mathbf{U} \mathbf{W}$ with $\mathbf{U} \in \mathbb{R}^{m \times r}$ and $\mathbf{W} \in \mathbb{R}^{r \times n}$, the plug-in approximation $\widehat{\mathbf{D}}^{-1} \widehat{\mathbf{A}} \mathbf{V} \triangleq \mathrm{diag}(\mathbf{U} \mathbf{W} \mathbf{1}_n)^{-1} \mathbf{U} \mathbf{W} \mathbf{V}$ can be computed with $O(mrd + nrd)$ operations and $O((m+n)(r+d))$ memory by multiplying the weights $\mathbf{W}$ with $(\mathbf{V}, \mathbf{1}_n)$ before applying $\mathbf{U}$. This observation, combined with our next result shows that any low-rank rowwise-accurate estimate of the attention matrix $\mathbf{A}$ can be efficiently transformed into an entrywise-accurate estimate of the softmax matrix $\mathbf{O}$.

**Lemma 1** (Approximate attention guarantee)**.** *Let $\widehat{\mathbf{A}}$ be an approximation to $\mathbf{A}$, $\widehat{\mathbf{D}} = \mathrm{diag}(\widehat{\mathbf{A}} \mathbf{1}_n)$, and $\widehat{\mathbf{O}} \triangleq \mathrm{clip}(\widehat{\mathbf{D}}^{-1} \widehat{\mathbf{A}} \mathbf{V}, \mathbf{v}_{\min}, \mathbf{v}_{\max})$ for $\mathbf{v}_{\min j} = \min_{l \in [n]} \mathbf{v}_{lj}$ and $\mathbf{v}_{\max j} = \max_{l \in [n]} \mathbf{v}_{lj}$. Then*

$$\|\mathbf{O} - \widehat{\mathbf{O}}\|_{\max} \leq \|\mathbf{V}\|_{\max} \min\left( \frac{\frac{3}{\sqrt{n}} \|\mathbf{A} - \widehat{\mathbf{A}}\|_{2,\infty}}{\min\limits_{i \in [m], j \in [n]} \mathbf{A}_{ij}}, 2 \right). \quad (2)$$

In this statement, proved in App. A, we additionally constrain each estimate $\widehat{\mathbf{O}}_{ij}$ to lie in the value range $[\mathbf{v}_{\min j}, \mathbf{v}_{\max j}]$ as the target entry $\mathbf{O}_{ij}$ also satisfies this property. Our next step is to identify a high-quality low-rank approximation $\widehat{\mathbf{A}}$ to seed our output estimate $\widehat{\mathbf{O}}$.

### 2.2. Optimal weighting via Nyström approximation

A widely used tool for constructing low-rank approximations for symmetric positive definite (s.p.d.) matrices is the Nyström method (Williams & Seeger, 2000). To understand how we can exploit approximations of s.p.d. matrices, note that the attention matrix $\mathbf{A} = \exp(\beta \mathbf{Q} \mathbf{K}^\top)$ can be written in terms of the *exponential kernel function*[1] $h(\mathbf{q}, \mathbf{k}) \triangleq \exp(\beta \langle \mathbf{q}, \mathbf{k} \rangle)$.

The kernel perspective suggests the following construction of a low-rank approximation of $\mathbf{A} = h(\mathbf{Q}, \mathbf{K})$. The kernel features $\{h(\cdot, \mathbf{k}_l) \mid l \in [n]\}$ span an (at most) $n$-dimensional vector space $\mathcal{H}$ with inner product $\langle h(\cdot, \mathbf{x}), h(\cdot, \mathbf{y}) \rangle_{\mathcal{H}} = h(\mathbf{x}, \mathbf{y})$. Accordingly, $\{h(\cdot, \mathbf{k}_l) \mid l \in \mathcal{S}\}$ for a subset $\mathcal{S} \subseteq [n]$ with $|\mathcal{S}| = r$ defines an at most $r$-dimensional subspace $\mathcal{H}_{\mathcal{S}} \subset \mathcal{H}$. The orthogonal projection of the kernel features $h(\cdot, \mathbf{k}_l)$ onto $\mathcal{H}_{\mathcal{S}}$ is called a Nyström approximation and takes the form

$$h_{\mathrm{nys}}(\cdot, \mathbf{k}_l) \triangleq h(\cdot, \mathbf{K}_{\mathcal{S}}) h(\mathbf{K}_{\mathcal{S}}, \mathbf{K}_{\mathcal{S}})^+ h(\mathbf{K}_{\mathcal{S}}, \mathbf{k}_l) \in \mathcal{H}_{\mathcal{S}} .$$

The *Nyström weights* $\mathbf{w}_l \triangleq h(\mathbf{K}_{\mathcal{S}}, \mathbf{K}_{\mathcal{S}})^+ h(\mathbf{K}_{\mathcal{S}}, \mathbf{k}_l)$ constitute the *optimal* weighting of $h(\cdot, \mathbf{K}_{\mathcal{S}})$ to minimise the difference between $h(\cdot, \mathbf{k}_l)$ and $h(\cdot, \mathbf{K}_{\mathcal{S}}) \mathbf{w}$. If we adopt the

---

[1]See App. B for relevant background on reproducing kernels.

low-rank approximation $\widehat{\mathbf{A}} = h_{\text{nys}}(\mathbf{Q}, \mathbf{K})$, then the approximation error is governed by the residual kernel function $h_{\text{res}} = h - h_{\text{nys}}$ as $\mathbf{A} - \widehat{\mathbf{A}} = h_{\text{res}}(\mathbf{Q}, \mathbf{K})$. Our next result, proven in App. C, provides a precise guarantee for the rowwise error of this Nyström-based approximation.

**Lemma 2** (Nyström guarantee). *Let $\mathcal{S} \subseteq [n]$ be a subset with $|\mathcal{S}| = r$ and $\mathbf{K}_{\mathcal{S}} = (\mathbf{k}_l)_{l \in \mathcal{S}}$ the associated rows of $\mathbf{K}$. Then, $\widehat{\mathbf{A}} \triangleq h(\mathbf{Q}, \mathbf{K}_{\mathcal{S}})h(\mathbf{K}_{\mathcal{S}}, \mathbf{K}_{\mathcal{S}})^+ h(\mathbf{K}_{\mathcal{S}}, \mathbf{K})$ has rank $\leq r$ and satisfies the following guarantee for $R_{\mathbf{Q}} \triangleq \|\mathbf{Q}\|_{2,\infty}$:*

$$\|\mathbf{A} - \widehat{\mathbf{A}}\|_{2,\infty}^2 \leq \exp(\beta R_{\mathbf{Q}}^2)\, \|h_{\text{res}}(\mathbf{K}, \mathbf{K})\|_{\text{op}}.$$

Lem. 2 shows that, to obtain an entrywise-accurate Nyström estimate of $\mathbf{A}$, it suffices to accurately approximate the s.p.d. key kernel matrix $\mathbf{H} \triangleq h(\mathbf{K}, \mathbf{K})$. Since the quality of a Nyström approximation is determined by the quality of its coreset $\mathbf{K}_{\mathcal{S}}$, we now turn our attention to coreset selection.

## 2.3. Coreset construction with random pivoting

To select a coreset $\mathbf{K}_{\mathcal{S}}$ algorithmically from $\mathbf{K}$ we adapt the randomly pivoted Cholesky (RPC) algorithm of Chen et al. (2022a). RPC builds a partial Cholesky decomposition of the kernel matrix $\mathbf{H}$. Central to its guarantees is the pivoting rule, which samples each coreset point from the diagonal of the current residual kernel. We adopt the same pivoting rule but construct the Nyström weights $\mathbf{W} \triangleq h(\mathbf{K}_{\mathcal{S}}, \mathbf{K}_{\mathcal{S}})^+ h(\mathbf{K}_{\mathcal{S}}, \mathbf{K})$ instead.

Starting with an empty coreset $\mathcal{S} \leftarrow \emptyset$ and the diagonal of the residual kernel $h_{\text{res}}^0(\mathbf{k}_l, \mathbf{k}_l) = h(\mathbf{k}_l, \mathbf{k}_l)$ for each key $\mathbf{k}_l$, we sample in each round a pivot index $s \sim \mathbf{p}^r$, where

$$\mathbf{p}_l^r \triangleq \frac{h_{\text{res}}^r(\mathbf{k}_l, \mathbf{k}_l)}{\sum_{l \in [n]} h_{\text{res}}^r(\mathbf{k}_l, \mathbf{k}_l)} \quad \text{for} \quad l \in [n]. \quad (3)$$

As long as $\mathbf{H}$ is not fully approximated, $h_{\text{res}}^r(\mathbf{k}_s, \mathbf{k}_s) > 0$ by construction, and the kernel matrix associated with the new coreset $\mathcal{S}' \leftarrow \mathcal{S} \cup \{s\}$ remains invertible. We maintain $h(\mathbf{K}_{\mathcal{S}'}, \mathbf{K}_{\mathcal{S}'})^{-1}$ via rank-one updates: Upon adding a pivot $s$, we use for $\mathbf{g}^{\top} \triangleq \frac{(h(\mathbf{k}_s, \mathbf{K}_{\mathcal{S}})h(\mathbf{K}_{\mathcal{S}}, \mathbf{K}_{\mathcal{S}})^{-1}, -1)}{\sqrt{h_{\text{res}}^r(\mathbf{k}_s, \mathbf{k}_s)}}$ the following recursive relations for the inverse of the kernel matrix

$$h(\mathbf{K}_{\mathcal{S}'}, \mathbf{K}_{\mathcal{S}'})^{-1} = \begin{pmatrix} h(\mathbf{K}_{\mathcal{S}}, \mathbf{K}_{\mathcal{S}})^{-1} & 0 \\ 0 & 0 \end{pmatrix} + \mathbf{g}\mathbf{g}^{\top}$$

and the diagonal of the residual kernel for $l \in [n]$

$$h_{\text{res}}^{r+1}(\mathbf{k}_l, \mathbf{k}_l) = h_{\text{res}}^r(\mathbf{k}_l, \mathbf{k}_l) - (\mathbf{g}^{\top} h(\mathbf{K}_{\mathcal{S}}, \mathbf{k}_l))^2.$$

We provide a justification for the recursive update rule in App. K. The complete algorithm, RPNYS, is summarised in Alg. 1. Note that RPNYS only accesses $O(nr)$ entries of $h(\mathbf{K}, \mathbf{K})$ and runs in $O(nr^2 + nrd)$ time.

In App. D, we mildly adapt the arguments of Epperly & Moreno (2023, Thm. 7) to provide the following operator norm guarantee for RPNYS.

---

**Algorithm 1:** Randomly pivoted Nyström (RPNYS)

**Input:** dataset $\mathbf{K} = (\mathbf{k}_l)_{l \in [n]}$, kernel $h$, rank $r$
$\mathbf{p} \leftarrow (h(\mathbf{k}_l, \mathbf{k}_l))_{l \in [n]}$ ▷ *Compute kernel diagonal*
$\mathbf{M} \leftarrow \mathbf{0}_{r \times r}; \quad \mathbf{R} \leftarrow \mathbf{0}_{r \times n}; \quad \mathbf{g} \leftarrow \mathbf{0}_r$
$\mathcal{S} \leftarrow \emptyset$ ▷ *Initialize empty coreset*
**for** $i = 1, \ldots, r$ **do**
$\quad \mathcal{S} \leftarrow \mathcal{S} \cup \{s\} \quad$ for $\quad s \sim \frac{\mathbf{p}}{\sum_{l=1}^n \mathbf{p}_l}$ ▷ *Sample pivot*
$\quad$ // Update kernel inverse
$\quad$ **if** $i > 0$ **then**
$\quad\quad \mathbf{g}_{[i-1]} \leftarrow \mathbf{M}_{[i-1],[i-1]}\mathbf{R}_{[i-1],s}$ ▷ $O(i^2)$ *operations*
$\quad$ **end if**
$\quad \mathbf{g}_i \leftarrow -1; \quad \mathbf{g}_{[i]} \leftarrow \mathbf{g}_{[i]}/\sqrt{\mathbf{p}_s}$
$\quad \mathbf{M} \leftarrow \mathbf{M} + \mathbf{g}\mathbf{g}^{\top}$
$\quad$ // Update pivot distribution
$\quad \mathbf{R}_{i,[n]} \leftarrow h(\mathbf{k}_s, \mathbf{K})$ ▷ $O(nd)$ *operations*
$\quad \delta \leftarrow \mathbf{g}_{[i]}^{\top}\mathbf{R}_{[i],[n]}$ ▷ $O(ni)$ *operations*
$\quad \mathbf{p} \leftarrow \mathbf{p} - \delta^2$ ▷ *Entrywise power function*
$\quad \mathbf{p}_s \leftarrow 0$ ▷ *For numerical stability*
**end for**
**Return:** Coreset $\mathbf{K}_{\mathcal{S}}$; Nyström weights $\mathbf{W} = \mathbf{M}\mathbf{R}$

---

**Theorem 1** (RPNYS guarantee). *Fix any $\varepsilon > 0$ and consider the coreset $\mathbf{K}_{\mathcal{S}}$ and Nyström weights $\mathbf{W}$ outputted by* RPNYS *(Alg. 1) with kernel $h$, dataset $\mathbf{K}$, and rank parameter $r$. If $\mathbf{H} \triangleq h(\mathbf{K}, \mathbf{K})$ and $\widehat{\mathbf{H}}^r \triangleq h(\mathbf{K}, \mathbf{K}_{\mathcal{S}})\mathbf{W}$, then $\mathbb{E}\|\mathbf{H} - \widehat{\mathbf{H}}^r\|_{\text{op}} \leq \varepsilon$ whenever, for some $\mathbf{T} \preceq \mathbf{H}$,*

$$r \geq \text{rank}(\mathbf{T})\log\left(\frac{\|\mathbf{H}\|_{\text{op}}}{\varepsilon}\right) + \text{tr}(\mathbf{H} - \mathbf{T})\left(\frac{1}{\varepsilon} - \frac{1}{\|\mathbf{H}\|_{\text{op}}}\right).$$

Thm. 1 links the approximation error of the randomly pivoted Nyström method to the approximability of $\mathbf{H}$ by *any* low-rank operator $\mathbf{T} \preceq \mathbf{H}$. In Lem. E.2 we show that an order $s$ Taylor approximation of the exponential function yields an operator $\mathbf{T}^s$ with rank $\leq \binom{s+d}{d}$ and

$$\text{tr}(\mathbf{H} - \mathbf{T}^s) \leq n\exp(\beta\|\mathbf{K}\|_{2,\infty}^2)\left(\frac{e\beta\|\mathbf{K}\|_{2,\infty}^2}{s+1}\right)^{s+1}$$

The fast decay of this approximation in the order parameter $s$ will allow us to derive fast-decaying error rates for the RPNYS approximation in Sec. 3.

## 2.4. Invariance of attention under shift and rescaling

Interestingly, the softmax matrix $\mathbf{O} = (\mathbf{o}_i)_{i \in [m]}$ is invariant under a global recentring of the keys. This follows as

$$\mathbf{o}_i = \frac{h(\mathbf{q}_i, \mathbf{K})\mathbf{V}}{h(\mathbf{q}_i, \mathbf{K})\mathbf{1}_n}\frac{\exp(-\beta\langle\mathbf{q}_i, \bar{\mathbf{k}}\rangle)}{\exp(-\beta\langle\mathbf{q}_i, \bar{\mathbf{k}}\rangle)} = \frac{h(\mathbf{q}_i, \mathbf{K} - \mathbf{1}_n\bar{\mathbf{k}}^{\top})\mathbf{V}}{h(\mathbf{q}_i, \mathbf{K} - \mathbf{1}_n\bar{\mathbf{k}}^{\top})\mathbf{1}_n}$$

for any row vector $\bar{\mathbf{k}} \in \mathbb{R}^d$ and each $i \in [m]$. Hence our approximation algorithms are also free to operate on recentred keys. Hereafter, we choose $\bar{\mathbf{k}} \triangleq \frac{1}{n}\sum_{i=1}^n \mathbf{k}_i$ and treat $\mathbf{K}$ as the matrix of recentred keys, $\mathbf{k}_i - \bar{\mathbf{k}}$.

Note also that the attention matrix $\mathbf{A}$ is invariant under rescalings of the keys and queries, i.e., we are allowed to

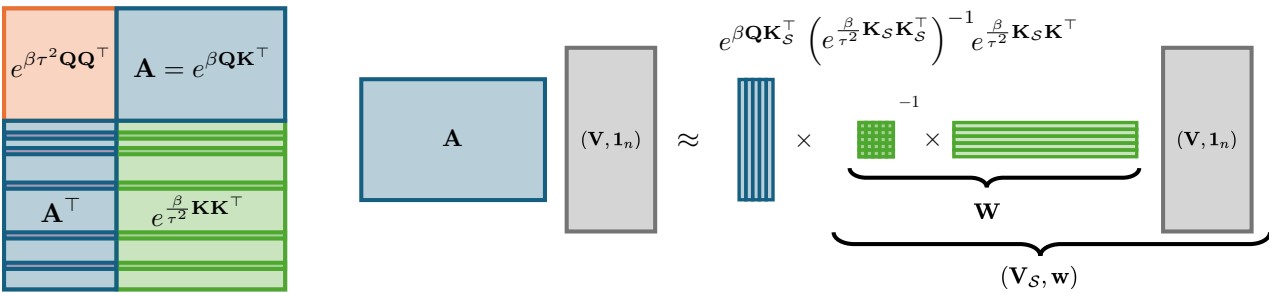

*Figure 1.* **Visualisation of the WILDCAT methodology.** Our goal is the approximation of the off-diagonal block $\mathbf{A}$ through a Nyström approximation $\widehat{\mathbf{A}}_\tau \triangleq h(\mathbf{Q}, \mathbf{K}_\mathcal{S}) h(\frac{1}{\tau}\mathbf{K}_\mathcal{S}, \frac{1}{\tau}\mathbf{K}_\mathcal{S})^{-1} h(\frac{1}{\tau}\mathbf{K}_\mathcal{S}, \frac{1}{\tau}\mathbf{K})$. With the right order of operations, the computation cost for $\mathbf{AV}$ decreases from $O(mnd)$ to $O(rnd + mrd + nr^2)$. For the exponential kernel, the off-diagonal block is invariant under $\mathbf{Q} \to \tau\mathbf{Q}$, $\mathbf{K} \to \frac{1}{\tau}\mathbf{K}$. Since we only select coreset points from $\mathbf{K}$, we can optimise for low-rank approximability.

rescale $\mathbf{K} \leftarrow \tau^{-1}\mathbf{K}$ and $\mathbf{Q} \leftarrow \tau\mathbf{Q}$ without changing $\mathbf{A}$. However, when we form our approximation of $\mathbf{A}$ using rescaled keys and queries, the approximation error is affected in two ways. On the one hand, increasing $\tau$ makes the data matrix $\mathbf{H}_\tau \triangleq h(\tau^{-1}\mathbf{K}, \tau^{-1}\mathbf{K})$ increasingly low-rank approximable. In fact, in the extreme case $\mathbf{H}_\tau \xrightarrow{\tau \to \infty} \mathbf{1}_n\mathbf{1}_n^\top$ becomes a rank one matrix. On the other hand, $\tau$ increases the query-based error inflation factor $\exp(\beta\tau^2 R_\mathbf{Q}^2)$ of Lem. 2. For $R_\mathbf{K} \triangleq \|\mathbf{K}\|_{2,\infty}$, our theory in Sec. 3 suggests the following closed-form rescaling parameter that reflects the asymmetric roles of the keys and queries in our attention reconstruction:

$$\tau \triangleq \sqrt{\frac{R_\mathbf{K}}{R_\mathbf{Q}} \frac{b_0}{2W_0(b_0/(2\rho_0))}} \quad \text{with} \quad b_0 \triangleq \frac{\log(n)}{\beta R_\mathbf{Q} R_\mathbf{K}} + 2. \quad (4)$$

Here, $z \mapsto W_0(z)$ denotes the Lambert-W function which is defined as the solution to $z = we^w$ (see App. L for more details), and $\rho_0 \triangleq \sqrt{1 + e^{W_0(2/e^2)+2}} \approx 3.19$. We use the temperature scaling (4) in both our theory and experiments.

### 2.5. Maximising throughput via binning

The sequential nature of RPNYS does not make full use of the massive parallelism available on a GPU. To maximise throughput, we employ a divide-and-conquer strategy that partitions the input into $B$ bins and identifies a coreset of size $r/B$ for each bin in parallel. With this strategy the total operation count is reduced to $O(nr^2/B^2 + nrd/B)$ while the parallel runtime is even faster, $O(nr^2/B^3 + nrd/B^2)$. Our guarantees in Sec. 3 account for this binning and allow the user to flexibly trade off between speed and the level of guaranteed accuracy.

### 2.6. Summary of the methodology

Our core methodology is summarised in COMPRESSKV (Alg. 2) and visualised in Fig. 1. After a low-cost recentring and temperature selection using (4), the keys serve as input data for the RPNYS algorithm with kernel function $h_\tau \triangleq \exp(\beta\langle\cdot,\cdot\rangle/\tau^2)$. Using the obtained coreset indices $\mathcal{S}$ and

the Nyström weights $\mathbf{W} \triangleq h_\tau(\mathbf{K}_\mathcal{S}, \mathbf{K}_\mathcal{S})^{-1} h_\tau(\mathbf{K}_\mathcal{S}, \mathbf{K})$, we form the compressed key and value tensors $\mathbf{K}_\mathcal{S} \triangleq (\mathbf{k}_l)_{l \in \mathcal{S}}$, $\mathbf{V}_\mathcal{S} \triangleq \mathbf{W}\mathbf{V} \in \mathbb{R}^{r \times d}$. Note that *all* keys and values are involved in the compression of $\mathbf{V}$. In addition, we form the new softmax normalisation vector $\mathbf{w} \triangleq \mathbf{W}\mathbf{1}_n$ to compute $\widehat{\mathbf{D}}_i = \sum_{l \in \mathcal{S}} \exp(\beta\langle\mathbf{q}_i, \mathbf{k}_l\rangle)\mathbf{w}_l$.

---

**Algorithm 2: COMPRESSKV**

**Input:** keys $\mathbf{K}$, values $\mathbf{V}$, radius $R_\mathbf{Q}$, scale $\beta$, rank $r$, bins $B$
$\bar{\mathbf{k}} \leftarrow \mathbf{K}.\texttt{rowsmean}()$; $\quad \mathbf{K} \leftarrow \mathbf{K} - \bar{\mathbf{k}}$ $\quad \triangleright$ *Recenter keys*
Evenly divide (or reshape) rows of $\mathbf{K}$ into bins $\mathbf{K}^1, \dots, \mathbf{K}^B$
**for** $b = 1, \dots, B$ **do in parallel**
$\quad R_\mathbf{K} \leftarrow \max_{l \in [n]} \sqrt{\sum_{j=1}^d (\mathbf{K}_{l,j}^b)^2}$
$\quad \tau \leftarrow \texttt{getTemperature}(\beta, R_\mathbf{Q}, R_\mathbf{K}, n)$ using (4)
$\quad \mathbf{K}_\mathcal{S}^b, \mathbf{W}^b \leftarrow \text{RPNYS}(\mathbf{K}^b, \exp(\beta\langle\cdot,\cdot\rangle/\tau^2), r/B)$
**end**
// Concatenate (or reshape) bin results
$\mathbf{K}_\mathcal{S} \leftarrow (\mathbf{K}_\mathcal{S}^b)_{b \in [B]}$; $\mathbf{W} \leftarrow (\mathbf{W}^b)_{b \in [B]}$
$\mathbf{K}_\mathcal{S} \leftarrow \mathbf{K}_\mathcal{S} + \bar{\mathbf{k}}$
$\mathbf{V}_\mathcal{S} \leftarrow \mathbf{W}\mathbf{V}$; $\mathbf{w} \leftarrow \mathbf{W}\mathbf{1}_n$ $\quad \triangleright$ *Compress values*
**Return:** $\mathbf{K}_\mathcal{S}, \mathbf{V}_\mathcal{S}, \mathbf{w}$

---

**Algorithm 3: WTDATTN**

**Input:** queries $\mathbf{Q}$, keys $\mathbf{K}_\mathcal{S}$, values $\mathbf{V}_\mathcal{S}$, weights $\mathbf{w}$,
$\quad\quad$ range $(\mathbf{v}_{\min}, \mathbf{v}_{\max})$, scale $\beta$
$\widehat{\mathbf{A}} \leftarrow \exp(\beta\mathbf{Q}\mathbf{K}_\mathcal{S}^\top)$
$\widehat{\mathbf{O}} \leftarrow \text{diag}(\widehat{\mathbf{A}}\mathbf{w})^{-1}\widehat{\mathbf{A}}\mathbf{V}_\mathcal{S}$ where $\widehat{\mathbf{A}}\mathbf{w} > 0$ else 0
**Return:** $\widehat{\mathbf{O}} \leftarrow \text{clip}(\widehat{\mathbf{O}}, \mathbf{v}_{\min}, \mathbf{v}_{\max})$

---

**Algorithm 4: WILDCAT**

**Input:** queries $\mathbf{Q}$, keys $\mathbf{K}$, values $\mathbf{V}$, scale $\beta$, rank $r$, bins $B$
$(\mathbf{v}_{\min}, \mathbf{v}_{\max}) \leftarrow (\min_{l \in [n]} \mathbf{V}_{l,[d]}, \max_{l \in [n]} \mathbf{V}_{l,[d]})$
$R_\mathbf{Q} \leftarrow \max_{l \in [n]} \sqrt{\sum_{j=1}^d \mathbf{Q}_{l,j}^2}$
$\mathbf{K}_\mathcal{S}, \mathbf{V}_\mathcal{S}, \mathbf{w} \leftarrow \text{COMPRESSKV}(\mathbf{K}, \mathbf{V}, R_\mathbf{Q}, \beta, r, B)$
**Return:** $\widehat{\mathbf{O}} \triangleq \text{WTDATTN}(\mathbf{Q}, \mathbf{K}_\mathcal{S}, \mathbf{V}_\mathcal{S}, \mathbf{w}, \mathbf{v}_{\min}, \mathbf{v}_{\max}, \beta)$

---

In autoregressive models, keys and values of previously processed tokens are stored in KV caches which can incur prohibitive $\Omega(nd)$ memory requirements. In this context (often called the *prefill phase*), we will use COMPRESSKV

for *KV cache compression*, requiring only $O(rd)$ in storage for the output.[2] The compressed keys and values can then be incorporated into any subsequent attention computations (e.g., to generate new tokens in the *decoding phase*) using the weighted attention forward pass, WTDATTN (Alg. 3).

For non-autoregressive models, we embed COMPRESSKV and WTDATTN into our custom attention module WILDCAT (Alg. 4). In the canonical attention approximation setting with $m \sim n$, WILDCAT enjoys $O(nr^2 + nrd)$ runtime, which is near-linear for $r \in n^{o(1)}$.

## 3. Approximation Guarantees

We next derive efficient attention approximation guarantees for WILDCAT based on the high quality and low runtime of RPNYS. Recall that Thm. 1 allows us to bound the error of RPNYS in terms of any benchmark approximation $\mathbf{T} \preceq \mathbf{H}_\tau$. To obtain a concrete bound, we consider $\mathbf{T}^s$ induced by an order $s$ Taylor approximation of the exponential function:

$$\mathbf{T}^s_{il} \triangleq \sum_{p=0}^{s} \frac{1}{p!}\left(\frac{\beta}{\tau^2}\langle \mathbf{k}_i, \mathbf{k}_l\rangle\right)^p.$$

Our next lemma, proved in App. E, characterises the trade-off between order and approximation accuracy.

**Lemma 3** (Taylor guarantee). *Define the order parameter*

$$\tilde{s}(\varepsilon) \triangleq \frac{\log(n/\varepsilon) + \beta R_{\mathbf{K}}^2/\tau^2}{W_0\left(\frac{\log(n/\varepsilon)\tau^2}{e\beta R_{\mathbf{K}}^2} + \frac{1}{e}\right)}$$

*for $\varepsilon > 0$ where $W_0$ is the primary branch of the Lambert-W function. Then, $\mathrm{tr}(\mathbf{H}_\tau - \mathbf{T}^s) \leq \varepsilon$ for all $s \geq \lfloor\tilde{s}(\varepsilon)\rfloor$.*

Meanwhile, Lem. 4, proved in App. F, bounds the rank of $\mathbf{T}^s$ in terms of its order.

**Lemma 4** (Taylor rank bound). *For any $s \in \mathbb{N}$,*

$$\mathrm{rank}(\mathbf{T}^s) \leq \frac{1}{\sqrt{\pi}} n^{(\sigma+\delta)\mathrm{Ent}\left(\frac{\sigma}{\sigma+\delta}\right)} \text{ for } (\sigma, \delta) \triangleq \left(\frac{s}{\log(n)}, \frac{d}{\log(n)}\right)$$

*and $\mathrm{Ent}(p) \triangleq -p\log(p) - (1-p)\log(1-p)$.*

Combining Lems. 1 to 4 with Thm. 1, we arrive at the following attention approximation guarantee for WILDCAT (proved in App. H).

**Theorem 2** (WILDCAT guarantee). *Let $\widehat{\mathbf{O}}_r$ be the output of WILDCAT (Alg. 4) with rank parameter $r$ and $B = 1$. Fix $a \geq \frac{1}{2}$ and define the entry and dimension growth parameters,*

$$\gamma \triangleq \frac{\beta R_{\mathbf{Q}} R_{\mathbf{K}}}{\log(n)} \quad \text{and} \quad \delta \triangleq \frac{d}{\log(n)},$$

---

[2]While we focus on its memory reduction benefits, KV cache compression also has a complementary computational benefit: $m$ new tokens can be generated in $O(rmd + m^2 d)$ time instead of $\Theta(nmd + m^2 d)$ time.

*Table 1.* **Practical approximation guarantees.** For each approximation $\widehat{\mathbf{O}}$ to the softmax matrix $\mathbf{O}$ (1) with $m = n$, we report, up to constants, the best worst-case error bound on $\|\mathbf{O} - \widehat{\mathbf{O}}\|_{\max}$ given bounded dimension $d \in O(1)$, bounded entries $\beta R_{\mathbf{Q}}^2, \beta R_{\mathbf{K}}^2 \leq R^2 \in O(1)$, and $O(dn^{1+t})$ runtime. Here, the ratios $\|\mathbf{V}\|_{\mathrm{op}}/\|\mathbf{V}\|_{\max}$ and $\|\mathbf{V}\|_F/\|\mathbf{V}\|_{\max}$ lie in $[1, \sqrt{nd}]$, $\xi \triangleq 0.173 + o(1)$, and $\kappa \triangleq e^{-1}(2\rho_0 + 1)$. See App. J for the proof of each guarantee.

| Approximation | Guarantee |
|---|---|
| **Thinformer** | $\frac{\sqrt{\log(\|\mathbf{V}\|_{\max})}\log n}{n^t} \cdot \|\mathbf{V}\|_{2,\infty}$ |
| **BalanceKV** | $\frac{(\log n)^3}{n^t} \cdot \|\mathbf{V}\|_F$ |
| **KDEformer** | $\frac{n^{\xi/2}}{n^{t/2}} \cdot \|\mathbf{V}\|_{\mathrm{op}}$ |
| **HyperAttention** | $\frac{(\log n)^{1/6}}{n^{t/6}} \cdot \|\mathbf{V}\|_{\mathrm{op}}$ |
| **WILDCAT** | $\frac{\log n}{n^{0.14t\log(e+\log(n)/(\kappa R))}} \cdot \|\mathbf{V}\|_{\max}$ |

*along with the Taylor growth parameter*

$$\sigma \triangleq \frac{a+\gamma}{W_0\left(\frac{1}{2\rho_0\gamma} + \frac{1}{\rho_0}\right)}. \tag{5}$$

*Then, $\mathbb{E}\|\mathbf{O} - \widehat{\mathbf{O}}_r\|_{\max} \leq 3\|\mathbf{V}\|_{\max} n^{-a}$ provided*

$$r \geq 1 + \frac{1}{\sqrt{\pi}} n^{(\sigma+\delta)\mathrm{Ent}\left(\frac{\sigma}{\sigma+\delta}\right)} \log\left(n^{2a+\sigma+3\gamma}\right). \tag{6}$$

*For $B > 1$, the same result holds with the effective sequence length and rank $(n_{\mathrm{eff}}, r_{\mathrm{eff}}) = (\lfloor\frac{n}{B}\rfloor, \lceil\frac{r}{B}\rceil)$ in place of $(n, r)$.*

Thm. 2 lets us easily identify conditions under which WILDCAT guarantees super-polynomial accuracy in near-linear time. For example, in Tab. 1, we compare the error decay guarantees of Thm. 2 with those of various practical attention approximations assuming bounded dimension, bounded entries, $m = n$, and $O(dn^{1+t})$ runtime. Notably, Thinformer (Carrell et al., 2025), BalanceKV (Han et al., 2025), KDEformer (Zandieh et al., 2023), and HyperAttention without masking (Han et al., 2024) all guarantee at best polynomial error decay, while WILDCAT provides *super-polynomial* $O(n^{-\Omega(\log(\log(n)))t})$ error decay.

Plugging in $t = 8/\sqrt{\log\log(n)}$, we observe that WILDCAT can even deliver super-polynomial $O(n^{-\sqrt{\log\log(n)}})$ error decay in near-linear $O(dn^{1+o(1)})$ time. In fact, this remains true even when the entries and dimension are allowed to grow with the sequence length:

**Corollary 1** (Super-polynomial error decay in near-linear time). *Under the assumptions of Thm. 2, suppose $\beta R_{\mathbf{Q}} R_{\mathbf{K}} \in O(\log(n)^\alpha)$ with $\alpha \in (0, 1)$, $d \in o(\log(n))$, and $a(n) \in o(\log\log(n))$. Then*

$$\mathbb{E}\|\mathbf{O} - \widehat{\mathbf{O}}_r\|_{\max} \leq 3\|\mathbf{V}\|_{\max} n^{-a(n)} \text{ for some } r \in n^{o(1)}.$$

*Proof.* For $\beta R_{\mathbf{Q}} R_{\mathbf{K}} \in O(\log(n)^{\alpha})$ and $a(n) \in o(\log(\log(n)))$, a short calculation (see Lem. I.2) shows that there exists a $c > 0$ and $n_0 > 0$ such that $\sigma(n) \leq c \frac{a(n)}{\log(1+\gamma(n)^{-1})} \leq c \frac{a(n)}{\alpha \log(\log(n))} \in o(1)$ for all $n > n_0$. Furthermore, by definition, $\delta = d/\log(n) \in o(1)$. Therefore, we obtain from Thm. 2 that it is sufficient to take $r \sim n^{(\sigma+\delta)\mathrm{Ent}\left(\frac{\sigma}{\sigma+\delta}\right)} \log(n^{2a+\sigma+3\gamma}) \in n^{o(1)}$ to guarantee $n^{-a(n)}$ error decay. $\square$

Meanwhile, the HyperAttention, KDEformer, and Thinformer guarantees deliver, at best, near-constant $n^{-o(1)}$ error in near-linear time and require quadratic time to guarantee super-polynomial error decay.

While we followed Alman & Song (2023); Keles et al. (2023); Carrell et al. (2025) in stating entrywise error guarantees, our results also improve upon the operator norm guarantees established for KDEformer and Hyperattention and the $\|\cdot\|_{2,\infty}$ guarantees established for BalanceKV. Indeed, even using the lossy conversions $\|\mathbf{O} - \widehat{\mathbf{O}}\|_{\mathrm{op}} \leq \sqrt{nd}\|\mathbf{O} - \widehat{\mathbf{O}}\|_{\max}$ and $\|\mathbf{O} - \widehat{\mathbf{O}}\|_{2,\infty} \leq \sqrt{d}\|\mathbf{O} - \widehat{\mathbf{O}}\|_{\max}$, our guarantees for $\|\mathbf{O} - \widehat{\mathbf{O}}\|_{\max}$ imply super-polynomial decay in near-linear time for the other norms. Meanwhile, the guarantees for prior work remain exactly as in Tab. 1 with each requiring quadratic time for super-polynomial error decay and achieving at best sub-polynomial, near-constant decay in near-linear time.

Pushing beyond the limits of prior work on the computational hardness of attention (Alman & Song, 2023; Keles et al., 2023), our next corollary shows that WILD-CAT can achieve super-polynomial error decay in near-linear time even when the dimension or entries grow super-logarithmically in $n$. Our proof in App. I uses the entropy factor in Thm. 2 to refine the analysis in Cor. 1.

**Corollary 2** (Refined super-polynomial error decay in near–linear time)**.** *Instantiate the assumptions of Thm. 2. If* $\gamma(n) \triangleq \frac{\beta R_{\mathbf{Q}} R_{\mathbf{K}}}{\log(n)} \in o(1)$, $\delta(n) \triangleq \frac{d}{\log(n)}$, *and*

$$a(n) \in o\left(\frac{\log(1/\gamma(n))}{\max\{\log(\delta(n)), 1\}}\right) \cap n^{o(1)},$$

*then* $\mathbb{E}\|\mathbf{O} - \widehat{\mathbf{O}}_r\|_{\max} \leq 3\|\mathbf{V}\|_{\max} n^{-a(n)}$ *for some* $r \in n^{o(1)}$. *The same conclusion holds if, alternatively,* $\delta(n) \in o(1)$, $\gamma(n) \in \Omega(1) \cap n^{o(1/d)}$, *and* $a(n) \in n^{o(1/d)}$.

Let us consider two important implications of Cor. 2. First, when $d$ is bounded (the typical case when one is working with a fixed model and focused on increasing its context length), Cor. 2 supports unbounded entries with *any* form of near-constant $\beta R_{\mathbf{Q}} R_{\mathbf{K}} \in n^{o(1)}$ growth, even superlogarithmic $\omega(\log(n))$ growth. In contrast, the near-linear-time theory of Alman & Song (2023, Thm. 3.8) only guarantees polynomial error decay for $\beta R_{\mathbf{Q}} R_{\mathbf{K}} \in o(\log n)$.

Second, Cor. 2 also supports super-logarithmic dimension growth, $d = \omega(\log n)$. For example, when the scaled entries $\beta R_{\mathbf{Q}} R_{\mathbf{K}}$ are bounded, any choice of $d = \log(n) \exp(o(\log(\log n)))$ still leads to super-polynomial decay in near-linear time. In contrast, the near-linear-time theory of Alman & Song (2023) only guarantees polynomial error decay for $d \in O(\log n)$. Interestingly, Keles et al. (2023) also provide a *lower* bound on the speed of attention approximation when $d \in \omega(\log n)$. Assuming the strong exponential-time hypothesis (a widely-believed conjecture in complexity theory), Keles et al. (2023, Thm. 6) shows that, for any $\varepsilon > 0$, approximating attention to absolute error $n^{-3d} e^{-3d^2}$ with $d = \omega(\log n)$ requires $\Omega(n^{2-\varepsilon})$ time. However, this lower bound still allows for super-polynomial error decay in near-linear time like that established by Cor. 2.

**Guarantees for KV cache compression:** The COM-PRESSKV methodology enables us to reduce the memory footprint of KV caches from linear $\Theta(nd)$ to near-constant $\mathcal{O}(rd)$ whenever $r \in n^{o(1)}$. Long-context inference via WTDATTN with the compressed cache $(\mathbf{K}_{\mathcal{S}}, \mathbf{V}_{\mathcal{S}}, \mathbf{w})$ then still obeys the accuracy guarantees of Thm. 2 and Cor. 2.

### 3.1. Additional related work

Two other fast attention methods use the Nyström method in their methodology: Xiong et al. (2021) approximate the matrix $\mathbf{D}^{-1}\mathbf{A}$ with a Nyström approximation, directly. The evaluation of any entry in $\mathbf{D}^{-1}\mathbf{A}$ requires the realisation of $m \times n$ entries. Nyströmformer therefore requires additional sketches of the softmax matrix and does not offer strong guarantees. Chen et al. (2021b), on the other hand, use the Nyström method to approximate a distinct Gaussian attention mechanism. Finally, Chen et al. (2022b) uses sketching methods to construct low-rank approximations of the attention mechanism.

Our method relies on the reduction of a dataset to a representative weighted coreset with similarity of datapoints measured by the attention kernel. From this standpoint, it falls into the class of *distribution compression* methods which aim to succinctly summarize an empirical or population distribution using a small collection of representative points and into the class of *kernel quadrature* methods which aim to accurately approximate expectations of functions in an reproducing kernel Hilbert space (Aronszajn, 1950). Early works that use kernels for distribution compression with unweighted coresets include Dwivedi & Mackey (2024; 2022); Shetty et al. (2022); Gong et al. (2024); Carrell et al. (2025). Additional strategies to achieve compression with guarantees for weighted coresets were proposed in Hayakawa et al. (2022); Epperly & Moreno (2023); Li et al. (2024a). A method to approximate ratios of kernel sums using the Nyström method was explored empirically in Gong et al. (2024) but without providing guarantees.

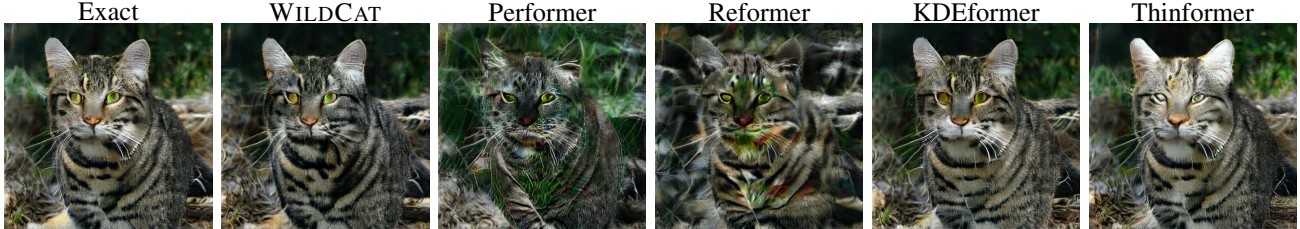

Exact     WILDCAT     Performer     Reformer     KDEformer     Thinformer

*Figure 2.* **Example generations from BigGAN with exact or approximate attention.**

*Table 2.* **Quality of attention approximations for BigGAN image generation.** We report speed-ups over 10 batches of 32 images and mean degradation ($\pm 1$ standard deviation across five seeds) of the Inception Score (IS) and Frechet Inception Distance (FID) between 5K generations and the ImageNet 2012 validation set.

| Attention Algorithm | Speed-up over Exact | IS Degradation (%) | FID Degradation (%) |
|:---:|:---:|:---:|:---:|
| **Reformer** | $0.69\times$ | $66.55 \pm 0.52$ | $124.20 \pm 1.19$ |
| **ScatterBrain** | $1.75\times$ | $36.77 \pm 0.50$ | $20.87 \pm 1.25$ |
| **Performer** | $3.46\times$ | $35.14 \pm 0.82$ | $4.01 \pm 0.91$ |
| **KDEformer** | $0.72\times$ | $2.02 \pm 0.81$ | $\mathbf{0.00 \pm 0.00}$ |
| **Thinformer** | $2.32\times$ | $1.79 \pm 0.31$ | $\mathbf{0.00 \pm 0.00}$ |
| **WILDCAT** | $\mathbf{4.33\times}$ | $\mathbf{1.22 \pm 0.87}$ | $\mathbf{0.00 \pm 0.00}$ |

## 4. Experiments

We now turn to an empirical evaluation of our new tools on a suite of standard approximate attention benchmarks. See

https://github.com/microsoft/wildcat

for open-source PyTorch (Paszke et al., 2019) code recreating all experiments and App. M for supplementary experiment details.

### 4.1. Benchmarking image generation

We begin with the BigGAN image generation benchmark of Carrell et al. (2025). BigGAN (Brock et al., 2019) is a generative adversarial network for image generation containing a single attention layer, which, for images of size $512 \times 512$, has input tensors $\mathbf{Q} \in \mathbb{R}^{4096 \times 64}$, $\mathbf{K} \in \mathbb{R}^{1024 \times 64}$, and $\mathbf{V} \in \mathbb{R}^{1024 \times 256}$. The BigGAN benchmark evaluates the quality of attention approximations used as drop-in replacements for exact attention in a BigGAN model pretrained on ImageNet (Deng et al., 2009). Using the settings and implementations provided by Carrell et al. (2025), we benchmark WILDCAT (with $r = 96$ and $B = 8$) against exact attention and five leading approximate attention mechanisms: Reformer, ScatterBrain, Performer, KDEformer, and Thinformer. Fig. 2 displays example generations, and Tab. 2 reports the gain in speed and the loss in quality from using each approximation to generate 5000 images. We observe that WILDCAT yields the largest speed-up ($4.33\times$), the smallest degradation in Inception Score (IS, Salimans et al., 2016) (just 1.22%), and, surprisingly, no degradation in Frechet Inception Distance (FID, Heusel et al., 2017).

### 4.2. Benchmarking image classification

We next replicate the Tokens-to-Token Vision Transformer (T2T-ViT) image classification benchmark of Carrell et al. (2025), where attention approximations are used as drop-in replacements for exact attention in the computationally demanding tokens-to-token module. T2T-ViT (Yuan et al., 2021) splits an input image into a large number of overlapping patches which are progressively reduced to a smaller number of tokens by two attention layers. The T2T-ViT benchmark uses a model pretrained on ImageNet with images of size $224 \times 224$, layers of size $(n_1, d_1) = (3136, 64)$ and $(n_2, d_2) = (784, 64)$, and a computational cost dominated by the larger first layer.

In Tab. 3, we benchmark WILDCAT, with $(r_1, B_1) = (224, 224)$ for the first layer and $(r_2, B_2) = (196, 196)$ for the second, against exact attention and the five leading approximate attention mechanisms of Sec. 4.1 using the settings and implementations provided by Carrell et al. (2025). Amongst the approximations, WILDCAT provides the highest mean Top-1 accuracy (82.19% vs. 82.55% for exact) while also yielding the lowest runtime for each layer, including an $11.59\times$ speed-up for the dominant layer 1.

### 4.3. Benchmarking KV cache compression

Finally, we evaluate the performance of COMPRESSKV on 13 benchmark KV cache compression tasks with the Qwen2.5-7B-Instruct language model (Qwen et al., 2024). In transformer-based autoregressive generative models, only the queries, keys, and values associated with the last decoded token of the sequence have to be computed from

*Table 3.* **Quality of attention approximations for T2T-ViT ImageNet classification.** We report speed-ups over 50 batches of 64 images and mean Top-1 accuracy $\pm 1$ standard deviation across five seeds.

| Attention Algorithm | Top-1 Accuracy (%) | Layer 1 Speed-up | Layer 2 Speed-up |
|---|---|---|---|
| **Exact** | $82.55 \pm 0.00$ | $1.00\times$ | $1.00\times$ |
| **Performer** | $80.91 \pm 0.18$ | $7.29\times$ | $1.82\times$ |
| **Reformer** | $81.47 \pm 0.06$ | $2.35\times$ | $0.92\times$ |
| **KDEformer** | $82.04 \pm 0.02$ | $3.28\times$ | $0.49\times$ |
| **ScatterBrain** | $82.05 \pm 0.02$ | $2.65\times$ | $0.77\times$ |
| **Thinformer** | $82.16 \pm 0.02$ | $8.84\times$ | $2.61\times$ |
| **WILDCAT** | $82.19 \pm 0.04$ | $11.59\times$ | $2.65\times$ |

*Table 4.* **Quality of KV cache compression for LongBench-E long-context language understanding.**

| Method | qasper | multifield | hotpot | 2wiki | gov | multinews | trec | trivia | samsum | p.count | p.ret | lcc | repo-p | average |
|---|---|---|---|---|---|---|---|---|---|---|---|---|---|---|
| | | | | | | **75.0% Compression** | | | | | | | | |
| **Exact** | 43.76 | 50.08 | 56.43 | 43.71 | 34.16 | 24.32 | 64.00 | 87.53 | 38.56 | 14.67 | 99.67 | 69.96 | 62.17 | 53.00 |
| **StreamingLLM** | 23.17 | 25.49 | 26.45 | 20.89 | 29.65 | 22.25 | 52.33 | 75.17 | 35.79 | 12.50 | 24.83 | 68.48 | 56.22 | 36.40 |
| **PyramidKV** | 21.59 | 29.96 | 39.38 | 30.24 | 27.12 | 21.13 | 43.33 | 86.77 | 38.27 | 15.88 | 61.06 | 67.40 | 59.20 | 41.64 |
| **BalanceKV** | 29.50 | 36.57 | 37.89 | 23.71 | 30.27 | 21.98 | 55.00 | 73.83 | 34.56 | 12.67 | 71.67 | 65.48 | 62.57 | 42.75 |
| **Uniform** | 26.91 | 37.51 | 38.46 | 25.93 | 30.02 | 21.86 | 54.33 | 81.71 | 35.37 | 15.33 | 63.44 | 64.84 | 61.34 | 42.85 |
| **SnapKV** | 25.52 | 30.13 | 44.36 | 31.80 | 29.70 | 22.10 | 49.33 | 88.32 | 37.15 | 16.67 | 89.06 | 69.16 | 56.33 | 45.36 |
| **CompressKV** | 33.23 | 38.13 | 43.43 | 33.37 | 30.30 | 22.26 | 54.33 | 86.15 | 35.38 | 14.33 | 98.00 | 64.85 | 60.43 | **47.25** |
| | | | | | | **87.5% Compression** | | | | | | | | |
| **Exact** | 43.76 | 50.08 | 56.43 | 43.71 | 34.16 | 24.32 | 64.00 | 87.53 | 38.56 | 14.67 | 99.67 | 69.96 | 62.17 | 53.00 |
| **StreamingLLM** | 19.27 | 24.62 | 24.34 | 21.55 | 26.40 | 20.49 | 47.33 | 71.71 | 33.61 | 10.67 | 15.67 | 67.24 | 59.13 | 34.00 |
| **PyramidKV** | 17.69 | 24.43 | 31.94 | 26.08 | 25.48 | 20.34 | 40.33 | 87.41 | 37.68 | 13.89 | 42.22 | 66.61 | 58.74 | 37.91 |
| **BalanceKV** | 17.90 | 28.87 | 27.79 | 17.94 | 27.45 | 20.78 | 45.67 | 62.84 | 33.29 | 10.67 | 32.22 | 60.95 | 60.90 | 34.41 |
| **Uniform** | 16.79 | 30.22 | 27.98 | 18.50 | 27.10 | 20.90 | 44.67 | 68.40 | 33.95 | 13.00 | 26.00 | 62.08 | 59.91 | 34.58 |
| **SnapKV** | 16.36 | 25.74 | 35.15 | 24.96 | 26.38 | 20.76 | 45.17 | 88.29 | 37.09 | 14.00 | 58.11 | 68.84 | 56.41 | 39.79 |
| **CompressKV** | 23.16 | 30.14 | 35.27 | 24.96 | 27.70 | 21.16 | 42.33 | 83.23 | 34.33 | 14.67 | 87.06 | 63.46 | 60.05 | **42.12** |
| | | | | | | **93.75% Compression** | | | | | | | | |
| **Exact** | 43.76 | 50.08 | 56.43 | 43.71 | 34.16 | 24.32 | 64.00 | 87.53 | 38.56 | 14.67 | 99.67 | 69.96 | 62.17 | 53.00 |
| **StreamingLLM** | 13.98 | 22.17 | 23.05 | 21.25 | 23.27 | 18.34 | 38.33 | 65.88 | 31.82 | 6.67 | 8.11 | 63.30 | 56.67 | 30.22 |
| **PyramidKV** | 11.46 | 22.21 | 30.97 | 23.14 | 22.94 | 18.86 | 32.00 | 85.48 | 36.66 | 11.67 | 18.78 | 64.06 | 56.46 | 33.44 |
| **BalanceKV** | 10.09 | 24.12 | 19.80 | 18.34 | 24.17 | 19.62 | 24.33 | 51.58 | 32.23 | 8.67 | 5.33 | 58.16 | 59.15 | 27.35 |
| **Uniform** | 11.24 | 24.06 | 21.17 | 16.70 | 24.22 | 19.38 | 31.83 | 54.58 | 32.97 | 11.67 | 7.00 | 55.81 | 58.48 | 28.39 |
| **SnapKV** | 11.47 | 22.52 | 30.79 | 23.13 | 23.18 | 18.89 | 31.33 | 86.09 | 36.63 | 11.67 | 18.78 | 64.51 | 55.67 | 33.44 |
| **CompressKV** | 15.03 | 24.71 | 28.99 | 24.01 | 24.63 | 19.51 | 26.00 | 80.27 | 34.13 | 13.00 | 62.33 | 62.28 | 60.26 | **36.55** |

hidden states, while $(\mathbf{k}_l, \mathbf{v}_l)$ for $l < n$ can be stored in a cache to avoid recomputation. However, as the context length $n$ increases, the KV cache memory eventually becomes a bottleneck, limiting the maximum number of past tokens that can be considered during inference. KV cache compressors conserve memory by extracting a smaller set of $r$ keys and values from the context and attending only over those $r$ context pairs during generation.

We begin with an empirical verificiation of the assumptions underlying our strongest compression guarantees. Specifically, we test the assumptions of Cor. 2 using Qwen2.5-7B-Instruct and document-grounded question answering

sequences from the QASPER-E dataset (Bai et al., 2024). Since $d$ is constant for any fixed model, Cor. 2 applies as long as $\beta R_{\mathbf{Q}} R_{\mathbf{K}} \in n^{o(1)}$. In Tab. 5, we find that $\gamma(n) = \frac{\beta R_{\mathbf{Q}} R_{\mathbf{K}}}{\log(n)}$, averaged all layers and the first 10 sequences with $n \geq 16384$, is not only bounded but is in fact decreasing with $n$. By Cor. 2, COMPRESSKV can therefore approximate attention with super-polynomially decaying error using a near-constant cache size $r$ for this model and task. Interestingly, this concordance with our assumptions is also implied by the work of Veličković et al. (2025, proof of Thm. 2.2), who showed that *any* fixed transformer-based model with a finite vocabulary has all query and key norms bounded independently of the sequence length $n$.

*Table 5.* For document-grounded question answering with Qwen2.5-7B-Instruct, the entry growth factor $\gamma(n) = \frac{\beta R_{\mathbf{Q}} R_{\mathbf{K}}}{\log(n)}$ of Cor. 2 decreases as a function of context length $n$.

| $n$ | 4 | 16 | 64 | 256 | 1024 | 4096 | 16384 |
|---|---|---|---|---|---|---|---|
| $\gamma(n)$ | 14.95 | 9.55 | 7.48 | 6.70 | 6.23 | 5.86 | 5.63 |

Following the experimental setup of Han et al. (2025), we next benchmark compression quality using the 13 LongBench-E tasks of Bai et al. (2024). These diverse tasks cover a wide range of long-context language understanding applications including single and multi-document question answering, summarisation, few-shot learning, and code completion. Following Han et al. (2025), we compress each cache by 75%, 87.75%, and 93.75% and benchmark COMPRESSKV with $B = \frac{r}{12}$ against no cache compression ("Exact") and five leading cache compression methods: StreamingLLM (Xiao et al., 2024), PyramidKV (Cai et al., 2025), BalanceKV and Uniform (Han et al., 2025), and SnapKV (Li et al., 2024b). We use the implementations of Han et al. (2025) for BalanceKV and Uniform and those provided by KVPress (Devoto et al., 2025) for the remaining methods. As in Han et al. (2025), BalanceKV, Uniform, and COMPRESSKV all retain the first and last 32 context tokens and compress the remaining tokens to achieve the desired compression level.

Tab. 4 reports a standard measure of compression quality for each LongBench-E task as well as the average compression quality across all 13 tasks. Remarkably, for each compression level, COMPRESSKV yields the highest average compression quality and the highest individual task quality on a plurality of the 13 tasks.

### 4.4. Benchmarking against FlashAttention 2

We additionally benchmark WILDCAT attention with $r = 64$ and $B = 16$ against the highly-optimized, I/O-aware FlashAttention 2 (FA2, Dao, 2024) implementation of exact attention using $(\mathbf{Q}, \mathbf{K}, \mathbf{V})$ inputs with $d = 64$, $n$ ranging from $2^{13}$ to $2^{18}$, and independent standard Gaussian entries. As the sequence length increases, we observe in Fig. 3 both a steady increase in speed-up over FA2 (from $1.1\times$ to $68\times$) and a steady decrease in approximation error $\|\mathbf{O} - \widehat{\mathbf{O}}\|_{\max}$. Additional ablations over the $r$ and $B$ parameters can be found in App. M.4.

### 5. Conclusions

We introduced WILDCAT, a principled method for cheaply, accurately, and practically approximating softmax attention. WILDCAT efficiently distills the information from all keys and values into a small coreset optimally weighted for attention reconstruction. Our fast but spectrally-accurate subsampling procedure and optimised weighting allow us

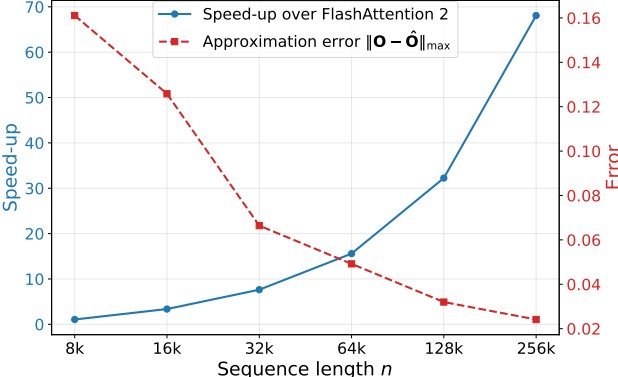

*Figure 3.* WILDCAT approximation error and speed-up over Flash Attention 2. See Sec. 4.4 for more details.

to achieve super-polynomially decaying error guarantees while maintaining near-linear runtime and near constant KV cache sizes. To bridge the gap between theory and practice, we additionally developed a GPU-optimised PyTorch implementation and demonstrated the practical benefits of WILDCAT for image generation, image classification, and KV cache compression.

That said, this work is not without its limitations. For example, this work does not address the important problem of streaming generation with causal masking, and we suspect such an extension is possible via prefix sums as in Choromanski et al. (2021) or divide-and-conquer evaluations as in Han et al. (2024). In future work, we also aim to address the path dependence and sequential nature of the pivot selection process in RPNYS. This could be achieved by oversampling mechanisms, such as those used in accelerated RPCholesky (Epperly et al., 2025) and recursive leverage-score sampling methods (Musco & Musco, 2017; Rudi et al., 2018). Fortunately, our modular analysis does allow for us to substitute any of these fast, spectrally accurate subsampling algorithms for RPNYS. However, these alternative procedures adaptively adjust coreset sizes across iterations, introducing additional challenges for batch-wise processing.

Finally, while our methodology naturally extends to other forms of kernelised attention (Tsai et al., 2019), more work may be required to extend our analysis. Such extensions are possible with sufficient knowledge of the spectral properties of the kernel data matrix. One set of tools for this purpose consists of sampling inequalities that express the low-rank approximability of a kernel matrix in terms of the fill distance of the underlying domain (Narcowich et al., 2005; Rieger & Zwicknagl, 2008; Fuselier & Wright, 2012; Altschuler et al., 2019). We suspect such tools will also yield improved runtime and error guarantees under additional smoothness or manifold (Zhu et al., 2018) assumptions on the attention inputs.

## Impact Statement

By improving the trade-off between resource consumption and model quality, WILDCAT and COMPRESSKV have the potential to reduce energy costs, to extend model access to resource-constrained settings, and to facilitate scientific discovery. However, we caution that any approximate attention tool should be deployed responsibly and only after evaluating the suitability and safety of the associated model.

## Acknowledgments

The authors thank Insu Han for sharing his code, model settings, and valuable advice concerning the image generation and image classification benchmarks. TS was supported by an EPSRC-DTP scholarship, partially funded by the Department of Mathematics at Imperial College London. TS thanks G-Research for financial support to attend the conference. Part of this research was conducted during TS's internship at Microsoft Research New England.

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

## Appendix Contents

## A. Proof of Lem. 1: Approximate attention guarantee

Introduce the shorthand $\mathbf{A}_{\min} \triangleq \min_{i \in [m], l \in [n]} \mathbf{A}_{il}$. Since $\mathbf{D}^{-1}\mathbf{A}$ is a row-stochastic matrix, and $\widehat{\mathbf{O}} = \mathrm{clip}(\widehat{\mathbf{D}}^{-1}\widehat{\mathbf{A}}\mathbf{V}, \mathbf{v}_{\min}, \mathbf{v}_{\max})$ we immediately have the upper bound

$$\|\mathbf{O} - \widehat{\mathbf{O}}\|_{\max} \leq \|\mathbf{v}_{\max} - \mathbf{v}_{\min}\|_{\max} \leq 2\|\mathbf{V}\|_{\max}.$$

We now consider two cases. First suppose that $\|\mathbf{A} - \widehat{\mathbf{A}}\|_{2,\infty} \geq \sqrt{n}\mathbf{A}_{\min}$. Then $\frac{\frac{3}{\sqrt{n}}\|\mathbf{A} - \widehat{\mathbf{A}}\|_{2,\infty}}{\mathbf{A}_{\min}} \geq 3 > 2$, so the advertised bound (2) holds.

Next suppose that $\|\mathbf{A} - \widehat{\mathbf{A}}\|_{2,\infty} < \sqrt{n}\mathbf{A}_{\min}$. In this case, $\widehat{\mathbf{D}}$ has all positive entries as, for each $i \in [m]$,

$$\widehat{\mathbf{D}}_{ii} = \mathbf{D}_{ii} + \widehat{\mathbf{D}}_{ii} - \mathbf{D}_{ii} = \mathbf{e}_i^\top \mathbf{A}\mathbf{1}_n + \mathbf{e}_i^\top(\widehat{\mathbf{A}} - \mathbf{A})\mathbf{1}_n \geq n\min_{l \in [n]}\mathbf{A}_{il} - \|\widehat{\mathbf{A}} - \mathbf{A}\|_{2,\infty}\|\mathbf{1}_n\|_2 > 0,$$

where we have used Hölder's inequality and the definition of $\|\cdot\|_{2,\infty}$.

Now consider the decomposition

$$\mathbf{O} - \widehat{\mathbf{O}} = (\mathbf{D}^{-1}\mathbf{A}\mathbf{V} - \mathbf{D}^{-1}\widehat{\mathbf{D}}\widehat{\mathbf{O}}) + (\mathbf{D}^{-1}\widehat{\mathbf{D}}\widehat{\mathbf{O}} - \widehat{\mathbf{O}}) = \mathbf{D}^{-1}(\mathbf{A}\mathbf{V} - \widehat{\mathbf{D}}\widehat{\mathbf{O}}) + (\mathbf{D}^{-1} - \widehat{\mathbf{D}}^{-1})\widehat{\mathbf{D}}\widehat{\mathbf{O}}. \qquad (7)$$

Using Hölder's inequality, the definition of $\|\cdot\|_{2,\infty}$, and the clipping of $\widehat{\mathbf{O}}$, we find that, for each $i \in [m]$ and $j \in [n]$,

$$|((\mathbf{D}^{-1} - \widehat{\mathbf{D}}^{-1})\widehat{\mathbf{D}}\widehat{\mathbf{O}})_{ij}| = |(\mathbf{D}^{-1}(\widehat{\mathbf{D}} - \mathbf{D})\widehat{\mathbf{O}})_{ij}| = \frac{|\mathbf{e}_i^\top(\widehat{\mathbf{A}} - \mathbf{A})\mathbf{1}_n|}{\mathbf{e}_i^\top \mathbf{A}\mathbf{1}_n}|\widehat{\mathbf{O}}_{ij}| \leq \frac{\frac{1}{\sqrt{n}}\|\mathbf{A} - \widehat{\mathbf{A}}\|_{2,\infty}\|\mathbf{V}\|_{\max}}{\mathbf{A}_{\min}}. \qquad (8)$$

Next, since $\widehat{\mathbf{D}}$ has positive entries, we can write

$$\widehat{\mathbf{D}}\widehat{\mathbf{O}} = \mathrm{clip}(\widehat{\mathbf{A}}\mathbf{V}, \widehat{\mathbf{D}}\mathbf{v}_{\min}, \widehat{\mathbf{D}}\mathbf{v}_{\max}), \quad \mathbf{AV} = \mathrm{clip}(\mathbf{AV}, \mathbf{D}\mathbf{v}_{\min}, \mathbf{D}\mathbf{v}_{\max}),$$

and therefore, by the triangle inequality, Hölder's inequality, and the definition of $\|\cdot\|_{2,\infty}$

$$
\begin{aligned}
\|\mathbf{AV} - \widehat{\mathbf{D}}\widehat{\mathbf{O}}\|_{\max} &\leq \|\mathrm{clip}(\mathbf{AV}, \mathbf{D}\mathbf{v}_{\min}, \mathbf{D}\mathbf{v}_{\max}) - \mathrm{clip}(\mathbf{AV}, \widehat{\mathbf{D}}\mathbf{v}_{\min}, \widehat{\mathbf{D}}\mathbf{v}_{\max})\|_{\max} \\
&\quad + \|\mathrm{clip}(\mathbf{AV}, \widehat{\mathbf{D}}\mathbf{v}_{\min}, \widehat{\mathbf{D}}\mathbf{v}_{\max}) - \mathrm{clip}(\widehat{\mathbf{A}}\mathbf{V}, \widehat{\mathbf{D}}\mathbf{v}_{\min}, \widehat{\mathbf{D}}\mathbf{v}_{\max})\|_{\max} \\
&\leq \|\mathbf{D} - \widehat{\mathbf{D}}\|_{\max}\|\mathbf{V}\|_{\max} + \max_{i\in[m], j\in[n]} |\mathbf{e}_i^\top (\mathbf{A} - \widehat{\mathbf{A}})\mathbf{V}\mathbf{e}_j| \\
&\leq \sqrt{n}\|\mathbf{A} - \widehat{\mathbf{A}}\|_{2,\infty}\|\mathbf{V}\|_{\max} + \max_{j\in[n]} \|\mathbf{A} - \widehat{\mathbf{A}}\|_{2,\infty}\|\mathbf{V}\mathbf{e}_j\|_2 \leq 2\sqrt{n}\|\mathbf{A} - \widehat{\mathbf{A}}\|_{2,\infty}\|\mathbf{V}\|_{\max}.
\end{aligned}
$$

(9)

Finally, since $\|\mathbf{D}^{-1}\|_{\max} \leq 1/(n\mathbf{A}_{\min})$, we conclude that $\|\mathbf{O} - \widehat{\mathbf{O}}\|_{\max} \leq \frac{\frac{3}{\sqrt{n}}\|\mathbf{A}-\widehat{\mathbf{A}}\|_{2,\infty}\|\mathbf{V}\|_{\max}}{\mathbf{A}_{\min}}$ from (7), (8) and (9).

## B. Background on Reproducing Kernel Hilbert Spaces

A function $h : \mathbb{R}^d \times \mathbb{R}^d \to \mathbb{R}$ is called a kernel if there exists a Hilbert space $(\mathcal{H}, \langle\cdot,\cdot\rangle_{\mathcal{H}})$ and a (feature) map $\Phi : \mathbb{R}^d \to \mathcal{H}$ such that $h(\mathbf{x}, \mathbf{y}) = \langle\Phi(\mathbf{x}), \Phi(\mathbf{y})\rangle_{\mathcal{H}}$ (Christmann & Steinwart, 2008, Definition 4.1). The function $h$ is a kernel function if and only if for every dataset $\mathcal{D} \subseteq \mathbb{R}^d$ the data-matrix $\mathbf{H} \triangleq h(\mathcal{D}, \mathcal{D})$ is symmetric and positive semi-definite (Christmann & Steinwart, 2008, Theorem 4.16). In particular, the exponential kernel $h(\mathbf{x}, \mathbf{y}) = \exp(\beta\langle\mathbf{x}, \mathbf{y}\rangle)$ for $\beta > 0$ is a kernel function.

We will mostly deal with finite-dimensional subspaces of $\mathcal{H}$. Specifically, for a subset $\mathcal{C} \subseteq \mathbb{R}^d$, the span of feature maps evaluated at the $\mathcal{C}$ defines a finite-dimensional sub-space of $\mathcal{H}$:

$$\mathcal{H}_{\mathcal{C}} \triangleq \overline{\{\Phi(\mathbf{x}_l) \mid \mathbf{x}_l \in \mathcal{C}\}} \subseteq \mathcal{H}$$

Hence, we can define a projection operator

$$P_{\mathcal{C}} : \mathcal{H} \to \mathcal{H}_{\mathcal{C}}, \quad \Phi(\mathbf{y}) \mapsto \Phi(\mathcal{C})\langle\Phi(\mathcal{C}), \Phi(\mathcal{C})\rangle_{\mathcal{H}}^+ \langle\Phi(\mathcal{C}), \Phi(\mathbf{y})\rangle_{\mathcal{H}},$$

where $\langle\Phi(\mathcal{C}), \Phi(\mathbf{y})\rangle_{\mathcal{H}} = h(\mathcal{C}, \mathbf{y})$, $\mathbf{H}^+$ denotes the pseudo-inverse of $\mathbf{H}$ and $\Phi(\mathcal{C}) = \sum_{\mathbf{x}_l \in \mathcal{C}} \Phi(\mathbf{x}_l)$. We identify $\langle\Phi(\mathcal{C}), \Phi(\mathcal{C})\rangle_{\mathcal{H}} = h(\mathcal{C}, \mathcal{C})$. Consequently, the inner product on this sub-space reads

$$\langle P_{\mathcal{C}}\Phi(\mathbf{x}), P_{\mathcal{C}}\Phi(\mathbf{y})\rangle_{\mathcal{H}_{\mathcal{C}}} = h(\mathbf{x}, \mathcal{C})h(\mathcal{C}, \mathcal{C})^+ h(\mathcal{C}, \mathbf{y}) \triangleq h_{\mathrm{nys}}(\mathbf{x}, \mathbf{y}).$$

This is the so-called *Nyström projection* of the kernel $h$ which we will heavily employ in the following. The immediate consequence is that $h_{\mathrm{res}} \triangleq h - h_{\mathrm{nys}}$ is also a kernel function associated with the orthogonal complement of $\mathcal{H}_{\mathcal{C}}$ in $\mathcal{H}$. In particular, the Cauchy-Schwarz inequality holds in both sub-spaces:

$$h_{\mathrm{nys}}(\mathbf{x}, \mathbf{y}) \leq \sqrt{h_{\mathrm{nys}}(\mathbf{x}, \mathbf{x})}\sqrt{h_{\mathrm{nys}}(\mathbf{y}, \mathbf{y})}, \quad h_{\mathrm{res}}(\mathbf{x}, \mathbf{y}) \leq \sqrt{h_{\mathrm{res}}(\mathbf{x}, \mathbf{x})}\sqrt{h_{\mathrm{res}}(\mathbf{y}, \mathbf{y})}.$$

## C. Proof of Lem. 2: Nyström guarantee

Let $[\mathbf{Q}; \mathbf{K}]$ denote the row-wise concatenation of $\mathbf{Q}$ and $\mathbf{K}$. Then, using the variational formulation of $\|\cdot\|_{2,\infty}$, the positive-definiteness of the residual kernel matrix $h_{\mathrm{res}}([\mathbf{Q}; \mathbf{K}], [\mathbf{Q}; \mathbf{K}])$, and Cauchy-Schwarz, we find that

$$
\begin{aligned}
\|\mathbf{A} - \widehat{\mathbf{A}}\|_{2,\infty} = \|h_{\mathrm{res}}(\mathbf{Q}, \mathbf{K})\|_{2,\infty} &= \max_{i\in[m]} \sup_{\mathbf{u}:\|\mathbf{u}\|_2=1} \mathbf{e}_i^\top h_{\mathrm{res}}(\mathbf{Q}, \mathbf{K})\mathbf{u} \\
&= \max_{i\in[m]} \sup_{\mathbf{u}:\|\mathbf{u}\|_2=1} [\mathbf{e}_i; \mathbf{0}_n]^\top h_{\mathrm{res}}([\mathbf{Q}; \mathbf{K}], [\mathbf{Q}; \mathbf{K}])[\mathbf{0}_m; \mathbf{u}] \\
&\leq \max_{i\in[m]} \sup_{\mathbf{u}:\|\mathbf{u}\|_2=1} \sqrt{[\mathbf{e}_i; \mathbf{0}_n]^\top h_{\mathrm{res}}([\mathbf{Q}; \mathbf{K}], [\mathbf{Q}; \mathbf{K}])[\mathbf{e}_i; \mathbf{0}_n]}\sqrt{[\mathbf{0}_m; \mathbf{u}]^\top h_{\mathrm{res}}([\mathbf{Q}; \mathbf{K}], [\mathbf{Q}; \mathbf{K}])[\mathbf{0}_m; \mathbf{u}]} \\
&= \max_{i\in[m]} \sup_{\mathbf{u}:\|\mathbf{u}\|_2=1} \sqrt{\mathbf{e}_i^\top h_{\mathrm{res}}(\mathbf{Q}, \mathbf{Q})\mathbf{e}_i}\sqrt{\mathbf{u}^\top h_{\mathrm{res}}(\mathbf{K}, \mathbf{K})\mathbf{u}} = \sqrt{\exp(\beta\|\mathbf{Q}\|_{2,\infty}^2) \cdot \|h_{\mathrm{res}}(\mathbf{K}, \mathbf{K})\|_{\mathrm{op}}}.
\end{aligned}
$$

## D. Proof of Thm. 1: RPNYS guarantee

The $\mathbf{H}$ estimate $\widehat{\mathbf{H}}^r \triangleq h(\mathbf{K}, \mathbf{K}_{\mathcal{S}})\mathbf{W}$ produced by RPNYS is identical to that produced by the randomly pivoted Cholesky (RPC) algorithm studied in Epperly & Moreno (2023). Hence, Thm. 7 of Epperly & Moreno (2023) already states a slightly looser upper bound on $\mathbb{E}\|\mathbf{H} - \widehat{\mathbf{H}}^r\|_{\mathrm{op}}$. We revisit the arguments of Epperly & Moreno (2023, Thm. 7) to derive the refined bound in Thm. 1.

We begin by computing the expected value of the residual kernel matrix after one iteration of RPNYS. Suppose $s$ is sampled according to the $\mathbf{p}^r$ pivoting distribution (3), and let $\mathcal{S}' = \mathcal{S} \cup \{s\}$. Then improvement of $\widehat{\mathbf{H}}^{(r+1)}$ over $\widehat{\mathbf{H}}^r$ is given by

$$
\begin{aligned}
\widehat{\mathbf{H}}^{(r+1)} - \widehat{\mathbf{H}}^r &= h(\mathbf{K}, \mathbf{K}_{\mathcal{S}'})\mathbf{g}\,\mathbf{g}^\top h(\mathbf{K}_{\mathcal{S}'}, \mathbf{K}) \\
&= \frac{(h_{\mathrm{nys}}^r(\mathbf{K}, \mathbf{k}_s) - h(\mathbf{K}, \mathbf{k}_s))(h_{\mathrm{nys}}^r(\mathbf{K}, \mathbf{k}_s) - h(\mathbf{K}, \mathbf{k}_s))^\top}{h_{\mathrm{res}}^r(\mathbf{k}_s, \mathbf{k}_s)} = \frac{h_{\mathrm{res}}^r(\mathbf{K}, \mathbf{k}_s)h_{\mathrm{res}}^r(\mathbf{k}_s, \mathbf{K})^\top}{h_{\mathrm{res}}^r(\mathbf{k}_s, \mathbf{k}_s)}\,.
\end{aligned}
$$

Defining the residual kernel matrices $\mathbf{H}_{\mathrm{res}}^q \triangleq \mathbf{H} - \widehat{\mathbf{H}}^q$ for $q \in \{r, r+1\}$, we therefore have

$$
\mathbb{E}\left[\mathbf{H}_{\mathrm{res}}^{(r+1)}\right] = \mathbb{E}\left[\mathbf{H} - \widehat{\mathbf{H}}^r + \widehat{\mathbf{H}}^r - \widehat{\mathbf{H}}^{(r+1)}\right] = \mathbb{E}\left[\mathbf{H}_{\mathrm{res}}^r - \frac{h_{\mathrm{res}}^r(\mathbf{K}, \mathbf{x}_s)h_{\mathrm{res}}^r(\mathbf{x}_s, \mathbf{K})}{h_{\mathrm{res}}^r(\mathbf{x}_s, \mathbf{x}_s)}\right] = \mathbf{H}_{\mathrm{res}}^r - \frac{{\mathbf{H}_{\mathrm{res}}^r}^2}{\mathrm{tr}(\mathbf{H}_{\mathrm{res}}^r)}\,.
$$

Using this identity one obtains the following lemma:

**Lemma D.1** (Iterated expected residual bound). *Consider the map $\Phi(\mathbf{A}) = \mathbf{A} - \frac{\mathbf{A}^2}{\mathrm{tr}(\mathbf{A})}$ defined for symmetric positive definite matrices $\mathbf{A}$. It holds that*

$$
\mathbb{E}\|\mathbf{H} - \widehat{\mathbf{H}}^r\|_{\mathrm{op}} \le \lambda_1(\Phi^r(\mathbf{H}))\,.
$$

*Proof.* Using the tower-property of conditional expectations we have

$$
\mathbb{E}\left[\mathbf{H} - \widehat{\mathbf{H}}^r\right] = \mathbb{E}\left[\mathbb{E}\left[\mathbf{H} - \widehat{\mathbf{H}}^r \mid \mathbf{H}^{(r-1)}\right]\right] = \mathbb{E}\left[\Phi(\mathbf{H} - \widehat{\mathbf{H}}^{(r-1)})\right] \preceq \Phi\left(\mathbb{E}\left[\mathbf{H} - \widehat{\mathbf{H}}^{(r-1)}\right]\right)
$$

where in the final step we used Jensen's inequality and the concavity of $\Phi$ (Chen et al., 2022a, Lem. 5.3). By the same lemma, $\Phi$ is monotone in the sense that $\mathbf{A} \preceq \mathbf{B} \implies \Phi(\mathbf{A}) \preceq \Phi(\mathbf{B})$, and we can thus iterate the argument and conclude $\mathbb{E}\left[\mathbf{H} - \widehat{\mathbf{H}}^r\right] \preceq \Phi^r(\mathbf{H})$. We now have for any $\mathbf{k} \in \mathbb{R}^n$ with $\|\mathbf{k}\|_2 \le 1$

$$
\mathbb{E}\left[\mathbf{k}^\top(\mathbf{H} - \widehat{\mathbf{H}}^r)\mathbf{k}\right] = \mathbf{k}^\top \mathbb{E}\left[\mathbf{H} - \widehat{\mathbf{H}}^r\right]\mathbf{k} \le \mathbf{k}^\top \Phi^r(\mathbf{H})\mathbf{k} \le \lambda_1(\Phi^r(\mathbf{H}))\,.
$$

$\square$

Our proof will also make use of a second lemma that bounds the maximum eigenvalue of $\Phi^r(\mathbf{H})$ in terms of an ordinary differential equation.

**Lemma D.2** (Differential equation bound). *Let $q \in [n]$ be arbitrary. Then, $\lambda_1(\Phi^r(\mathbf{H})) \le \eta(r)$, where $\eta$ is the decreasing solution of the ordinary differential equation*

$$
\frac{\mathrm{d}\eta(t)}{\mathrm{d}t} = -\frac{\eta(t)^2}{q\eta(t) + \sum_{l=q+1}^n \lambda_l(\mathbf{H})} \quad \text{with} \quad \eta(0) = \lambda_1(\mathbf{H})\,. \tag{10}
$$

*Proof.* We follow the proof of Epperly & Moreno (2023, Thm. 7). Firstly, the ordered eigenvalues of $\Phi^r(\mathbf{H})$ are non-negative and satisfy the following recurrence relation:

$$
\lambda_i(\Phi^r(\mathbf{H})) = \lambda_i(\Phi^{r-1}(\mathbf{H})) - \frac{\lambda_i(\Phi^{r-1}(\mathbf{H}))^2}{\sum_{l=1}^n \lambda_l(\Phi^{r-1}(\mathbf{H}))}\,. \tag{11}
$$

It follows that $\lambda_i(\Phi^r(\mathbf{H})) \le \lambda_i(\Phi^{r-1}(\mathbf{H}))$. In addition, one can show from this that $\lambda_{i+1}(\Phi^r(\mathbf{H})) \le \lambda_i(\Phi^r(\mathbf{H}))$, i.e., the recurrence relation preserves the ordering of the eigenvalues (Epperly & Moreno, 2023, proof of Thm. 7). Using these two facts, one can bound the trace of $\Phi^{r-1}(\mathbf{H})$ as

$$
\sum_{i=1}^n \lambda_i(\Phi^{r-1}(\mathbf{H})) \le q\lambda_1(\Phi^{r-1}(\mathbf{H})) + \sum_{i=q+1}^n \lambda_i(\mathbf{H})\,.
$$

Plugging this into (11) yields

$$\lambda_1(\Phi^r(\mathbf{H})) \le \lambda_1(\Phi^{r-1}(\mathbf{H})) - \frac{\lambda_1(\Phi^{r-1}(\mathbf{H}))^2}{q\lambda_1(\Phi^{r-1}(\mathbf{H})) + \sum_{i=q+1}^n \lambda_i(\mathbf{H})}\,.$$

Let now $\eta(t)$ be the solution to the ordinary differential equation in (10). Since $\mathrm{d}\eta(t)/\mathrm{d}t$ is negative, the solution $\eta(t)$ is monotonically decreasing. Since additionally, $x \mapsto -\frac{x^2}{qx+\sum_{i=q+1}^n \lambda_i(\mathbf{H})}$ is decreasing it follows that $\mathrm{d}\eta(t)/\mathrm{d}t$ is decreasing and consequently

$$\eta(r+1) = \eta(r) + \int_r^{r+1} \frac{\mathrm{d}\eta(t)}{\mathrm{d}t}\mathrm{d}t \le \eta(r) - \frac{\eta(r)^2}{q\eta(r) + \sum_{i=q+1}^n \lambda_i(\mathbf{H})}\,.$$

Finally, we notice that the function

$$x \mapsto \varphi(x) \triangleq x - \frac{x^2}{qx + \sum_{i=q+1}^n \lambda_i(\mathbf{H})}$$

is monotonically increasing in $x$ for any $q \ge 1$. Hence, we conclude inductively from $\lambda_1(\mathbf{H}) \le \eta(0)$, $\lambda_1(\Phi^{r+1}(\mathbf{H})) \le \varphi(\lambda_1(\Phi^r(\mathbf{H})))$, $\eta(r+1) \le \varphi(\eta(r))$, and the monotonicity of $\varphi$ that $\lambda_1(\Phi^r(\mathbf{H})) \le \eta(r)$ for all $r \in \mathbb{N}$. $\qquad\square$

Finally, we will use a variant of the Eckart-Young-Mirsky (Mirsky, 1960) theorem for the trace norm:

**Lemma D.3** (Eckart-Young-Mirsky for nuclear norm). *Let $\mathbf{A}$ be positive semi-definite and let $\lambda_1(\mathbf{A}) \ge \lambda_2(\mathbf{A}) \ge \cdots \ge \lambda_n(\mathbf{A}) \ge 0$ be the ordinally sorted eigenvalues of $\mathbf{A}$. Then,*

$$\sum_{i=q+1}^n \lambda_i(\mathbf{A}) = \min_{\substack{\mathrm{rank}(\Phi)\le q \\ \Phi\Phi^\top \preceq \mathbf{A}}} \mathrm{tr}(\mathbf{A} - \Phi\Phi^\top)$$

*Proof.* Let $\mathbf{P}_q(\mathbf{A})$ be the best rank $q$ approximation of $\mathbf{A}$. By the Eckart-Young-Mirsky theorem for the spectral norm it holds for $i = 1, 2, \ldots, n$ that

$$\lambda_1(\mathbf{A} - \mathbf{P}_q(\mathbf{A})) = \lambda_{q+1}(\mathbf{A})$$

and consequently for any $\Phi\Phi^\top \preceq \mathbf{A}$

$$\sum_{i=1}^n \lambda_i(\mathbf{A} - \Phi\Phi^\top) = \sum_{i=1}^n \lambda_1(\mathbf{A} - \Phi\Phi^\top - \mathbf{P}_{i-1}(\mathbf{A} - \Phi\Phi^\top))\,.$$

Next, we use that the matrix $\Phi\Phi^\top + \mathbf{P}_{i-1}(\mathbf{A} - \Phi\Phi^\top)$ is at most a matrix of rank $q + i - 1$. Thus, we get

$$\lambda_1(\mathbf{A} - \Phi\Phi^\top - \mathbf{P}_{i-1}(\mathbf{A} - \Phi\Phi^\top)) \ge \lambda_1(\mathbf{A} - \mathbf{P}_{q+i-1}(\mathbf{A})) = \lambda_{q+i}(\mathbf{A})$$

Consequently, it follows

$$\sum_{i=1}^n \lambda_i(\mathbf{A} - \Phi\Phi^\top) \ge \sum_{i=1}^n \lambda_{q+i}(\mathbf{A}) = \sum_{i=q+1}^n \lambda_i(\mathbf{A})\,.$$

$\qquad\square$

We now conclude with a proof of the theorem statement:

**Proof of Thm. 1: RPNYS guarantee**

*Proof.* From Lem. D.1 and Lem. D.2 we have after $r$ iterations of RPNYS

$$\mathbb{E}\|\mathbf{H} - \widehat{\mathbf{H}}^r\|_{\mathrm{op}} \le \lambda_1(\Phi^r(\mathbf{H})) \le \eta(r)\,.$$

Next, we find a time parameter $r_\varepsilon$ such that $\eta(r_\varepsilon) = \varepsilon$. By a separation of variables we have for $q \in [n]$

$$r_\varepsilon = \int_{\lambda_1(\mathbf{H})}^\varepsilon \left(-\frac{q\eta + \sum_{i=q+1}^n \lambda_i(\mathbf{H})}{\eta^2}\right)\mathrm{d}\eta = q\log\left(\frac{\lambda_1(\mathbf{H})}{\varepsilon}\right) + \left(\sum_{i=q+1}^n \lambda_i(\mathbf{H})\right)\left(\frac{1}{\varepsilon} - \frac{1}{\lambda_1(\mathbf{H})}\right)$$

From Lem. D.3 we have that $\sum_{i=q+1}^n \lambda_i(\mathbf{H}) \le \mathrm{tr}(\mathbf{H} - \mathbf{T})$ for any $\mathbf{T} \preceq \mathbf{H}$ with $\mathrm{rank}(\mathbf{T}) = q$. Since $\eta$ is decreasing in its argument, and $r \ge r_\varepsilon$, we have $\eta(r) \le \eta(r_\varepsilon) = \varepsilon$ yielding the claim. $\qquad\square$

# E. Proof of Lem. 3: Taylor guarantee

The guarantees of RPNYS depend on the low-rank approximability of $\mathbf{H}$ through $\mathrm{rank}(\mathbf{T})$ and $\mathrm{tr}(\mathbf{H} - \mathbf{T})$ for $\mathbf{T} \preceq \mathbf{H}$. We determine explicit expressions for these to quantities via a Taylor approximation of the exponential kernel. For a dataset $\mathbf{K} = (\mathbf{k}_l)_{l \in [n]}$, define the kernel matrix $\mathbf{H} \triangleq h(\mathbf{K}, \mathbf{K})$, where $h(\mathbf{k}_1, \mathbf{k}_2) \triangleq \exp(\beta \langle \mathbf{k}_1, \mathbf{k}_2 \rangle)$ is the exponential kernel. To construct a suitable low-rank approximation, we find the polynomial expansion of the exponential kernel following Cotter et al. (2011). We use the notation of multi-indices $\alpha \in \mathbb{N}_0^d$:

$$|\alpha| = \alpha_1 + \alpha_2 + \cdots + \alpha_d \quad \text{and} \quad \binom{s}{\alpha} = \frac{s!}{\alpha_1! \alpha_2! \ldots \alpha_d!}.$$

**Lemma E.1** (Exponential kernel feature expansion). *The exponential kernel has the following expansion into features of rank one:*

$$\exp(\beta \langle \mathbf{k}_1, \mathbf{k}_2 \rangle) = \sum_{s=0}^{\infty} \sum_{|\alpha|=s} \phi_\alpha(\mathbf{k}_1) \phi_\alpha(\mathbf{k}_2)$$

*with*

$$\phi_\alpha(\mathbf{k}) = \sqrt{\frac{1}{|\alpha|!} \binom{|\alpha|}{\alpha} \beta^{|\alpha|}} \mathbf{k}^\alpha$$

*Proof.* A Taylor expansion of the exponential shows

$$h(\mathbf{k}, \mathbf{k}') = \sum_{s=0}^{\infty} \frac{1}{s!} \left( \sum_{j=1}^{d} \beta \mathbf{k}_j \mathbf{k}'_j \right)^s = \sum_{s=0}^{\infty} \frac{1}{s!} \left( \sum_{|\alpha|=s} \binom{s}{\alpha} \beta^s \mathbf{k}^\alpha \mathbf{k}'^\alpha \right) = \sum_{s=0}^{\infty} \sum_{|\alpha|=s} \frac{1}{s!} \binom{s}{\alpha} \beta^s \mathbf{k}^\alpha \mathbf{k}'^\alpha.$$

$\square$

We then have the following low-rank approximation result for $\mathbf{H} = h(\mathbf{K}, \mathbf{K})$:

**Lemma E.2** (Error from Taylor polynomial truncation). *Let $\mathbf{T}^s$ be the order $s$ Taylor approximation of the exponential kernel with entries $\mathbf{T}_{il}^s \triangleq \sum_{|\alpha| \leq s} \phi_\alpha(\mathbf{k}_i) \phi_\alpha(\mathbf{k}_l)$. Then, $\mathbf{T}^s$ has rank $\leq \binom{s+d}{d}$ and satisfies $\mathbf{T}^s \preceq \mathbf{H}$. Furthermore, the Taylor residual satisfies*

$$\mathrm{tr}(\mathbf{H} - \mathbf{T}^s) \leq n \exp(\beta \|\mathbf{K}\|_{2,\infty}^2) \left( \frac{e \beta \|\mathbf{K}\|_{2,\infty}^2}{s+1} \right)^{s+1}.$$

*Proof.* By definition, $\mathbf{T}_{il}^s$ is the sum of at most $\#\{\alpha \in \mathbb{N}_0^d \mid |\alpha| \leq s\} = \binom{s+d}{d}$ rank one matrices. We bound the trace by the worst case approximation error of any of the entries of $\mathbf{H}$:

$$\mathrm{tr}(\mathbf{H} - \mathbf{T}^s) \leq n \max_{i,l \in [n]} |h(\mathbf{k}_i, \mathbf{k}_l) - \mathbf{T}_{il}^s|.$$

Without loss of generality let this maximum be attained for $\mathbf{k}_i, \mathbf{k}_l$ where $i, l \in [n]$. By Taylor's theorem, the error between the exponential kernel and it's rank $s$ approximation is bounded by

$$\left| \exp(\beta \langle \mathbf{k}_i, \mathbf{k}_l \rangle) - \sum_{|\alpha| \leq s} \phi_\alpha(\mathbf{k}_1) \phi_\alpha(\mathbf{k}_2) \right| = \sum_{i=s+1}^{\infty} \frac{1}{i!} (\beta \langle \mathbf{k}_i, \mathbf{k}_l \rangle)^i \leq \frac{\exp(\beta \langle \mathbf{k}_i, \mathbf{k}_l \rangle)}{(s+1)!} (\beta \|\mathbf{K}\|_{2,\infty}^2)^{s+1}$$

$$\leq \exp(\beta \langle \mathbf{k}_i, \mathbf{k}_l \rangle) \left( \frac{e \beta \|\mathbf{K}\|_{2,\infty}^2}{s+1} \right)^{s+1}$$

where we first invoked the bound of the Taylor residual of the exponential function on $\mathbb{R}$, and afterwards used the lower bound of the factorial $n! > (n/e)^n$. Hence, we obtain for the polynomial approximation up to order $s$:

$$|h(\mathbf{k}_i, \mathbf{k}_l) - \mathbf{T}_{il}^s| \leq \exp(\beta \langle \mathbf{k}_i, \mathbf{k}_l \rangle) \left( \frac{e \beta \|\mathbf{K}\|_{2,\infty}^2}{s+1} \right)^{s+1} \leq \exp(\beta \|\mathbf{K}\|_{2,\infty}^2) \left( \frac{e \beta \|\mathbf{K}\|_{2,\infty}^2}{s+1} \right)^{s+1}.$$

$\square$

We are now in the position to invert the bound on the trace-norm, which is the main result of this section:

**Proof of Lem. 3: Taylor guarantee**

*Proof.* First, assume $\tau = 1$ such that $\mathbf{H}_\tau = \mathbf{H}$. Define $z \triangleq \log(n \exp(\beta R_{\mathbf{K}}^2)/\varepsilon)$ and write

$$\tilde{s} \triangleq e\beta R_{\mathbf{K}}^2 \exp\left(W_0\left(\frac{z}{e\beta R_{\mathbf{K}}^2}\right)\right),$$

We use the identities $(e\beta R_{\mathbf{K}}^2/\tilde{s})^{\tilde{s}} = \exp(\tilde{s}\log(e\beta R_{\mathbf{K}}^2/\tilde{s}))$ and $z = \exp(W_0(z))W_0(z)$ to compute

$$\exp\left(\tilde{s}\log\left(\frac{e\beta R_{\mathbf{K}}^2}{\tilde{s}}\right)\right) = \exp\left(-e\beta R_{\mathbf{K}}^2 \exp\left(W_0\left(\frac{z}{e\beta R_{\mathbf{K}}^2}\right)\right)W_0\left(\frac{z}{e\beta R_{\mathbf{K}}^2}\right)\right) = \exp(-z) = \frac{\varepsilon}{n\exp(\beta R_{\mathbf{K}}^2)}.$$

Next, let $s \triangleq \lfloor \tilde{s} \rfloor$. Note that $t \mapsto (e\beta R_{\mathbf{K}}^2/t)^t$ is maximised at $t = \beta R_{\mathbf{K}}^2$ and decreasing for $t > \beta R_{\mathbf{K}}^2$. Furthermore, $\tilde{s} \geq \beta R_{\mathbf{K}}^2$ by definition. Hence,

$$n\exp(\beta R_{\mathbf{K}}^2)\left(\frac{e\beta R_{\mathbf{K}}^2}{s+1}\right)^{s+1} \leq n\exp(\beta R_{\mathbf{K}}^2)\left(\frac{e\beta R_{\mathbf{K}}^2}{\tilde{s}}\right)^{\tilde{s}} = n\exp(\beta R_{\mathbf{K}}^2)\exp(-z) = \varepsilon.$$

Thus, the claim follows from Lem. E.2 and a rescaling by $\tau$ $\mathbf{k} \to \mathbf{k}/\tau$, $\mathbf{H} \to \mathbf{H}_\tau$, $R_{\mathbf{K}} \to R_{\mathbf{K}}/\tau$. □

## F. Proof of Lem. F.1: Bounds for the binomial coefficient

Since $\mathrm{rank}(\mathbf{T}^s) \leq \binom{s+d}{d}$ by Lem. E.2, the result follows immediately from the following more precise bound on the binomial coefficient:

**Lemma F.1** (Bounds for the binomial coefficient). *For any $s, d \in \mathbb{N}$, it holds*

$$\binom{s+d}{d} \leq \frac{1}{\sqrt{2\pi}}\sqrt{\frac{1}{s}+\frac{1}{d}}\exp\left((s+d)\mathrm{Ent}\left(\frac{s}{s+d}\right)\right)$$

*with* $\mathrm{Ent}(p) = -p\log(p) - (1-p)\log(1-p)$.

*Proof.* We have by Robbin's version of Stirlings formula (Robbins, 1955):

$$\sqrt{2\pi n}\left(\frac{n}{e}\right)^n e^{\frac{1}{12n+1}} \leq n! \leq \sqrt{2\pi n}\left(\frac{n}{e}\right)^n e^{\frac{1}{12n}} \quad \text{for all} \quad n \in \mathbb{N}.$$

Thus, invoking these bounds shows in a direct calculation

$$\binom{s+d}{d} = \frac{(s+d)!}{s!\,d!} \leq \frac{\sqrt{2\pi}\sqrt{s+d}(s+d)^s(s+d)^d}{2\pi\sqrt{sd}s^s\,d^d} = \frac{1}{\sqrt{2\pi}}\sqrt{\frac{1}{d}+\frac{1}{s}}\left(1+\frac{s}{d}\right)^d\left(1+\frac{d}{s}\right)^s.$$

Finally, we identify that

$$\left(1+\frac{s}{d}\right)^d\left(1+\frac{d}{s}\right)^s = \exp\left((s+d)\mathrm{Ent}\left(\frac{s}{s+d}\right)\right).$$

□

## G. Choosing a Rescaling Parameter for Keys and Queries

In this section we describe a rescaling rule of the form $\mathbf{q} \mapsto \mathbf{q}\tau$, $\mathbf{k} \mapsto \mathbf{k}/\tau$, which we use in our empirical and theoretical results. The key idea is the following: Suppose we perform the rescaling prescribed above and define the approximate attention matrix as the Nyström approximation obtained from rescaled queries and keys:

$$\widehat{\mathbf{A}}_\tau \triangleq h(\tau\mathbf{q}, \tau^{-1}\mathbf{K}_{\mathcal{S}})h(\tau^{-1}\mathbf{K}_{\mathcal{S}}, \tau^{-1}\mathbf{K}_{\mathcal{S}})^{-1}h(\tau^{-1}\mathbf{K}_{\mathcal{S}}, \tau^{-1}\mathbf{K}). \tag{12}$$

By Lem. 2, we find for RPNYS applied to $\mathbf{H}_\tau \triangleq h(\tau^{-1}\mathbf{K}, \tau^{-1}\mathbf{K})$:

$$\|\mathbf{A} - \widehat{\mathbf{A}}_\tau\|_{2,\infty} \leq \exp\left(\frac{\tau^2\beta}{2}R_\mathbf{Q}^2\right)\sqrt{\|\mathbf{H}_\tau - \widehat{\mathbf{H}}_\tau\|_\mathrm{op}}\,.$$

Increasing $\tau$ makes the matrix $\mathbf{H}_\tau$ increasingly low-rank approximable. Note that in the extreme case, $\mathbf{H}_\tau \xrightarrow{\tau \to \infty} \mathbf{1}_n\mathbf{1}_n^\top$ becomes a rank one matrix. On the other hand, increasing $\tau$ comes at the cost of the error inflation factor $\exp\left(\frac{\tau^2\beta}{2}R_\mathbf{Q}^2\right)$. We therefore aspire to strike a balance between the low-rank approximability of $\mathbf{H}_\tau$ and the control of $\exp\left(\frac{\tau^2\beta}{2}R_\mathbf{Q}^2\right)$. We start the analysis by characterising the rowwise approximation error of $\mathbf{A}$ in terms of the ratio $R_\mathbf{Q}/R_\mathbf{K}$ and $\tau$. We combine Lem. 2, Thm. 1, and Lem. 3 to obtain the following rowwise approximation guarantee for $\mathbf{A}_l = \exp(\beta\mathbf{q}^\top\mathbf{k}_l)$:

**Lemma G.1** (Rescaling guarantee). *Let $\mathbf{K}_\mathcal{S} \subseteq \mathbf{K}$ and $\mathbf{W}$ be the coreset and Nyström weights outputted by* RPNYS *for the kernel function $h_\tau(\mathbf{k}_1, \mathbf{k}_2) = \exp\left(\frac{\beta}{\tau^2}\langle\mathbf{k}_1, \mathbf{k}_2\rangle\right)$. Define the associated rank-r Nyström approximation $\widehat{\mathbf{A}}_\tau^r \triangleq h(\mathbf{q}, \mathbf{K}_\mathcal{S})\mathbf{W}$ of $\mathbf{A}$. Then,*

$$\mathbb{E}\|\mathbf{A} - \widehat{\mathbf{A}}_\tau^r\|_{2,\infty} \leq \varepsilon \quad \textit{for any} \quad r \geq \binom{s(\varepsilon) + d}{d}\left(\log(\|\mathbf{H}\|_\mathrm{op}\exp(\beta\tau^2R_\mathbf{Q}^2)/\varepsilon^2) + 1\right) \tag{13}$$

*provided*

$$s(\varepsilon) \geq \left\lfloor e\beta R_\mathbf{Q}R_\mathbf{K}\frac{\frac{1}{e}(\rho^2 + b(\varepsilon)\rho + 1)}{\rho W_0\left(\frac{1}{e}(\rho^2 + b(\varepsilon)\rho + 1)\right)}\right\rfloor, \tag{14}$$

*where $b(\varepsilon) = \frac{\log(n/\varepsilon^2)}{\beta R_\mathbf{Q}R_\mathbf{K}}$ and $\rho = \tau^2 R_\mathbf{Q}/R_\mathbf{K}$.*

*Proof.* We first apply Lem. 2 to rescaled datasets $\tau\mathbf{Q}$ and $\tau^{-1}\mathbf{K}$. From Lem. 2 it immediately follows from a rearrangement of the bound that

$$\mathbb{E}\left\|h_\mathrm{res}(\tau^{-1}\mathbf{K}, \tau^{-1}\mathbf{K})\right\|_\mathrm{op} \leq \varepsilon^2\exp(-\beta\tau^2R_\mathbf{Q}^2) \quad \Rightarrow \quad \mathbb{E}\|\mathbf{A} - \widehat{\mathbf{A}}_\tau^r\|_{2,\infty} \leq \varepsilon.$$

Using $\varepsilon^2\exp(-\beta\tau^2R_\mathbf{Q}^2)$ in place of $\varepsilon$ in Thm. 1 and Lem. E.2 we can get the claimed requirement for the runtime parameter in (13) for a polynomial with large enough rank $q = \binom{s+d}{d}$ such that

$$\mathrm{tr}(\mathbf{H}_\tau - \mathbf{T}^s) \leq \exp(-\beta\tau^2R_\mathbf{Q}^2)\varepsilon^2\,.$$

Lem. 3 states that a sufficient condition for this trace-bound is

$$s \geq \left\lfloor e\beta\frac{R_\mathbf{K}^2}{\tau^2}\exp\left(W_0\left(\frac{\log(n\exp(\beta\tau^2R_\mathbf{Q}^2 + \beta\tau^{-2}R_\mathbf{K}^2)/\varepsilon^2)}{e\beta\tau^{-2}R_\mathbf{K}^2}\right)\right)\right\rfloor. \tag{15}$$

At this point we can express the argument of the product logarithm in terms of the relative scales between $R_\mathbf{Q}$ and $R_\mathbf{K}$

$$\tau^{-2}R_\mathbf{K}^2 = \tau^{-2}\frac{R_\mathbf{K}}{R_\mathbf{Q}}R_\mathbf{Q}R_\mathbf{K} = \frac{R_\mathbf{Q}R_\mathbf{K}}{\rho} \quad \text{and} \quad \tau^2R_\mathbf{Q}^2 = \tau^2\frac{R_\mathbf{Q}}{R_\mathbf{K}}R_\mathbf{Q}R_\mathbf{K} = \rho R_\mathbf{Q}R_\mathbf{K}$$

which yields

$$\frac{\log(n\exp(\beta\tau^2R_\mathbf{Q}^2 + \beta\tau^{-2}R_\mathbf{K}^2)/\varepsilon^2)}{e\beta\tau^{-2}R_\mathbf{K}^2} = \frac{\log(n/\varepsilon^2)\rho + \beta R_\mathbf{Q}R_\mathbf{K}\rho^2 + \beta R_\mathbf{Q}R_\mathbf{K}}{e\beta R_\mathbf{Q}R_\mathbf{K}}\,.$$

Invoking the identity $\exp(W_0(z)) = z/W_0(z)$ yields for any $\rho$ the equivalent expression (14) for (15) as claimed. $\square$

**Remark 1.** *The result in Lem. G.1 describes guarantees in terms of $\rho = \tau^2 R_\mathbf{Q}/R_\mathbf{K}$. The temperature $\tau$ is a free parameter in our algorithm that we can tune to obtain improved performance or guarantees. Let $R_\mathbf{Q}, R_\mathbf{K}$ be fixed, define $\rho_\mathrm{in} \triangleq R_\mathbf{Q}/R_\mathbf{K}$, and let $\rho_\mathrm{out} > 0$. Then we have Lem. G.1 with $\rho = \rho_\mathrm{out}$ instead of $\rho = \rho_\mathrm{in}$ by running RPNYS on $\exp(\beta\mathbf{KK}^\top/\tau^2)$ with $\tau^2 = \rho_\mathrm{out}/\rho_\mathrm{in}$. The value of $\rho$ is fixed in the theoretical analysis once we invoke the Cauchy-Schwartz inequality in Lem. 2 and isolate the data matrix $\mathbf{H}_\tau$. To summarise, the approximation $\widehat{\mathbf{A}}_\tau$ and the guarantees in Lem. G.1 have the following connection:*

$$\textit{find } \rho_\mathrm{out} \;\rightarrow\; \textit{define } \tau \triangleq \sqrt{\rho_\mathrm{out}\frac{R_\mathbf{K}}{R_\mathbf{Q}}} \;\rightarrow\; \textit{run } \mathrm{RPNYS} \textit{ with } h_\tau = \exp\left(\frac{\beta}{\tau^2}\langle\cdot,\cdot\rangle\right) \xrightarrow{(12)} \widehat{\mathbf{A}}_\tau^{(r)} \rightarrow \textit{Lem. G.1 with } \rho = \rho_\mathrm{out}\,.$$

Finding the right value for $\tau$ turns out to be empirically important: with no temperature adjustment and insufficient choice of $r$, the Hilbert space $\mathcal{H}_{\mathbf{K}}$ associated with the data-kernel features is poorly approximated by $\mathcal{H}_{\mathcal{S}}$. This leads to large outliers in the Nyström weights $h(\mathbf{K}_{\mathcal{S}}, \mathbf{K}_{\mathcal{S}})^{-1} h(\mathbf{K}_{\mathcal{S}}, \mathbf{K})$ and poor recovery of the target attention matrix. When $\rho$ is too large, on the other hand, the RPNYS selection algorithm converges to uniform sampling without replacement, a less accurate choice. We now describe tight estimates for $\rho$ that minimise the dominating rank parameter $\binom{s+d}{d}$ in Lem. G.1.

**Lemma G.2** (Optimal choice of $\rho$). *For each $b \geq 0$, define the optimisation objective*

$$l_b(\rho) \triangleq \frac{\frac{1}{e}(\rho^2 + b\rho + 1)}{\rho W_0\left(\frac{1}{e}(\rho^2 + b\rho + 1)\right)}$$

*over $\rho > \sqrt{2}$. The unique minimiser of $l_0$ is given by*

$$\rho_0 \triangleq \sqrt{1 + e^{W_0(2/e^2)+2}}. \tag{16}$$

*More generally, $l_b$ is uniquely minimised by a solution $\rho_b$ of the ordinary differential equation*

$$\frac{\mathrm{d}}{\mathrm{d}b}\rho_b = \frac{\rho_b^2}{(\rho_b^2 + 1)\log(\rho_b^2 - 1)} = \frac{\rho_b}{2\rho_b + b}\left(1 - \frac{2}{\rho_b^2 + 1}\right) > 0 \quad \text{with} \quad \rho_0 \quad \text{as in} \quad (16). \tag{17}$$

*Furthermore, the minimal value of $l_b$ satisfies*

$$l_b(\rho_b) = \frac{\rho_b^2 - 1}{e\rho_b} < \frac{\rho_b}{e}.$$

*Proof.* Our proof proceeds in five steps. First, we characterise the stationary points of $\log l_b$. Second, we prove that any stationary point is a unique global minimizer of $l_b$. Fourth, we derive the closed-form minimiser of $l_0$ (16). Fifth, we derive the differential equation (17) defining the minimisers of $l_b$. Finally, we compute and bound the minimal value of $l_b$.

**Characterising the stationary points** We begin by characterising the stationary points of $\log l_b$. Using the identity $\exp(W_0(x)) = x/W_0(x)$, we find that $\log l_b(\rho) = W_0\left(\frac{1}{e}q(b, \rho)\right) - \log(\rho)$ for $q(b, \rho) \triangleq \frac{1}{e}(\rho^2 + b\rho + 1)$. The derivatives of the component functions are, using (31),

$$\frac{\mathrm{d}}{\mathrm{d}q}W_0(q) = \frac{W_0(q)}{q(1 + W_0(q))} \quad \text{and} \quad \frac{\mathrm{d}}{\mathrm{d}\rho}q(b, \rho) = \frac{1}{e}(2\rho + b)$$

Using the identity $q(b, \rho) - \frac{1}{e}(2\rho^2 + b\rho) = -q(b, \rho) + \frac{1}{e}(b\rho + 2)$ we therefore obtain the equivalent stationary point conditions

$$\frac{\mathrm{d}}{\mathrm{d}\rho}\log l_b(\rho) = \frac{\mathrm{d}}{\mathrm{d}\rho}W_0(q(b, \rho)) - \frac{1}{\rho} = \frac{\frac{1}{e}(2\rho + b)W_0(q(b, \rho))}{q(b, \rho)(W_0(q(b, \rho)) + 1)} - \frac{1}{\rho} = 0$$

$$\Leftrightarrow \quad \frac{1}{e}(2\rho^2 + b\rho) = q(b, \rho) + \frac{q(b, \rho)}{W_0(q(b, \rho))}$$

$$\Leftrightarrow \quad g(b, \rho) \triangleq \frac{q(b, \rho)}{W_0(q(b, \rho))} - q(b, \rho) + \frac{1}{e}b\rho + \frac{2}{e} = 0 \tag{18}$$

$$\Leftrightarrow \quad W_0(q(b, \rho)) = \frac{e\,q(b, \rho)}{\rho^2 - 1}. \tag{19}$$

Furthermore, for $b \geq 0$ and $\rho > 0$ we have $q(b, \rho) > 0$, and repeating the above calculations keeping track of the sign we find that

$$\mathrm{sgn}\left(\frac{\mathrm{d}}{\mathrm{d}\rho}\log l_b(\rho)\right) = \mathrm{sgn}\left(-\frac{g(b, \rho)}{q(b, \rho)}\right) = \mathrm{sgn}\left(-\frac{1}{W_0(q(b, \rho))} + \frac{\rho^2 - 1}{eq(b, \rho)}\right).$$

**Stationarity implies optimality**  Now suppose that for some $b \geq 0$ there is a stationary point $\rho_b \in \{\rho > \sqrt{2} \mid g(b,\rho) = 0\}$. We will show that $\rho_b$ is the unique global minimiser of $l_b(\rho)$ on $(\sqrt{2}, \infty)$. For $b \geq 0$ and all $\rho > \sqrt{2}$, the map $\rho \mapsto -\frac{1}{W_0(q(b,\rho))} + \frac{\rho^2-1}{eq(b,\rho)}$ is increasing because of

$$\frac{\partial}{\partial \rho} q(b,\rho) = \frac{1}{e}(2\rho + b) > 0,$$

$$\frac{\partial}{\partial \rho}\left(\frac{\rho^2-1}{eq(b,\rho)}\right) = \frac{2\rho(\rho^2 + b\rho + 1) - (\rho^2-1)(2\rho + b)}{e^2 q(b,\rho)^2} = \frac{b\rho^2 + 4\rho + b}{e^2 q(b,\rho)^2} > 0,$$

and the monotonicity of the Lambert-W function. Therefore $l(\rho)$ is increasing/decreasing for $\rho \gtrless \rho_b$ since

$$\text{sgn}\left(\frac{\mathrm{d}}{\mathrm{d}\rho} \log l_b(\rho)\right) = \text{sgn}\left(-\frac{1}{W_0(q(b,\rho))} + \frac{\rho^2-1}{eq(b,\rho)}\right) \gtrless \text{sgn}\left(-\frac{1}{W_0(q(b,\rho_b))} + \frac{\rho_b^2-1}{eq(b,\rho_b)}\right) = 0.$$

Hence, $\rho_b$ must be the only stationary point and the unique global minimiser.

**Closed-form minimiser of $l_0$**  Next, we identify the closed-form minimiser of $l_0$. For $b = 0$, the condition $\frac{q}{W_0(q)} - q + \frac{2}{e} = 0$ is satisfied at

$$q_0 = \frac{2}{e} + e^{W_0\left(\frac{2}{e^2}\right)+1},$$

and the quadratic equation $q(0, \rho_0) = q_0$ is satisfied by $\rho_0 \triangleq \sqrt{eq_0 - 1} \approx 3.19$.

**Differential equation for $\rho_b$**  The optimality condition (19) additionally implies that

$$q(b,\rho) = \exp(W_0(q(b,\rho)))W_0(q(b,\rho)) = \frac{eq(b,\rho)}{\rho^2-1}\exp\left(\frac{eq(b,\rho)}{\rho^2-1}\right),$$

$$(\rho^2-1)\log((\rho^2-1)/e) = eq(b,\rho) = \rho^2 + b\rho + 1, \quad \text{and}$$

$$b = \frac{(\rho^2-1)\log((\rho^2-1)/e) - (\rho^2+1)}{\rho} = \frac{(\rho^2-1)\log(\rho^2-1)}{\rho} - 2\rho.$$

Since $\rho \mapsto b_\rho \triangleq \frac{(\rho^2-1)\log(\rho^2-1)}{\rho} - 2\rho$ is continuously differentiable with range $(-2\sqrt{2}, \infty)$ and $\frac{\mathrm{d}}{\mathrm{d}\rho}b_\rho = \frac{(\rho^2+1)\log(\rho^2-1)}{\rho^2} > 0$ for $\rho > \sqrt{2}$, the inverse function theorem (Price, 1984, Thm. 31.1) implies that there exists a continuously differentiable inverse function $b \mapsto \rho_b$ on $b > -2\sqrt{2}$ with

$$\frac{\mathrm{d}}{\mathrm{d}b}\rho_b = \frac{1}{b'(\rho_b)} = \frac{\rho_b^2}{(\rho_b^2+1)\log(\rho_b^2-1)} = \frac{\rho_b(\rho_b^2-1)}{(\rho_b^2+1)(b+2\rho_b)} = \frac{\rho_b}{2\rho_b+b}\left(1 - \frac{2}{\rho_b^2+1}\right).$$

Furthermore, $\rho_0 > 0$, $\frac{\mathrm{d}}{\mathrm{d}b}\rho_b|_{b=0} > 0$, and $\frac{\mathrm{d}}{\mathrm{d}b}\rho_b$ is increasing in $\rho_b$ since

$$\frac{\mathrm{d}}{\mathrm{d}\rho}\left(\frac{\rho}{2\rho+b}\left(1 - \frac{2}{\rho^2+1}\right)\right) = \frac{b(\rho^4 + 4\rho^2 - 1) + 8\rho^3}{(\rho^2+1)^2(b+2\rho)^2} > 0.$$

Consequently, $\frac{\mathrm{d}}{\mathrm{d}b}\rho_b > 0$ for all $b \geq 0$. This establishes the claim (17). Finally, at the point of optimality we have from the first order condition (18)

$$l_b(\rho_b) = \frac{1}{\rho_b}\frac{q(b,\rho_b)}{W_0(q(b,\rho_b))} = \frac{1}{\rho_b}\left(q(b,\rho_b) - \frac{1}{e}b\rho_b - \frac{2}{e}\right) = \frac{\rho_b^2-1}{e\rho_b}.$$

$\square$

A tight upper and lower bound of this differential equation can be solved in closed form:

**Corollary G.1** (Bounds on optimal $\rho$). *For $b > 0$, a solution $\rho_b$ of the ordinary differential equation (17) obeys the bounds*

$$\max\left(\rho_0, \frac{4}{5}\frac{b}{2W_0\left(\frac{b}{2\rho_0}\right)}\right) \leq \rho_b \leq \frac{b}{2W_0\left(\frac{b}{2\rho_0}\right)}.$$

*Proof.* Since $b \mapsto \frac{\mathrm{d}}{\mathrm{d}b}\rho_b > 0$ by Lem. G.2 we have $\rho_b > \rho_0$ and therefore $2/(\rho_b^2 + 1) \le 2/(\rho_0^2 + 1)$ for $b > 0$. One can check numerically that $2/(\rho_0^2 + 1) \le 1/5$. Hence,

$$\frac{4}{5}\frac{\rho_b}{2\rho_b + b} \le \frac{\mathrm{d}}{\mathrm{d}b}\rho_b \le \frac{\rho_b}{2\rho_b + b}\,.$$

The simplified differential equation $\frac{\mathrm{d}}{\mathrm{d}b}\tilde{\rho}_b = \frac{\tilde{\rho}_b}{2\tilde{\rho}_b + b}$ with $\tilde{\rho}_0 = \rho_0$ now has the closed-form solution $\tilde{\rho}_b = b/(2W_0(b/(2\rho_0)))$. $\square$

## H. Proof of Thm. 2: WILDCAT guarantee

Cauchy-Schwarz and the definitions of $R_{\mathbf{Q}} = \|\mathbf{Q}\|_{2,\infty}$ and $R_{\mathbf{K}} = \|\mathbf{K}\|_{2,\infty}$ imply that

$$\min_{i\in[m],l\in[n]} \mathbf{A}_{il} = \exp\left(\beta \min_{i\in[m],l\in[n]} \langle \mathbf{q}_i, \mathbf{k}_l \rangle\right) \ge \exp\left(-\beta \max_{i\in[m],l\in[n]} \|\mathbf{q}_i\|_2 \|\mathbf{k}_l\|_2\right) \ge \exp(-\beta R_{\mathbf{Q}} R_{\mathbf{K}}).$$

Hence, to conclude that $\mathbb{E}\|\mathbf{O} - \widehat{\mathbf{O}}_r\|_{\max} \le 3\|\mathbf{V}\|_{\max}\varepsilon$ for $\varepsilon \triangleq n^{-a}$, it suffices to show that

$$\mathbb{E}\|\mathbf{A} - \widehat{\mathbf{A}}_\tau\|_{2,\infty} \le \varepsilon\sqrt{n}\exp(-\beta R_{\mathbf{Q}} R_{\mathbf{K}}) \tag{20}$$

by Lem. 1. Moreover, by Lem. G.1, the rowwise bound (20) holds whenever the RPNYS rank parameter $r$ satisfies

$$r \ge \binom{s+d}{d}\left(\log(\|\mathbf{H}\|_{\mathrm{op}}\exp(\beta\tau^2 R_{\mathbf{Q}}^2 + 2\beta R_{\mathbf{Q}} R_{\mathbf{K}}))/(\varepsilon\sqrt{n})^2\right) + 1 \quad \text{for some} \tag{21}$$

$$s \ge \lfloor e\beta R_{\mathbf{Q}} R_{\mathbf{K}} l_b(\rho) \rfloor \quad \text{with} \quad l_b(\rho) \triangleq \frac{\frac{1}{e}(\rho^2 + b\rho + 1)}{\rho W_0\big(\frac{1}{e}(\rho^2 + b\rho + 1)\big)}, \quad b \triangleq \frac{\log\big(\frac{1}{\varepsilon^2}\big)}{\beta R_{\mathbf{Q}} R_{\mathbf{K}}} + 2, \quad \text{and} \quad \rho \triangleq \tau^2 \frac{R_{\mathbf{Q}}}{R_{\mathbf{K}}}\,. \tag{22}$$

Thus, we will prove that our assumed constraint on $r$ (6) implies the rank bound (21).

We begin by identifying a relevant Taylor approximation order $s$ that satisfies the constraint (22). For each $b' \ge 0$, let $\rho_{b'}$ be the optimiser of $l_{b'}$ in Lem. G.2, and recall the definitions (4)

$$\tau \triangleq \sqrt{\frac{R_{\mathbf{K}}}{R_{\mathbf{Q}}}\frac{b_0}{2W_0(b_0/(2\rho_0))}} \quad \text{with} \quad b_0 \triangleq \frac{\log(n)}{\beta R_{\mathbf{Q}} R_{\mathbf{K}}} + 2$$

which imply $\rho = \frac{b_0}{2W_0(b_0/(2\rho_0))}$. By Cor. G.1 we have $\rho_0 \le \rho_{b_0} \le \rho$. By Cor. G.1 and Lem. L.1, we also have

$$\rho_{b'} \ge \frac{4}{5}\frac{b'}{2W_0(b'/(2\rho_0))} = \frac{4}{5}\rho_0 \exp(W_0(b'/(2\rho_0))) \quad \text{for all} \quad b' \ge 0.$$

Since $b' \mapsto \rho_{b'}$ is continuous for $b' \ge 0$ by Lem. G.2 and $b' \mapsto \rho_0 \exp(W_0(b'/(2\rho_0)))$ is coercive as $b' \to \infty$ by Lem. L.4, there exists a $b_\tau \ge b_0$ such that $\rho_{b_\tau} = \rho$. Fix any such $b_\tau$.

We now use the concavity of $x \mapsto x/W_0(x)$ and the derivative $\frac{\mathrm{d}}{\mathrm{d}x}(x/W_0(x)) = (W_0(x) + 1)^{-1}$ (see Lem. L.3) to upper bound $l_b(\rho)$ by a linear approximation in $b$ with expansion point $b_\tau$:

$$l_b(\rho) \le l_{b_\tau}(\rho) + \frac{\frac{1}{e}(b - b_\tau)\rho}{\rho\big(W_0\big(\frac{1}{e}(\rho^2 + b_\tau\rho + 1)\big) + 1\big)} \le \frac{1}{e}\rho + \frac{\frac{1}{e}(b - b_\tau)}{W_0\big(\frac{1}{e}(\rho^2 + b_\tau\rho + 1)\big) + 1}$$

$$= \frac{1}{e}\rho + \frac{1}{e}(b - b_\tau)\frac{\rho^2 - 1}{2\rho^2 + b_\tau\rho}\,.$$

Above, the second inequality follows from Lem. G.2 since $\rho_{b_\tau} = \rho$, and the equality follows from the first order condition (19) characterising $\rho_{b_\tau}$.

To further upper bound $l_b(\rho)$, we will consider two cases. We first recall that $\rho \ge \rho_0 > 1$ and $b_0 \le b_\tau$ and note that $b_0 \le b$ since $\varepsilon \le n^{-\frac{1}{2}}$. Hence, if $b < b_\tau$, then

$$\frac{1}{e}\rho + \frac{1}{e}(b - b_\tau)\frac{\rho^2 - 1}{2\rho^2 + b_\tau\rho} \le \frac{1}{e}\rho = \frac{1}{e}\frac{b_0}{2W_0\big(\frac{b_0}{2\rho_0}\big)} \le \frac{1}{e}\frac{b}{2W_0\big(\frac{b_0}{2\rho_0}\big)}\,.$$

Alternatively, if $b \geq b_\tau$, we have

$$
\frac{1}{e}\rho + \frac{1}{e}(b - b_\tau)\frac{\rho^2 - 1}{2\rho^2 + b_\tau\rho} = \frac{1}{e}\left(\frac{b_0}{2W_0\left(\frac{b_0}{2\rho_0}\right)} + (b - b_\tau)\frac{\rho^2 - 1}{2\rho^2 + b_\tau\rho}\right)
$$

$$
\leq \frac{1}{e}\left(\frac{b_0}{2W_0\left(\frac{b_0}{2\rho_0}\right)} + (b - b_0)\frac{\rho}{b_0}\right) = \frac{1}{e}\left(\frac{b_0}{2W_0\left(\frac{b_0}{2\rho_0}\right)} + \frac{b - b_0}{2W_0\left(\frac{b}{2\rho_0}\right)}\right) = \frac{1}{e}\frac{b}{2W_0\left(\frac{b_0}{2\rho_0}\right)}.
$$

Hence, the following choice of $s$ satisfies the Taylor approximation order constraint (21):

$$
s \triangleq \left\lfloor \beta R_\mathbf{Q} R_\mathbf{K} \frac{b}{2W_0\left(\frac{b_0}{2\rho_0}\right)} \right\rfloor = \left\lfloor \frac{\log\left(\frac{1}{\varepsilon^2}\right) + 2\beta R_\mathbf{Q} R_\mathbf{K}}{2W_0\left(\frac{\log(n)}{2\rho_0\beta R_\mathbf{Q} R_\mathbf{K}} + \frac{1}{\rho_0}\right)} \right\rfloor \geq \lfloor e\beta R_\mathbf{Q} R_\mathbf{K} l_b(\rho) \rfloor. \tag{23}
$$

Note, moreover, that our Taylor growth parameter $\sigma \geq s/\log(n)$.

We will now prove that our assumed constraint on $r$ (6) implies the rank bound (21) with our particular choice of $s$ (23). By Lem. F.1, we have

$$
\binom{s + d}{d} \leq \frac{1}{\sqrt{\pi}} \exp\left((s + d)\mathrm{Ent}\left(\frac{s}{s + d}\right)\right) \leq \frac{1}{\sqrt{\pi}} n^{(\sigma + \delta)\mathrm{Ent}\left(\frac{\sigma}{\sigma + \delta}\right)}
$$

where we recall that $\delta = d/\log(n)$. Moreover, since $\|\mathbf{H}\|_{\mathrm{op}} \leq n\|\mathbf{H}\|_{\max} \leq n\exp(\beta\tau^{-2}R_\mathbf{K}^2)$ by Cauchy-Schwarz, we have

$$
\log(\|\mathbf{H}\|_{\mathrm{op}}\exp(\beta\tau^2 R_\mathbf{Q}^2 + 2\beta R_\mathbf{Q} R_\mathbf{K})/(\varepsilon\sqrt{n})^2) \leq \log\left(\frac{1}{\varepsilon^2}\right) + \left(\rho + \frac{1}{\rho} + 2\right)\beta R_\mathbf{Q} R_\mathbf{K}.
$$

In addition, $\rho^{-1} + 2 \leq \rho_0^{-1} + 2 \leq 3$, and, since $\varepsilon \leq n^{-\frac{1}{2}}$, we have $\beta R_\mathbf{Q} R_\mathbf{K}\rho \leq \sigma\log(n)$. Therefore, the runtime inflation factor is bounded as

$$
\log(\|\mathbf{H}\|_{\mathrm{op}}\exp(\beta\tau^2 R_\mathbf{Q}^2 + 2\beta R_\mathbf{Q} R_\mathbf{K})/(\varepsilon\sqrt{n})^2) \leq \log\left(\frac{1}{\varepsilon^2}\right) + 3\beta R_\mathbf{Q} R_\mathbf{K} + \sigma\log(n) = \log(n^{2a + 3\gamma + \sigma}),
$$

confirming the sufficiency of our constraint (6) on $r$.

## I. Proof of Cor. 2: Refined super-polynomial error decay in near-linear time

In this section we provide general conditions on regimes in which we can provide fast attention approximation guarantees. We note conditions under which the binomial coefficient, which is the dominating contribution to $r$, is near constant:

**Lemma I.1** (Binomial coefficient growth). *For $s, d \in \mathbb{N}$, we have $\binom{s+d}{d} \in n^{o(1)}$ whenever either of the following two conditions holds:*

1. $\frac{s}{\log(n)} \in o(1)$ *and* $\frac{d}{\log(n)} \in n^{o(1/s)}$.

2. $\frac{d}{\log(n)} \in o(1)$ *and* $\frac{s}{\log(n)} \in n^{o(1/d)}$.

*In particular, $\exp\left((s + d)\mathrm{Ent}\left(\frac{s}{s+d}\right)\right) \in n^{o(1)}$ under either condition.*

*Proof.* By Lem. F.1 we have

$$
\binom{s + d}{d} \leq \frac{1}{\sqrt{\pi}}\left(1 + \frac{s}{d}\right)^d\left(1 + \frac{d}{s}\right)^s = \frac{1}{\sqrt{\pi}}\exp\left((s + d)\mathrm{Ent}\left(\frac{s}{s + d}\right)\right).
$$

To conclude $\binom{s+d}{d} \in n^{o(1)}$ it therefore suffices to establish

$$
-s\log\left(\frac{s}{s + d}\right) - d\log\left(\frac{d}{s + d}\right) \in o(\log(n)).
$$

First, consider $s \in o(\log(n))$. Then,

$$\exp\left(-d\log\left(\frac{d}{s+d}\right)\right) = \left(1 + \frac{s}{d}\right)^d \leq \exp(s) \in n^{o(1)} .$$

Hence, it suffices to have $-s\log\left(\frac{s}{s+d}\right) \in o(\log(n))$ to prove the claim. We have

$$
\begin{aligned}
-s\log\left(\frac{s}{s+d}\right) \in o(\log(n)) \quad &\Leftrightarrow \quad \frac{s+d}{s} \in \exp\left(\frac{o(\log(n))}{s}\right) \\
&\Leftrightarrow \quad \frac{d}{\log(n)} \in \frac{s}{\log(n)}\left(\exp\left(o(1)\frac{\log(n)}{s}\right) - 1\right) \\
&\Leftarrow \quad \frac{d}{\log(n)} \in \exp\left(o(1)\frac{\log(n)}{s}\right) .
\end{aligned}
$$

In the final equation it was used that $s/\log(n) \in o(1)$, and for any $f(n) \in \omega(1)$ it holds that

$$\exp(o(f(n)))/f(n) = \exp\left(f(n)\left(o(1) + \frac{\log(1/f(n))}{f(n)}\right)\right) = \exp(o(f(n))) .$$

The second case follows by a symmetric argument. $\qquad \square$

## Proof of Cor. 2: Refined super-polynomial error decay in near-linear time

*Proof.* For any $a(n) \in o\left(\frac{\log(1/\gamma(n))}{\max\{\log(\delta(n)),1\}}\right) \cap n^{o(1)}$ it holds in particular that $a(n) \in o(\log(1/\gamma(n)))$, since $\max\{\log(\delta(n)),1\} \in \Omega(1)$ is asymptotically bounded from below. Since we further assumed that $\gamma(n) \in o(1)$, it holds by Lem. I.2 that

$$\sigma(n) \leq \frac{a(n) + \gamma(n)}{c_1\log\left(1 + \frac{1}{2\rho_0\gamma(n)} + \frac{1}{\rho_0}\right)} \in O\left(\frac{a(n)}{\log(1/\gamma(n))}\right) \subseteq o(1) .$$

Let now $a(n) \in o\left(\frac{\log(1/\gamma(n))}{\max\{\log(\delta(n)),1\}}\right) \cap n^{o(1)}$ be fixed. Then, there exists a sequence $\alpha(n) \in o(1)$ such that $a(n) \leq \alpha(n)\left(\frac{\log(1/\gamma(n))}{\max\{\log(\delta(n)),1\}}\right)$ and we have

$$\delta(n) \leq \exp(\max\{\log(\delta(n)),1\}) \leq \exp\left(\alpha(n)\frac{\log(1/\gamma(n))}{a(n)}\right) \leq \exp\left(\alpha(n)\frac{1}{c\sigma(n)}\right) \in \exp(o(1)/\sigma(n)) .$$

Therefore, by Lem. I.1, $n^{(\sigma(n)+\delta(n))\mathrm{Ent}\left(\frac{\sigma(n)}{\sigma(n)+\delta(n)}\right)} \in n^{o(1)}$. Finally, we know from $a(n) + \gamma(n) \in n^{o(1)}$ that the logarithmic runtime inflation term is near constant $\log(n^{2a+\sigma+\gamma}) \in n^{o(1)}$. Therefore, any

$$r \geq 1 + \frac{1}{\sqrt{\pi}}n^{(\sigma+\delta)\mathrm{Ent}\left(\frac{\sigma}{\sigma+\delta}\right)}\log\left(n^{2a+\sigma+3\gamma}\right) \in n^{o(1)}$$

suffices to achieve $\mathbb{E}\|\mathbf{O} - \widehat{\mathbf{O}}\|_{\max} \leq 3\|\mathbf{V}\|_{\max}n^{-a(n)}$.

Alternatively, assume that $\delta(n) \in o(1)$ and $\gamma(n) \in \Omega(1) \cap n^{o(1/d)}, a(n) \in n^{o(1/d)}$. In particular, $a(n) + \gamma(n) \in n^{o(1/d)}$. Since $\gamma(n) \in \Omega(1)$, it holds by Lem. I.2 that

$$\sigma(n) \in O(a(n) + \gamma(n)) \subseteq n^{o(1/d)} .$$

Therefore, we conclude as before that there exists a runtime function $r \in n^{o(1)}$ that satisfies

$$r \geq 1 + \frac{1}{\sqrt{\pi}}n^{(\sigma+\delta)\mathrm{Ent}\left(\frac{\sigma}{\sigma+\delta}\right)}\log(n)n^{o(1/d)} \in n^{o(1)} .$$

$\qquad \square$

**Lemma I.2** (Asymptotic behaviour of Taylor approximation order). *The order parameter of the Taylor approximation* $\mathbf{T}^s$ *has the following asymptotic behaviour:*

$$\sigma(n) \triangleq \frac{s(\varepsilon = n^{-a(n)})}{\log(n)} = \frac{a(n) + \gamma(n)}{W_0\left(\frac{1}{2\rho_0\gamma(n)} + \frac{1}{\rho_0}\right)} \in \begin{cases} O(a(n)\log(1/\gamma(n))^{-1}) & \text{if} \quad \gamma(n) \in o(1) \\ O(a(n) + \gamma(n)) & \text{if} \quad \gamma(n) \in \Omega(1) . \end{cases}$$

*Proof.* First, assume that $\gamma(n) \in o(1)$. Then, by definition of $\sigma(n)$ we have

$$\frac{s(\varepsilon = n^{-a})}{\log(n)} = \frac{a + \gamma(n)}{W_0\left(\frac{1}{2\rho_0\gamma(n)} + \frac{1}{\rho_0}\right)} \leq \frac{a + \gamma(n)}{c_1 \log\left(1 + \frac{1}{2\rho_0\gamma(n)} + \frac{1}{\rho_0}\right)} \in O(a(n)\log(1/\gamma(n))^{-1}) .$$

because $\log\left(1 + \frac{1}{2\rho_0\gamma(n)} + \frac{1}{\rho_0}\right) \in \omega(1)$. Here, we used that $W_0(x) \geq c_1 \log(1 + x)$ with $c_1 = 0.6321$ (Orabona, 2019). Conversely, assume that $\gamma(n) \in \Omega(1)$. In this case, $W_0\left(\frac{1}{2\rho_0\gamma(n)} + \frac{1}{\rho_0}\right) \in O(1)$ and therefore

$$\frac{s(\varepsilon = n^{-a})}{\log(n)} \in O(a(n) + \gamma(n)) .$$

$\square$

# J. Proof of Tab. 1: Practical approximation guarantees

In this section we derive the guarantees of Tab. 1 given $m = n$, bounded dimension $d \in O(1)$, bounded entries

$$\beta R_{\mathbf{Q}}^2, \beta R_{\mathbf{K}}^2 \leq R^2 \in O(1), \tag{24}$$

and $O(dn^{1+t})$ runtime.

## J.1. WILDCAT guarantee

The stated result follows immediately from the more precise guarantee in Cor. J.1.

**Corollary J.1** (WILDCAT error as a function of runtime). *Suppose* $m = n$, $t \in (0, 1)$, $d \in O(1)$, *and* $\beta R_{\mathbf{Q}} R_{\mathbf{K}} \leq R \in O(1)$. *Define* $\kappa \triangleq 2\rho_0 + 1$. *Let* $r = n^{t/2}$ *so that the runtime of* WILDCAT *lies in* $O(n^{1+t} + dn^{1+t/2})$. *Then there exists a constant* $C > 0$ *such that*

$$\mathbb{E}\|\mathbf{O} - \widehat{\mathbf{O}}\|_{\max} \leq C \frac{\log(n)}{n^{0.14t(1+\log(1+\log(n)/(\kappa R)))}} \cdot \|\mathbf{V}\|_{\max} .$$

*Proof.* Matching the exponents in Thm. 2, our goal is finding a function $\sigma(n) = s/\log(n)$ such that

$$(\sigma + \delta)\text{Ent}\left(\frac{\sigma}{\sigma + \delta}\right) \leq t/2 - \lambda(n) \tag{25}$$

with

$$\lambda(n) = \frac{\log((2a + \sigma + 3\gamma)\log(n))}{\log(n)} .$$

Since $\gamma \in o(1)$ we have from Lem. I.2 that $\sigma \in O(a/\log(1/\gamma)) \subseteq o(a)$. Furthermore, since $t < 1$ we have for our claimed error rate $a < 0.29 \log(e + \log(n)/(\kappa R)) \in o(\log(n))$. Therefore, for $n$ large enough

$$\lambda(n) \leq \frac{\log(2a(1 + o(1))\log(n))}{\log(n)} \leq \frac{\log(\log(n)^2)}{\log(n)} \in o(1) .$$

We therefore know that $\lambda$ remains small relative to the other exponents in (25). We proceed with simplifying the left-hand side as in Lem. I.1 as

$$(\sigma + \delta)\text{Ent}\left(\frac{\sigma}{\sigma + \delta}\right) \leq \sigma \log\left(\frac{\sigma + \delta}{\sigma}\right) + \sigma .$$

Setting the right-hand side equal to $t/2 - \lambda(n)$ we find

$$\frac{\sigma}{\delta} \log\left(e\frac{\sigma + \delta}{\sigma}\right) = \frac{t - \lambda}{2\delta}.$$

With Lem. L.5 we can invert this relationship and find the following expression for the order parameter $\sigma$ as a function of $t$:

$$\sigma = \delta g\left(\frac{t - \lambda}{2\delta}\right)$$

where $g(y) = \frac{y}{y + W_{-1}(-\exp(-y-1)y)}$. Next, we relate $\sigma$ to the decay rate $n^{-a}$. Re-arranging (5) we have

$$a = W_0\left(\frac{1}{2\rho_0\gamma} + \frac{1}{\rho_0}\right)\sigma - \gamma.$$

Next, we use $W_0(x) \geq 0.6321 \log(1 + x)$ (Orabona, 2019, Theorem C.3) to state

$$W_0\left(\frac{1}{2\rho_0\gamma} + \frac{1}{\rho_0}\right) \geq 0.6321 \log\left(1 + \frac{1}{2\rho_0\gamma} + \frac{1}{\rho_0}\right)$$

$$= 0.6321\left(\log\left(1 + \frac{1}{(2\rho_0 + 1)\gamma}\right) + \log\left(1 + \frac{1}{\rho_0}\right)\right)$$

$$\geq 0.6321\left(\log\left(1 + \frac{1}{(2\rho_0 + 1)\gamma}\right) + 1\right).$$

We therefore obtain the following lower bound on the decay rate in $n^{1+t}$ time:

$$a \geq 0.6321\left(\log\left(1 + \frac{1}{(2\rho_0 + 1)\gamma}\right) + 1\right)\delta g\left(\frac{t - \lambda}{2\delta}\right) - \gamma.$$

Since $(t - \lambda)\log(n)/2d \in \omega(1)$, we have for $n$ large enough that $(t - \lambda)\log(n)/2d \geq 1$. We now use the convexity of $g$ to obtain the following

$$\delta g\left(\frac{t - \lambda}{2\delta}\right) = \frac{t - \lambda}{2}\frac{2\delta}{t - \lambda}g\left(\frac{t - \lambda}{2\delta}\right)$$

$$= \frac{t - \lambda}{2}\left(\frac{2\delta}{t - \lambda} - 0\right)\left(g\left(\frac{t - \lambda}{2\delta}\right) - g(0)\right)$$

$$\geq \frac{t - \lambda}{2}(1 - 0)(g(1) - g(0)) = \frac{t - \lambda}{2}g(1).$$

With $0.6321 * g(1)/2 \geq 0.14$ we obtain the final closed form lower-bound on the decay rate

$$a \geq 0.6321\left(\log\left(1 + \frac{1}{(2\rho_0 + 1)\gamma}\right) + 1\right)\frac{t - \lambda}{2}g(1) - \gamma$$

$$\geq 0.14\left(1 + \log\left(1 + \frac{1}{(2\rho_0 + 1)\gamma}\right)\right)t - \lambda - \gamma.$$

Finally, the decay rate simplifies to

$$n^{-a} \leq \frac{n^{\gamma + \lambda}}{n^{0.14\left(1 + \log\left(1 + \frac{1}{(2\rho_0 + 1)\gamma}\right)\right)t}} \leq \frac{\exp(2R)\log(n)}{n^{0.14\left(1 + \log\left(1 + \frac{1}{(2\rho_0 + 1)\gamma}\right)\right)t}}.$$

$\square$

### J.2. Thinformer guarantee

The runtime analysis of Carrell et al. (2025, Sec. 4.1) allows for a maximum coreset of size $n_{\text{out}} = \Theta(n^t)$ in time $O(dn^{1+t})$. Plugging this choice into the error analysis of Carrell et al. (2025, Thm. 2) yields the following constant probability bound on $\|\mathbf{O} - \widehat{\mathbf{O}}\|_{\max}$ (up to constants):

$$\frac{\sqrt{d\log(R^2\|V\|_{\max})}\exp(2R^2)\|\mathbf{V}\|_{2,\infty}\log(n)}{n^t}.$$

### J.3. BalanceKV guarantee

The analysis of Han et al. (2025, Thm. 3.1) guarantees that BalanceKV with batch size $B$ provides a high probability bound of order

$$\frac{\sqrt{d}\log(dn)\log_2(n/B)\exp(2R^2)\|V\|_F}{B} \tag{26}$$

on $\|\mathbf{O} - \widehat{\mathbf{O}}\|_{\max}$ in order $dnB\log_2(n/B)$ time. Plugging $B = n^t/\log_2(n)$ into (26) to achieve $dn^{1+t}$ runtime yields the result.

### J.4. KDEformer guarantee

The bounded entries assumption (24) implies that, for all $i, j \in [n]$,

$$\mathbf{A}_{ij} \in [e^{-R^2}, e^{R^2}], \qquad 1 \le |(\mathbf{D}^{-1}\mathbf{A})_{ij}| \le \frac{e^{R^2}}{e^{R^2} + (n-1)e^{-R^2}} = \frac{e^{2R^2}}{e^{2R^2} + n - 1},$$

$$\|\mathbf{D}^{-1}\mathbf{A}\|_{2,\infty}^2 \le \frac{e^{4R^2}n}{(e^{2R^2} + n - 1)^2} \in O\left(\frac{1}{n}\right), \quad \text{and} \quad 1 \le \|\mathbf{D}^{-1}\mathbf{A}\|_{\mathrm{op}}^2 \le \|\mathbf{D}^{-1}\mathbf{A}\|_F^2 \le n\|\mathbf{D}^{-1}\mathbf{A}\|_{2,\infty}^2 \in O(1). \tag{27}$$

In addition, the bounded dimension assumption implies that

$$\|\mathbf{V}\|_{\mathrm{op}}^2 \le \|\mathbf{V}\|_F^2 \le d\|\mathbf{V}^\top\|_{2,\infty}^2 \le d\|\mathbf{V}\|_{\mathrm{op}}^2 \in O(\|\mathbf{V}\|_{\mathrm{op}}^2).$$

Hence, the runtime analysis of Zandieh et al. (2023, Thm. 3.5) guarantees that KDEformer provides a high probability bound of order

$$\varepsilon\|\mathbf{D}^{-1}\mathbf{A}\|_{\mathrm{op}}\|\mathbf{V}\|_{\mathrm{op}} \in \Theta(\varepsilon\|\mathbf{V}\|_{\mathrm{op}}) \tag{28}$$

on $\|\mathbf{O} - \widehat{\mathbf{O}}\|_{\max}$ in order

$$\frac{d\left(n^{1+\xi} + n\log(n)\left(\frac{\|\mathbf{D}^{-1}\mathbf{A}\|_F^2}{\|\mathbf{D}^{-1}\mathbf{A}\|_{\mathrm{op}}^2} + \frac{\|\mathbf{V}\|_F^2}{\|\mathbf{V}\|_{\mathrm{op}}^2}\right)\right)}{\varepsilon^2} \in \Theta\left(\frac{dn^{1+\xi}}{\varepsilon^2}\right)$$

time. Plugging $\varepsilon = n^{\xi/2}/n^{t/2}$ into (28) to achieve order $dn^{1+t}$ runtime yields the result.

### J.5. HyperAttention guarantee

The bounded entries assumption (24) implies that

$$1 \le \kappa \triangleq \frac{\max_{i\in[n]}\sum_{j\in[n]}\mathbf{A}_{ij}}{\min_{i\in[n]}\sum_{j\in[n]}\mathbf{A}_{ij}} \le \frac{ne^{R^2}}{ne^{-R^2}} = e^{2R^2}. \tag{29}$$

Using (27) and (29), we find that the Hyperattention without masking analysis of Han et al. (2024, Thm. 1, Lem. 1, and Lem. 2) requires order

$$dn\left(\frac{\kappa^7 n\|\mathbf{D}^{-1}\mathbf{A}\|_{2,\infty}^2\log(n)}{\varepsilon^6} + \frac{d\kappa^2\|\mathbf{D}^{-1}\mathbf{A}\|_F^2/\|\mathbf{D}^{-1}\mathbf{A}\|_{\mathrm{op}}^2}{\varepsilon^2}\right) \in \Theta(dn\log(n)/\varepsilon^6)$$

time to guarantee an order

$$\varepsilon\|\mathbf{D}^{-1}\mathbf{A}\|_{\mathrm{op}}\|\mathbf{V}\|_{\mathrm{op}} \in \Theta(\varepsilon\|\mathbf{V}\|_{\mathrm{op}}) \tag{30}$$

high-probability bound on $\|\mathbf{O} - \widehat{\mathbf{O}}\|_{\max}$. Plugging $\varepsilon = (\log n)^{1/6}/n^{t/6}$ into (30) to achieve order $dn^{1+t}$ runtime yields the result.

## K. Proof of Prop. K.1: Recursive update of kernel inverse

**Proposition K.1** (Recursive update of kernel inverse). *Let $h(\mathbf{K}_\mathcal{S}, \mathbf{K}_\mathcal{S})$ be invertible, let $\mathbf{k} \in \mathbb{R}^d$ with*

$$h_{\mathrm{res}}(\mathbf{k}, \mathbf{k}) \triangleq h(\mathbf{k}, \mathbf{k}) - h(\mathbf{k}, \mathbf{K}_\mathcal{S})h(\mathbf{K}_\mathcal{S}, \mathbf{K}_\mathcal{S})^{-1}h(\mathbf{K}_\mathcal{S}, \mathbf{k}) > 0.$$

*Define the vector*

$$\mathbf{g} \triangleq \frac{1}{\sqrt{h_{\mathrm{res}}(\mathbf{k}, \mathbf{k})}}(h(\mathbf{K}_\mathcal{S}, \mathbf{K}_\mathcal{S})^{-1}h(\mathbf{K}_\mathcal{S}, \mathbf{k}), -1)^\top$$

*Then,* $\begin{pmatrix} h(\mathbf{K}_\mathcal{S}, \mathbf{K}_\mathcal{S}) & h(\mathbf{K}_\mathcal{S}, \mathbf{k}) \\ h(\mathbf{k}, \mathbf{K}_\mathcal{S}) & h(\mathbf{k}, \mathbf{k}) \end{pmatrix}$ *is invertible and*

$$\begin{pmatrix} h(\mathbf{K}_\mathcal{S}, \mathbf{K}_\mathcal{S}) & h(\mathbf{K}_\mathcal{S}, \mathbf{k}) \\ h(\mathbf{k}, \mathbf{K}_\mathcal{S}) & h(\mathbf{k}, \mathbf{k}) \end{pmatrix}^{-1} = \begin{pmatrix} h(\mathbf{K}_\mathcal{S}, \mathbf{K}_\mathcal{S})^{-1} & \mathbf{0}_r \\ \mathbf{0}_r^\top & 0 \end{pmatrix} + \mathbf{g}\mathbf{g}^\top$$

*Proof.* We derive the recursion using Gaussian elimination. We start with

$$\left( \begin{array}{cc|cc} h(\mathbf{K}_\mathcal{S}, \mathbf{K}_\mathcal{S}) & h(\mathbf{K}_\mathcal{S}, \mathbf{k}) & \mathbf{1}_{r \times r} & \mathbf{0}_r \\ h(\mathbf{k}, \mathbf{K}_\mathcal{S}) & h(\mathbf{k}, \mathbf{k}) & \mathbf{0}_r^\top & 1 \end{array} \right)$$

Multiplying the upper $r$ rows with $h(\mathbf{K}_\mathcal{S}, \mathbf{K}_\mathcal{S})^{-1}$ from the left yields

$$\left( \begin{array}{cc|cc} \mathbf{1}_{r \times r} & h(\mathbf{K}_\mathcal{S}, \mathbf{K}_\mathcal{S})^{-1}h(\mathbf{K}_\mathcal{S}, \mathbf{k}) & h(\mathbf{K}_\mathcal{S}, \mathbf{K}_\mathcal{S})^{-1} & \mathbf{0}_r \\ h(\mathbf{k}, \mathbf{K}_\mathcal{S}) & h(\mathbf{k}, \mathbf{k}) & \mathbf{0}_r^\top & 1 \end{array} \right)$$

Subtracting $-h(\mathbf{k}, \mathbf{K}_\mathcal{S})$ times the first $r$ rows from the last row produces the following matrix:

$$\left( \begin{array}{cc|cc} \mathbf{1}_{r \times r} & h(\mathbf{K}_\mathcal{S}, \mathbf{K}_\mathcal{S})^{-1}h(\mathbf{K}_\mathcal{S}, \mathbf{k}) & h(\mathbf{K}_\mathcal{S}, \mathbf{K}_\mathcal{S})^{-1} & \mathbf{0}_r \\ \mathbf{0}_r^\top & h_{\mathrm{res}}(\mathbf{k}, \mathbf{k}) & -h(\mathbf{k}, \mathbf{K}_\mathcal{S})h(\mathbf{K}_\mathcal{S}, \mathbf{K}_\mathcal{S})^{-1} & 1 \end{array} \right)$$

Next, divide the last row by $h_{\mathrm{res}}(\mathbf{k}, \mathbf{k})$ and subtract $h(\mathbf{K}_\mathcal{S}, \mathbf{K}_\mathcal{S})^{-1}h(\mathbf{K}_\mathcal{S}, \mathbf{k})$ of the last row from the first block to obtain

$$\left( \begin{array}{cc|cc} \mathbf{1}_{r \times r} & \mathbf{0}_r & h(\mathbf{K}_\mathcal{S}, \mathbf{K}_\mathcal{S})^{-1} + \frac{h(\mathbf{K}_\mathcal{S}, \mathbf{K}_\mathcal{S})^{-1}h(\mathbf{K}_\mathcal{S}, \mathbf{k})h(\mathbf{k}, \mathbf{K}_\mathcal{S})h(\mathbf{K}_\mathcal{S}, \mathbf{K}_\mathcal{S})^{-1}}{h_{\mathrm{res}}(\mathbf{k}, \mathbf{k})} & \frac{-h(\mathbf{K}_\mathcal{S}, \mathbf{K}_\mathcal{S})^{-1}h(\mathbf{K}_\mathcal{S}, \mathbf{k})}{h_{\mathrm{res}}(\mathbf{k}, \mathbf{k})} \\ \mathbf{0}_r^\top & 1 & \frac{-h(\mathbf{k}, \mathbf{K}_\mathcal{S})h(\mathbf{K}_\mathcal{S}, \mathbf{K}_\mathcal{S})^{-1}}{h_{\mathrm{res}}(\mathbf{k}, \mathbf{k})} & \frac{1}{h_{\mathrm{res}}(\mathbf{k}, \mathbf{k})} \end{array} \right)$$

The right hand side is the inverse of $h(\mathbf{K}_\mathcal{S} \cup \{\mathbf{k}\}, \mathbf{K}_\mathcal{S} \cup \{\mathbf{k}\})$. Furthermore, we find that indeed

$$\mathbf{g}\mathbf{g}^\top = \frac{1}{h_{\mathrm{res}}(\mathbf{k}, \mathbf{k})} \begin{pmatrix} h(\mathbf{K}_\mathcal{S}, \mathbf{K}_\mathcal{S})^{-1}h(\mathbf{K}_\mathcal{S}, \mathbf{k})h(\mathbf{k}, \mathbf{K}_\mathcal{S})h(\mathbf{K}_\mathcal{S}, \mathbf{K}_\mathcal{S})^{-1} & -h(\mathbf{K}_\mathcal{S}, \mathbf{K}_\mathcal{S})^{-1}h(\mathbf{K}_\mathcal{S}, \mathbf{k}) \\ -h(\mathbf{k}, \mathbf{K}_\mathcal{S})h(\mathbf{K}_\mathcal{S}, \mathbf{K}_\mathcal{S})^{-1} & 1 \end{pmatrix}$$

$\square$

## L. Properties of the Lambert W Function

A useful function for our statements is the Lambert W function, also known as the product logarithm:

**Definition L.1** (Lambert W function). *The principal branch of the Lambert W function $w = W_0(z)$ is the unique solution $w \in (-1, \infty)$ to the equation $w \exp(w) = z$ for $z > -\frac{1}{e}$.*

From the definition one immediately gets the following identity:

**Lemma L.1** (Lambert W exponential). *The Lambert W function satisfies $\exp(W_0(z)) = \frac{z}{W_0(z)}$ for all $z \neq 0$, and $\exp(W_0(0)) = 1$.*

By implicit differentiation one further finds the following:

**Lemma L.2** (Lambert W derivative). *The Lambert W function is increasing on $(-\frac{1}{e}, \infty)$ and has the derivative*

$$\frac{\mathrm{d}}{\mathrm{d}z} W_0(z) = \begin{cases} \frac{W_0(z)}{z(1+W_0(z))} & z \neq 0 \\ 1 & z = 0 \end{cases} \tag{31}$$

*In addition, the ordinary differential equation $\frac{\mathrm{d}}{\mathrm{d}t}x(t) = \frac{x}{x+t}$ with $x(0) = x_0$ has the solution*

$$x(t) = \frac{t}{W_0\left(\frac{t}{x_0}\right)} .$$

*Proof.* $z \mapsto W_0(z)$ is increasing as the inverse of the increasing function $w \mapsto w\exp(w)$. By the definition of the Lambert W function we have for all $z \in (-\frac{1}{e}, \infty)$

$$\frac{\mathrm{d}}{\mathrm{d}z}(W_0(z)\exp(W_0(z))) = (1 + W_0(z))\frac{\mathrm{d}}{\mathrm{d}z}(W_0(z))\exp(W_0(z)) = 1 .$$

Rearranging for $\frac{\mathrm{d}}{\mathrm{d}z}(W_0(z))$ gives

$$\frac{\mathrm{d}}{\mathrm{d}z}(W_0(z)) = \frac{1}{\exp(W_0(z))(1+W_0(z))} .$$

We find $\frac{\mathrm{d}}{\mathrm{d}z}(W_0(z))|_{z=0} = 1$ by direct evaluation. Invoking $\exp(W_0(z)) = \frac{z}{W_0(z)}$ for $z \neq 0$ gives (31). Next, applying the derivative formula to $x(t)$ yields

$$\frac{\mathrm{d}}{\mathrm{d}t}x(t) = \frac{\mathrm{d}}{\mathrm{d}t}x_0 \exp\left(W_0\left(\frac{t}{x_0}\right)\right) = x_0 \exp\left(W_0\left(\frac{t}{x_0}\right)\right)\frac{1}{\exp\left(W_0\left(\frac{t}{x_0}\right)\right)\left(1+W_0\left(\frac{t}{x_0}\right)\right)}\frac{1}{x_0}$$

$$= \frac{1}{1+W_0\left(\frac{t}{x_0}\right)} = \frac{x(t)}{x(t)+t} .$$

Furthermore, $x(0) = x_0 \exp(W_0(0)) = x_0$. $\qquad\square$

**Lemma L.3** (Derivative of the Lambert W exponential). *For $z > 0$, the function $\frac{z}{W_0(z)}$ is concave and has derivative* $\frac{\mathrm{d}}{\mathrm{d}z}\left(\frac{z}{W_0(z)}\right) = (W_0(z)+1)^{-1}$.

*Proof.* Using (31) and $\frac{z}{W_0(z)} = \exp(W_0(z))$, we find

$$\frac{\mathrm{d}}{\mathrm{d}z}\left(\frac{z}{W_0(z)}\right) = \frac{\mathrm{d}}{\mathrm{d}z}\exp(W_0(z)) = \frac{W_0(z)}{z(W_0(z)+1)}\exp(W_0(z)) = \frac{1}{W_0(z)+1} .$$

In addition, $z \mapsto (1+W_0(z))^{-1}$ is decreasing for $z > 0$, and consequently $\frac{z}{W_0(z)}$ is concave. $\qquad\square$

We use the following logarithmic lower bound on the Lambert W function, proved by Orabona (2019, Thm. C.3):

**Lemma L.4** (Lambert W lower bound (Orabona, 2019, Thm. C.3)). *It holds for $z \geq 0$ that $W_0(z) \geq 0.6321\log(1+z)$.*

For numerical simulations we want a stable estimate of the Lambert W function and we find that standard implementations in Python packages like SciPy may be insufficient and not parallelised. A better estimate is obtained via the iterations proposed by Lóczi (2022).

**Theorem L.1** (Fast Lambert W calculation (Lóczi, 2022, Thms. 2.4 and 2.9)). *For $z > 0$, let*

$$\beta_0 = \begin{cases} \log(z) - \log\log(z) & \text{for} \quad z > e \\ \exp(\log(z) - 1) & \text{for} \quad z < e \end{cases}$$

*and define the iteration*

$$\beta_{n+1} = \frac{\beta_n}{1+\beta_n}(1 + \log(z) - \log(\beta_n)) .$$

*Then*

$$0 < \beta_n - W_0(z) < \max\left(0.32^{(2^n)}, \frac{1}{3}0.633^{(2^n)}\right) .$$

We will also use the following identity and bounds concerning the secondary branch $W_{-1}$ of the Lambda W function.

**Lemma L.5** (Lambda W secondary branch properties). *Define*

$$b : \mathbb{R}_{>0} \to \mathbb{R}_{>0}, \quad z \mapsto z \log\left(e\frac{z+1}{z}\right)$$

*and*

$$g : \mathbb{R}_{>0} \to \mathbb{R}_{>0}, \quad y \mapsto \triangleq -\frac{y}{y + W_{-1}(-\exp(-y-1)y)} \,.$$

*where $W_{-1}(\bullet)$ is the secondary branch of the Lambert-W function with input range $(-1/e, 0)$. Then, $b \circ g(y) = y$ and $g(y) \leq y$.*

*Proof.* Let $w \triangleq W_{-1}(-\exp(-y-1)y)$. First, computing the argument of the logarithm we find:

$$\frac{g(y)+1}{g(y)} = \frac{-y/(y+w)+1}{-y/(y+w)} = \frac{-y+y+w}{-y} = \frac{-w}{y} \,.$$

Using the definition of the Lambert-W function $W_{-1}(x)\exp(W_{-1}(x)) = x$ we find

$$-w = -W_{-1}(-\exp(-y-1)y) = \exp(-y-1)y\exp(-w) \,.$$

Therefore,

$$\log(-w) = \log(y) - y - 1 - w \quad \Longrightarrow \quad \log\left(\frac{g(y)+1}{g(y)}\right) = \log\left(\frac{-w}{y}\right) = -y - 1 - w \,.$$

Consequently,

$$b(g(y)) = \frac{-y}{y+w}\left(\log\left(\frac{g(y)+1}{g(y)}\right) + 1\right) = \frac{-y}{y+w}(-y-1-w+1) = y \,.$$

$\square$

# M. Supplementary Experiment Details

All experiments were run using Python 3.12.12 on an Ubuntu 22.04.5 LTS server with a single NVIDIA A100 GPU (80 GB memory, CUDA 13.0, driver version 580.126.09), two 48-core AMD EPYC 7V13 processors, and 220 GB RAM.

### M.1. Supplementary details for Sec. 4.1

The Tab. 2 experiment was run using PyTorch 2.10.0.dev20251019+cu129. We used CUDA events to time the forward pass through each (approximate) `attention-matrix` layer after initializing the GPU with 20 warm-up batches. The implementations and settings for all methods other than WILDCAT were taken from `https://github.com/microsoft/thinformer`, and our experiment builds on this open-source repository.

### M.2. Supplementary details for Sec. 4.2

The Tab. 3 experiment was run using PyTorch 2.10.0.dev20251019+cu129. Timings were based on the first 50 batches of the ImageNet 2012 validation set. We used CUDA events to time the forward pass through each (approximate) `attention_layer` after initializing the GPU with 10 warm-up batches. The implementations and settings for all methods other than WILDCAT were taken from `https://github.com/microsoft/thinformer`, and our experiment builds on this open-source repository.

### M.3. Supplementary details for Sec. 4.3

The Tab. 4 experiment was run using PyTorch 2.8.0+cu128 and `kvpress` version 0.3.0. The implementations and settings for BalanceKV and Uniform were taken from `https://github.com/ksheth96/BalanceKV`. The implementations and default settings of all other methods save WILDCAT were taken from `https://github.com/NVIDIA/kvpress`, and our experiment builds on this open-source repository.

Our KV cache compression experiments focus on the memory reduction benefits of compression. When memory is the primary bottleneck (as is often the case on resource-constrained devices or for especially large contexts), one is typically willing to incur additional runtime costs for improved memory efficiency. Fortunately, we find that the COMPRESSKV overhead is small relative to leading alternatives. For example, to process 32k tokens with $75\%$ compression, the prefill time with SnapKV vs. COMPRESSKV is 3.38s vs. 3.43s (2% overhead).

### M.4. Supplementary details for Sec. 4.4

The Fig. 3 experiment was run using PyTorch 2.12.0+cu130. For each $n$, we reported the median runtime and mean approximation error over 100 replicates of the experiment. We repeat this experiment with varying rank and bin count parameters $r \in \{64, 128, 256, 512\}$ and $B \in \{2, 16, 64\}$ and display the runtime vs. accuracy curves in Fig. M.1.

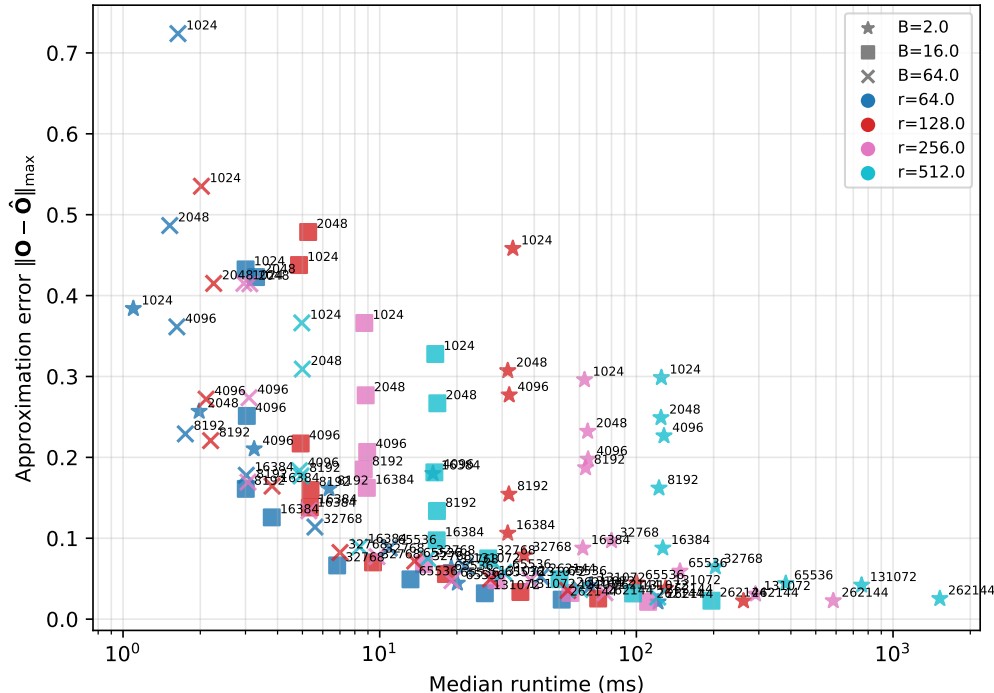

*Figure M.1.* Time-accuracy tradeoff curves for WILDCAT with varying rank and bin count parameters $(r, B)$.

