# OpenReview forum: "WildCat: Near-Linear Attention in Theory and Practice"
_ICML.cc/2026/Conference — ICML 2026 regular_

### Official Review · Reviewer_zBLn · 2026-03-11

**Soundness:** 3
**Presentation:** 3
**Significance:** 3
**Originality:** 3
**Overall Recommendation:** 5
**Confidence:** 3

**Summary:**

The paper proposes WILDCAT, a weighted coreset approach for approximating softmax attention. The key idea is to select a subset of keys via randomly pivoted Cholesky and then compute Nyström-style weights so that attention is performed on a compressed weighted support. The paper aims to combine three desirable properties: near-linear runtime, strong approximation guarantees, and practical GPU efficiency.

**Compliance With Llm Reviewing Policy:**

Affirmed.

**Final Justification:**

During the rebuttal, most of my concern have been addressed by the authors. I suggest to accept this submission.

**Key Questions For Authors:**

1. In Lemma 3, the paper defines $$ T^s_{il} = \sum_{p=0}^s \left(\frac{\beta}{\tau^2}\langle k_i, k_l\rangle\right)^p. $$ However, this expression does not appear to be the Taylor polynomial of the exponential kernel, since the factorial term is missing. For an (s)-th order Taylor truncation of (e^x), I would expect $$ \sum_{p=0}^s \frac{x^p}{p!}, $$ and therefore $$ T^s_{il} = \sum_{p=0}^s \frac{1}{p!}\left(\frac{\beta}{\tau^2}\langle k_i, k_l\rangle\right)^p. $$ Could the authors clarify whether the displayed definition in Lemma 3 contains a typo? If not, it is unclear to me how the subsequent trace-approximation claim $$ \operatorname{tr}(H_\tau - T^s) \le \varepsilon $$ is obtained from a Taylor-expansion argument.

2.In Lemma D.2, Eq. (9) is written as $$ \lambda_i(\Phi^r(H))
\lambda_i(\Phi^{r-1}(H))
\frac{\lambda_i(\Phi^{r-1}(H))^2}{\sum_{l=1}^n \lambda_i(\Phi^{r-1}(H))}. $$ I suspect the denominator should instead be $$ \sum_{l=1}^n \lambda_l(\Phi^{r-1}(H))
\operatorname{tr}(\Phi^{r-1}(H)), $$ since the definition $$ \Phi(A) = A - \frac{A^2}{\operatorname{tr}(A)} $$ would imply the eigenvalue-wise recurrence $$ \lambda_i(\Phi(A))
\lambda_i(A) - \frac{\lambda_i(A)^2}{\operatorname{tr}(A)}. $$ As currently written, the denominator repeats (\lambda_i) inside the summation, which seems inconsistent with the definition of (\Phi). Could the authors clarify whether this is a typo?

**Limitations:**

yes

**Strengths And Weaknesses:**

Strengths

1. Approximating softmax attention with both strong theoretical guarantees and practical efficiency is a central challenge. I find the motivation convincing, especially the claim that prior work often provides either good practical speedups without strong guarantees, or stronger guarantees without a comparably practical implementation.

2. Besides, the paper provides an operator-norm guarantee for the randomly pivoted Nyström procedure and then leverages polynomial approximation of the exponential kernel to argue fast decay of the kernel approximation error.

3. The empirical section is strong overall. I found the BigGAN and T2T-ViT results particularly convincing, since WILDCAT remains close to exact attention while still giving substantial speedups. The LongBench-E KV compression results are also competitive and appear to be the best on average among the compared methods.

weaknesses

My main concern is that the paper’s strongest theoretical claims are significantly more ambitious than the level of empirical validation currently provided. The paper’s key theoretical point is the super-polynomial error decay in near-linear time, but the experiments do not directly probe the regimes in which this asymptotic advantage becomes visible. The benchmarks demonstrate strong practical performance, but they do not clearly connect the asymptotic theory to observed scaling with sequence length, coreset size, or dimension.


While the proposed method has several meaningful design choices like coreset size r, number of bins B, the rescaling strategy, and so on. However, the paper mainly reports end-to-end comparisons, and does not provide enough sensitivity analysis which should show how robust the method is to these choices or how much each ingredient contributes.  It is particularly important because binning is introduced for GPU parallelism and may meaningfully affect approximation quality.

---

> ### Author Rebuttal · Authors · 2026-03-30
>
> Thank you for the detailed review. We are delighted that you find the empirical section strong and the motivation convincing. Below we address the opportunities for improvement.
>
> **Connecting theory and practice**
>
> In our revision we will include additional numerical evidence that the assumptions underlying our guarantees are indeed satisfied in real transformer layers used in practice. For example, Cor. 2 implies super-polynomially decaying error rates when the model dimension $d$ is bounded and the rescaled input entries $\gamma(n) := \frac{R_{\mathbf Q} R_{\mathbf K}}{\sqrt{d} \log(n)}$ are bounded. We compute $\gamma(n)$ from the queries and keys of the Qwen/Qwen2.5-7B-Instruct model for the first $n$ tokens of sequences sampled from the LongBench-E qasper_e dataset and observe that $\gamma(n)$ is *decreasing* with $n$ and hence also bounded:
>
> | n | gamma |
> | --- | --- |
> | 4 | 14.95 |
> | 16 | 9.55 |
> | 64 | 7.48 |
> | 256 | 6.70 |
> | 1024 | 6.23 |
> | 4096 | 5.86 |
> | 16384 | 5.63 |
>
> In addition, when one is focused on scaling the context length of a specific model, the dimension $d$ is always fixed and hence bounded. The assumptions to Cor. 2 are therefore satisfied in our KV cache compression experiment.
>
> In the revision, we can also complement this empirical evidence with the following theoretical intuition, describing sufficient conditions under which $\gamma(n)$ is necessarily bounded. If the queries and keys have i.i.d. sub-Gaussian entries, then, with probability 1, $\gamma(n) \in O\left(\frac{\sqrt{d}}{\log(n)} + \frac{1}{\sqrt{d}}\right)$.
>
> **New scaling and sensitivity experiments**
>
> In response to your feedback we have carried out several new parameter scaling and sensitivity experiments. The first benchmarks WildCat against the highly optimized FlashAttention 2 (FA) kernel with i.i.d. Gaussian (Q,K,V) inputs at various sequence lengths. With fixed $r = 32$ and $B = 16$, we observe that FA runtime increases quadratically while WildCat runtimes increase linearly, even as WildCat approximation error *improves* with n.
>
> | seq_len | flash (ms) | wildcat (ms) | max_abs_error |
> | --- | --- | --- | --- |
> | 1024 | 4.55 | 0.56 | 0.5020 |
> | 2048 | 17.91 | 8.79 | 0.3501 |
> | 4096 | 81.09 | 10.33 | 0.2783 |
> | 8192 | 317.70 | 20.51 | 0.1722 |
> | 16384 | 1158.68 | 96.29 | 0.0908 |
> | 32768 | 4709.06 | 188.92 | 0.0767 |
>
> Improved scaling can be observed for any fixed choice of r and B, as can be seen at https://imgur.com/a/PnitSNx. Varying r and B, as well as the choice of device, affects the point at which the runtimes of Flash Attention and WildCat intersect. Generally, increasing r/B increases runtime but decreases error (see https://imgur.com/a/subjJgS), and WildCat is faster than FA, even for relatively short sequences. We further believe that hardware-aware optimizations used by FA could also be integrated into WildCat, achieving even greater speed-ups. We will include our ablations in a revision of our work. We hope this addresses your question and strengthens our work.
>
> **Typos**
>
> We will fix both typos in the revision; thank you!

---

> > ### Author Rebuttal · Reviewer_zBLn · 2026-03-31
> >
> > Thank you for your detailed response, most of my concern have been addressed.

---

### Official Review · Reviewer_faYq · 2026-03-11

**Soundness:** 3
**Presentation:** 3
**Significance:** 3
**Originality:** 3
**Overall Recommendation:** 4
**Confidence:** 4

**Summary:**

This paper introduces WILDCAT, an attention mechanism compression framework designed to mitigate the quadratic computational bottleneck $O(n^2)$ inherent in standard Transformer architectures. The core innovation lies in the integration of the Randomly Pivoted Cholesky (RPC) algorithm with Nyström approximation to reconstruct the attention matrix using a spectrally accurate weighted coreset. The authors demonstrate that WILDCAT achieves super-polynomial error decay while maintaining near-linear time complexity $O(n^{1+o(1)})$, surmounting the limitations of previous methods that only achieved polynomial convergence under similar constraints. Empirically, a "Binning" strategy is proposed to parallelize the inherently sequential RPC algorithm. The framework's efficacy is validated across diverse tasks, including image generation (BigGAN), Vision Transformers (T2T-ViT), and KV cache compression for long-context Large Language Models (LLMs).

**Compliance With Llm Reviewing Policy:**

Affirmed.

**Final Justification:**

The authors addressed most of my concerns during the rebuttal; I therefore maintain my positive score.

**Key Questions For Authors:**

1. Regarding Causal Masking: During the decoding phase of LLMs, the sequence grows incrementally. Does WILDCAT support incremental RPC updates? If the compression weights $W$ must be recomputed at every generation step, can the computational overhead still be maintained at a near-linear level?

2. Sensitivity to Outliers: The calculation of the optimal parameter $\tau$ relies on the global maximum norms $R_Q$ and $R_K$. In practical inference, would the presence of outliers cause $\tau$ to become imbalanced, leading to numerical instability? Is there a smoothing or clipping mechanism to mitigate this?

3. Hardware Benchmarking: On A100/H100 architectures (or others), how does the end-to-end latency of the WILDCAT operator compare with highly optimized kernels like FlashAttention at common sequence lengths (e.g., 2k–4k tokens)?

**Limitations:**

yes

**Strengths And Weaknesses:**

Strengths

- Addressing the memory and compute demands of long-context processing, WILDCAT provides a "feature distillation" scheme with superior mathematical guarantees compared to heuristic "pruning" methods. It maintains high-fidelity approximation even when input dimensions grow super-logarithmically with sequence length.

- The work successfully bridges numerical linear algebra and deep learning by introducing the RPC algorithm to attention approximation, replacing traditional random feature projections. The use of the Lambert-W function to optimize the asymmetric rescaling balance between $Q$ and $K$ is a novel and insightful contribution.

- The paper presents a rigorous and complete theoretical framework. The proof chain is well-constructed: Lemma 1 establishes the impact of entry-wise errors on global output; Lemma 2 utilizes Reproducing Kernel Hilbert Space (RKHS) properties to shift the error analysis to the Key set; Theorem 1 proves the optimality of RPC in residual elimination; and Theorem 2 finalizes the super-polynomial convergence rate.

Weaknesses

- Causal Masking Constraints: While the method performs excellently on global attention tasks (e.g., vision, long-context pre-filling), it lacks native support for causal masking, which limits its immediate applicability to real-time autoregressive decoding/streaming tasks.

- Data Distribution Dependency: Despite strong theoretical bounds, the algorithm's empirical performance is highly contingent on the spectral properties of the input data. In scenarios where data lacks dominant principal components, the representativeness of the coreset may diminish significantly.

- Engineering Complexity: Compared to lightweight heuristic pruning (e.g., SnapKV), WILDCAT involves a more complex pre-processing pipeline (sampling, weight calculation, re-centering). The resulting computational overhead may offset the acceleration gains for short-to-medium sequence lengths.

---

> ### Author Rebuttal · Authors · 2026-03-30
>
> Thank you for your review and thoughtful questions. We are delighted that you found our theoretical framework rigorous and complete, our construction novel and insightful, and our work a successful bridge between numerical linear algebra and deep learning.
>
> **New FlashAttention 2 comparison**
>
> Following your suggestion we have benchmarked WildCat attention evaluation against the highly optimized FlashAttention 2 (FA) kernel for sequence lengths ranging from 1024 to 32768 on a RTX 3090 GPU. For $r = 32$ and $B=16$, WildCat’s speed-up over FA ranges from 2x at n=2048 to 25x at n=32K with **improved** approximation error as n increases. We will include such a comparison along with ablations over r and B in the revision. We further believe that hardware-aware optimizations used by FA could also be integrated into WildCat, achieving even greater speed-ups.
>
> | seq_len | flash (ms) | wildcat (ms) | max_abs_error |
> | --- | --- | --- | --- |
> | 1024 | 4.55 | 0.56 | 0.5020 |
> | 2048 | 17.91 | 8.79 | 0.3501 |
> | 4096 | 81.09 | 10.33 | 0.2783 |
> | 8192 | 317.70 | 20.51 | 0.1722 |
> | 16384 | 1158.68 | 96.29 | 0.0908 |
> | 32768 | 4709.06 | 188.92 | 0.0767 |
>
> **KV cache compression efficiency and comparison to heuristic pruning**
>
> In the revision we will clarify that KV cache compression presents two different potential benefits to users. The first, which we have focused on principally in this work, is the memory benefit, that is, reducing the memory costs of KV cache storage from n key-value pairs to $r \ll n$ key value pairs. When memory is the primary bottleneck (as is often the case on resource-constrained devices or for especially large contexts), one is typically willing to incur additional runtime costs in the prefill stage for improved memory efficiency (i.e., a higher-quality cache of a target budget size r) in the decoding stage. A second potential benefit (which we did not discuss in the submission) concerns speed improvements: m new tokens can be generated in order rm + m^2 time instead of nm + m^2 time. For this use case, the runtime of CompressKV is more relevant. Fortunately, we find that the CompressKV overhead is small relative to the prefill time required to compute the original n candidate key-value pairs. For example, to process 32K tokens and compress the cache to 1/4 of its size using Qwen2.5-7B-Instruct on an A100, the prefill time with SnapKV vs CompressKV(B=r/12) is 3.38 vs 3.43s (2% overhead).
>
> **Bounded entries imply spectral decay**
>
> We will clarify in the revision that our Cor. 2 implies the following universal guarantee for attention matrix spectral decay: for any given model, all $\gamma = \frac{R_{\mathbf Q} R_{\mathbf K}}{\sqrt{d} \log(n)}$-bounded token sequences generate sufficient spectral decay for WildCat to enjoy superpolynomial error in near-linear time. The $\gamma$-boundedness condition can be cheaply verified, and in doing so for the first $n$ tokens of sampled qasper_e sequences with Qwen2.5-7B-Instruct, we find that $\gamma$ is not only bounded but is in fact *decreasing* with $n$.
>
> | n | gamma |
> | --- | --- |
> | 4 | 14.95 |
> | 16 | 9.55 |
> | 64 | 7.48 |
> | 256 | 6.70 |
> | 1024 | 6.23 |
> | 4096 | 5.86 |
> | 16384 | 5.63 |
>
> **Outliers**
>
> In each of our attention approximation tasks we compute $R_Q$ and $R_K$ exactly as described in the paper and have not encountered any stability issues due to outliers. That said, any abnormally large query norm can be flagged during the norm computation step and separately handled by exactly computing the attention matrix row for that query. Similarly, any abnormally large key norm can be flagged and handled by deterministically including that key in the coreset (which ensures its attention matrix column is exactly computed).
>
> **Causal masking**
>
> Our submission focuses on non-causal attention approximation and KV cache compression, but, as outlined in Sec. 5, WildCat can be converted into a near-linear-time causal attention approximation via prefix sums or divide-and-conquer schemes (as in Han et al. 2024). The recursion tree can be parsed in causal order and requires no adjustments to the compression routine, the RPC sampling, or knowledge of the input tensors ahead of time. See figure https://imgur.com/a/FuiUzhr for an example decomposition of a causal attention matrix. Once a non-causal submatrix is fully realised, the associated keys and values are compressed, thereby reducing inference costs for future tokens in the stream. We view the rigorous theoretical and empirical analysis of causal WildCat as the ideal focal point of future work.
>
> **Comparison to previous work**
> > WILDCAT achieves super-polynomial error decay while maintaining near-linear time complexity , […] previous methods […] only achieved polynomial convergence under similar constraints.
> >
>
> In fact, as we note in the intro, previous guarantees for practical approximations only established *sub*-polynomial ($n^{-o(1)}$) convergence rates in near-linear time.

---

> > ### Author Rebuttal · Reviewer_faYq · 2026-04-03
> >
> > My concerns have been addressed. I maintain the positive score.

---

### Official Review · Reviewer_Yjbf · 2026-03-12

**Soundness:** 3
**Presentation:** 3
**Significance:** 3
**Originality:** 3
**Overall Recommendation:** 4
**Confidence:** 4

**Summary:**

This paper introduces WILDCAT, an approximate attention method based on selecting a small weighted coreset of keys using randomly pivoted Cholesky and then optimally reweighting them for attention reconstruction. The main claim is that WILDCAT closes part of the theory–practice gap: it achieves near-linear runtime when the coreset size is subpolynomial, while also giving super-polynomial approximation error decay under the paper’s assumptions. The authors further provide a GPU-optimized PyTorch implementation and evaluate the method on image generation, image classification, and KV-cache compression.

**Compliance With Llm Reviewing Policy:**

Affirmed.

**Final Justification:**

I maintain my score at present.

**Key Questions For Authors:**

1. How sensitive are the practical results to the coreset size 𝑟 and the number of bins 𝐵? A clearer speed-accuracy tradeoff study for these two knobs would strengthen the work.

2. The theoretical guarantees are one of the main selling points. Can the authors give more intuition for how often the bounded-input and related assumptions are approximately satisfied in real transformer layers used in practice?

3. For KV cache compression, could the authors add more evidence on runtime and memory tradeoffs in realistic long-context inference pipelines, not only benchmark quality at a fixed 25% cache budget?

**Limitations:**

Yes.

**Strengths And Weaknesses:**

Strength

1. The paper does not only propose a practical approximation, but also provides a nontrivial theoretical story for why the method should work.

2. The resulting complexity drops from quadratic attention cost to O(rnd+mrd+nr^2), and the paper argues stronger asymptotic guarantees than prior practical methods under the stated assumptions.

3. The empirical section is also strong.

Weaknesses

1. The strongest theoretical claims rely on assumptions that may be hard to map directly onto real transformer behavior, so the practical meaning of the asymptotic guarantees could be clarified more carefully.

2. The language-model side is narrower than the vision side: the KV-cache study uses one main LLM setup and reports compression quality rather than a broader end-to-end serving analysis.

3. It seems the method still inherits some sequential/path-dependent structure from the pivot selection routine, and although binning helps throughput, it is not obvious how this behaves in the most latency-sensitive causal generation settings.

---

> ### Author Rebuttal · Authors · 2026-03-30
>
> We thank you for your thorough review and are delighted that you found both our empirical evidence and our guarantees strong. We address each of your questions below.
>
> **Validity of theoretical assumptions in real transformer layers**
>
> Thank you for this recommendation! In our revision we will include numerical evidence that the assumptions underlying our guarantees are indeed satisfied in real transformer layers used in practice. For example, Cor. 2 implies super-polynomially decaying error rates when the model dimension $d$ is bounded and the rescaled input entries $\gamma(n) := \frac{R_{\mathbf Q} R_{\mathbf K}}{\sqrt{d} \log(n)}$ are bounded. We compute $\gamma(n)$ from the queries and keys of the Qwen/Qwen2.5-7B-Instruct model for the first $n$ tokens of sequences sampled from the LongBench-E qasper_e dataset and observe that $\gamma(n)$ is decreasing with $n$ and hence also bounded:
>
> | n | gamma |
> | --- | --- |
> | 4 | 14.95 |
> | 16 | 9.55 |
> | 64 | 7.48 |
> | 256 | 6.70 |
> | 1024 | 6.23 |
> | 4096 | 5.86 |
> | 16384 | 5.63 |
>
> In addition, when scaling the context length of a specific model, the dimension $d$ is fixed and hence bounded. The assumptions to Cor. 2 are therefore satisfied in our KV cache compression experiment.
>
> In the revision, we complement this empirical evidence with the following theoretical intuition, describing sufficient conditions under which $\gamma(n)$ is necessarily bounded. If the queries and keys have i.i.d. sub-Gaussian entries, then, with probability 1, $\gamma(n) \in O\left(\frac{\sqrt{d}}{\log(n)} + \frac{1}{\sqrt{d}}\right)$.
>
> **Sensitivity of practical results to the coreset size 𝑟 and the number of bins 𝐵**
>
> Following this suggestion, we have conducted several new speed-accuracy tradeoff analyses. The first benchmarks WildCat against the highly optimized FlashAttention 2 (FA) kernel with i.i.d. Gaussian (Q,K,V) inputs on a RTX 3090 GPU at various sequence lengths. With fixed $r = 32$ and $B = 16$, we observe that FA runtime increases quadratically while WildCat runtimes increase linearly, even as the approximation error improves with n.
>
> | seq_len | flash (ms) | wildcat (ms) | max_abs_error |
> | --- | --- | --- | --- |
> | 1024 | 4.55 | 0.56 | 0.5020 |
> | 2048 | 17.91 | 8.79 | 0.3501 |
> | 4096 | 81.09 | 10.33 | 0.2783 |
> | 8192 | 317.70 | 20.51 | 0.1722 |
> | 16384 | 1158.68 | 96.29 | 0.0908 |
> | 32768 | 4709.06 | 188.92 | 0.0767 |
>
> Improved scaling can be observed for any fixed choice of r and B, as can be seen at https://imgur.com/a/PnitSNx. Varying r and B, as well as the choice of device, affects the point at which the runtimes of Flash Attention and WildCat intersect. Generally, increasing r/B increases runtime but decreases error. (https://imgur.com/a/subjJgS) We will include these ablations in a revision of our work.
>
> **Runtime and memory tradeoffs in KV cache compression**
>
> Thank you for your recommendations. We will highlight in the revision that our KV cache compression experiments replicate the benchmark setup of Han et al., 2025 which uses a fixed 25% cache budget throughout. Our primary target was reducing the memory costs of KV cache storage from n key-value pairs to $r \ll n$ key value pairs. When memory is the primary bottleneck, such compression methods enable language modelling over extended context windows. In the revision, we will also include results for other cache budgets to give a fuller picture of quality-cost trade-offs. Here are example results for a 12.5% budget:
>
> | **Method** | **qasper** | **multifield** | **hotpot** | **2wiki** | **gov** | **multinews** | **trec** | **trivia** | **samsum** | **p.count** | **p.ret** | **lcc** | **repop** | **average** |
> | --- | --- | --- | --- | --- | --- | --- | --- | --- | --- | --- | --- | --- | --- | --- |
> | **SnapKV** | 16.36 | 25.74 | 35.15 | 24.96 | 26.38 | 20.76 | 45.17 | 88.29 | 37.09 | 14.00 | 58.11 | 68.84 | 56.41 | 39.79 |
> | **CompressKV** | 23.16 | 30.14 | 35.27 | 24.96 | 27.70 | 21.16 | 42.33 | 83.23 | 34.33 | 14.67 | 87.06 | 63.46 | 60.05 | **42.12** |
>
> While not the focus in our paper, reduced decoding time is a second potential benefit of KV cache compression: m new tokens can be generated with rm + m^2 time instead of nm + m^2 time. We find that the overhead of CompressKV is small relative to the computation time of the original n key-value pairs: To process 64K tokens using Qwen2.5-7B-Instruct on an A100, the prefill time with and without CompressKV is 8.94 vs 8.76s (2% overhead).
>
> **Causal generation**
>
> Our submission focuses on non-causal attention approximation, which is ideally suited for the compression problems that arise in vision and also for KV cache compression. However, as outlined in Sec. 5, WildCat can be converted into a causal attention approximation via prefix sums or divide and conquer schemes. We view the rigorous theoretical and empirical analysis of causal WildCat as the ideal focal point of future work.

---

> > ### Author Rebuttal · Reviewer_Yjbf · 2026-04-03
> >
> > I maintain the positive score.

---

### Official Review · Reviewer_heev · 2026-03-13

**Soundness:** 3
**Presentation:** 3
**Significance:** 3
**Originality:** 4
**Overall Recommendation:** 5
**Confidence:** 3

**Summary:**

This paper proposes WILDCAT, a new method for approximating softmax attention more efficiently. The core idea is to approximate the unnormalized attention matrix with a Nyström-style construction, using representative keys selected by a randomized low-rank kernel approximation procedure based on randomly pivoted Cholesky, and then use this approximation to compute attention at lower cost. The paper claims near-linear runtime together with strong approximation guarantees, including explicit super-polynomial error decay results under bounded-input assumptions and additional extensions beyond that regime. At a high level, the theoretical contribution is to show that this particular low-rank approximation can remain accurate while substantially reducing the cost of attention. The empirical section shows WILDCAT yields competitive results on image generation and image classification when compared to existing fast attention techniques, both in terms of speedup and performance retention.

**Compliance With Llm Reviewing Policy:**

Affirmed.

**Final Justification:**

I remain in favor of acceptance. The paper is technically solid and original, with clear presentation and strong empirical results. My main concerns were about separating the source of the speedups and clarifying how to interpret the broader guarantees, and the rebuttal addressed both. Overall, the response reinforced my initial view, and I continue to see this as a meaningful contribution to efficient attention.

**Key Questions For Authors:**

1. What steps were taken to isolate the source of the empirical speedups, separating the contributions of the approximation method itself from binning, parallelization, and other implementation-level optimizations?
2. Could the authors clarify how to interpret the broader Corollary 1/2 guarantees in practical terms, and how restrictive those conditions are for the experimental settings considered here?

**Limitations:**

yes

**Strengths And Weaknesses:**

Strengths
- The paper addresses an important and highly relevant problem. Reducing the quadratic cost of attention remains a central challenge in modern deep learning, and the paper does a good job motivating why this matters both theoretically and in practice.
- The proposed method is technically interesting and appears meaningfully distinct from standard fast-attention approaches. In particular, the combination of a Nyström-style approximation with randomly pivoted Cholesky gives the paper a clear technical identity and makes the approach stand out from more common low-rank or kernel-based alternatives.
- The paper is clearly written and generally easy to follow. The presentation is well organized, the notation is introduced carefully, and the progression from definitions to lemmas to the main results makes the technical development easier to understand.
- The method performs well across several settings, and the results suggest that it can deliver a good accuracy-efficiency tradeoff in practice. In particular, the reported results on BigGAN and T2T-ViT are very competitive and help support the paper’s practical relevance. This gives a strong and broad empirical evaluation.

Weaknesses
- The paper mixes algorithmic and systems contributions, but the empirical section does not cleanly isolate them. Since the method includes both a new approximation scheme and GPU-oriented implementation choices such as binning and parallelization, it is hard to determine how much of the wall-clock gain comes from the approximation itself versus implementation-specific optimizations. More ablations would make this section stronger, e.g., separating the effects of rank choice, binning, and kernel-level engineering.
- The strongest theoretical guarantee holds under a bounded-input regime, so its interpretation outside that setting could be discussed more explicitly. The paper does extend its theory beyond the bounded-input regime, but the broader guarantees are stated under more technical asymptotic conditions and are less easy to interpret than the main headline result. A brief discussion of how these conditions relate to the experimental settings would strengthen the theoretical section.
- The main guarantee is stated in max-entry norm, which is meaningful but quite specific: it gives coordinate-wise control of the output rather than a more global notion of approximation quality. It is weaker than operator- or Frobenius-norm control and only translates to those norms with dimension-dependent factors. Since downstream quality is often more closely tied to preserving full token representations than worst-coordinate error, a brief discussion of how this guarantee should be interpreted and compared to prior spectral/operator-norm results would improve clarity.

Overall, I found this to be a strong and well-executed paper. The problem is important, the proposed method is technically interesting and appears meaningfully distinct from prior approaches, the paper is clearly written, and the empirical results are convincing across several settings. My concerns are mainly about how some of the theoretical claims should be interpreted and compared to prior work. Taken together, I believe the paper makes a meaningful contribution to the literature on fast attention.

---

> ### Author Rebuttal · Authors · 2026-03-30
>
> Thank you for your positive and constructive feedback. We are delighted that you found our problem important and highly relevant, our solution technically interesting and meaningfully distinct, our empirical evaluation broad and strong, and our writing clear. Below we address each opportunity for improvement.
>
> **Connecting theory and practice**
>
> > A brief discussion of how these conditions relate to the experimental settings would strengthen the theoretical section. […] Could the authors clarify how to interpret the broader Corollary 1/2 guarantees in practical terms […]?
> >
>
> Following your suggestion, we will include concrete interpretations of our results along with numerical evidence that the assumptions underlying our guarantees are satisfied in real transformer layers used in practice. For example, Cor. 2 implies super-polynomially decaying error rates when the model dimension $d$ and the rescaled input entries $\gamma(n) := \frac{R_{\mathbf Q} R_{\mathbf K}}{\sqrt{d} \log(n)}$ are bounded. We compute $\gamma(n)$ from the queries and keys of the Qwen/Qwen2.5-7B-Instruct model for the first $n$ tokens of sequences sampled from the LongBench-E qasper_e dataset and observe that $\gamma(n)$ is *decreasing* with $n$ and hence also bounded:
>
> | n | gamma |
> | --- | --- |
> | 4 | 14.95 |
> | 16 | 9.55 |
> | 64 | 7.48 |
> | 256 | 6.70 |
> | 1024 | 6.23 |
> | 4096 | 5.86 |
> | 16384 | 5.63 |
>
> In addition, when one is focused on scaling the context length of a specific model, the dimension $d$ is always fixed and hence bounded. The assumptions to Cor. 2 are therefore satisfied in our KV cache compression experiment.
>
> In the revision, we can also complement this empirical evidence with the following theoretical intuition, describing sufficient conditions under which $\gamma(n)$ is necessarily bounded. If the queries and keys have i.i.d. sub-Gaussian entries, then, with probability 1, $\gamma(n) \in O\left(\frac{\sqrt{d}}{\log(n)} + \frac{1}{\sqrt{d}}\right)$.
>
> **New ablations**
>
> > More ablations would make this section stronger, e.g., separating the effects of rank choice, binning, and kernel-level engineering.
> >
>
> Following your recommendation we have conducted several new ablations exploring the impact of varying n, r, and B. The first benchmarks WildCat against the highly optimized FlashAttention 2 (FA) kernel with i.i.d. Gaussian (Q,K,V) inputs at various sequence lengths. With fixed $r = 32$ and $B = 16$, we observe that FA runtime increases quadratically while WildCat runtimes increase linearly, even as WildCat approximation error improves with n. We further believe that hardware-aware optimizations used by FA could also be integrated into WildCat, achieving even greater speed-ups.
>
> | seq_len | flash (ms) | wildcat (ms) | max_abs_error |
> | --- | --- | --- | --- |
> | 1024 | 4.55 | 0.56 | 0.5020 |
> | 2048 | 17.91 | 8.79 | 0.3501 |
> | 4096 | 81.09 | 10.33 | 0.2783 |
> | 8192 | 317.70 | 20.51 | 0.1722 |
> | 16384 | 1158.68 | 96.29 | 0.0908 |
> | 32768 | 4709.06 | 188.92 | 0.0767 |
>
> Improved scaling can be observed for any fixed choice of r and B, as can be seen at https://imgur.com/a/PnitSNx. Varying r and B, as well as the used device, affects the point at which the runtimes of Flash Attention and WildCat intersect. Generally, increasing r/B increases runtime but decreases error, and WildCat is faster than FA, even for relatively short sequences, see https://imgur.com/a/subjJgS. We will include these ablations in a revision of our work. We hope this addresses your question and strengthens our work.
>
> **Improvements over the best known operator and max row norm guarantees**
>
> > a brief discussion of how this guarantee should be interpreted and compared to prior spectral/operator-norm results would improve clarity
> >
>
> While we followed Carrell et al. "Low-Rank Thinning" (2025) in stating entrywise guarantees, our results also improve upon the best known guarantees for the operator norm (used by KDEformer and Hyperattention) and the max L2 row norm (used by BalanceKV). Even with the lossy conversion,  $\lVert O - \widehat O\rVert_{op} \leq \sqrt{nd} \lVert O - \widehat O \rVert_{max}$ and  $\lVert O - \widehat O\rVert_{2,\infty} \leq \sqrt{d} \lVert O - \widehat O \rVert_{max}$, the super-polynomial decay of $\lVert O - \widehat O \rVert_{max}$ for WildCat implies super-polynomial decay in near-linear time for the other norms as well. Meanwhile, the guarantees for prior work remain exactly as in Table 1 with each achieving at best sub-polynomial, near-constant decay in near-linear time.

---

> > ### Author Rebuttal · Reviewer_heev · 2026-04-03
> >
> > Thank you for your thorough response. My concerns have been addresses and I will keep my positive score.

---

### Decision · Program_Chairs · 2026-04-30

**Decision:**

Accept (regular)

**Comment:**

This paper proposes WildCat, a near-linear attention approximation using randomly pivoted Cholesky for coreset selection, achieving super-polynomial error decay $O(n^{-\sqrt{\log \log n}})$ in near-linear time.  The theoretical contributions are well-grounded and the claims are thoroughly validated through empirical experiments.

All four reviewers raised concerns regarding the separation of algorithmic and systems contributions, the practical interpretation of the bounded-input assumption, and the comparison to alternative norm guarantees. The authors addressed all of these points in the rebuttal with new ablations and numerical evidence, and all reviewers marked their concerns as fully resolved, unanimously recommending acceptance (final scores: 5, 4, 4, 5).

The AC agrees with the reviewers' assessment and recommends this paper for acceptance.